# Using Partial Monotonicity in Submodular Maximization

**Loay Mualem**
Department of Computer Science
University of Haifa
Haifa 3303221, Israel
loaymua@gmail.com

**Moran Feldman**
Department of Computer Science
University of Haifa
Haifa 3303221, Israel
moranfe@cs.haifa.ac.il

## Abstract

Over the last two decades, submodular function maximization has been the workhorse of many discrete optimization problems in machine learning applications. Traditionally, the study of submodular functions was based on *binary* function properties, but recent works began to consider *continuous* function properties such as the submodularity ratio and the curvature. The monotonicity property of set functions plays a central role in submodular maximization. Nevertheless, no continuous version of this property has been suggested to date (as far as we know), which is unfortunate since submoduar functions that are almost monotone often arise in machine learning applications. In this work we fill this gap by defining the *monotonicity ratio*, which is a continuous version of the monotonicity property. We then show that for many standard submodular maximization algorithms one can prove new approximation guarantees that depend on the monotonicity ratio; leading to improved approximation ratios for the common machine learning applications of movie recommendation, quadratic programming, image summarization and ride-share optimization.

## 1 Introduction

Over the last two decades, submodular function maximization has been the workhorse of many discrete optimization problems in machine learning applications such as data summarization [17, 19, 31, 32, 41, 50], social graph analysis [45], adversarial attacks [36], dictionary learning [15], sequence selection [42, 51], interpreting neural networks [18] and many more. Traditionally, the study of submodular functions was based on *binary* properties of functions. A function can be either submodular or non-submodular, monotone or non-monotone, etc. Such properties are simple, but they have an inherit weakness—if an algorithm assumes functions that have a particular property, then it provides no guarantee for functions that violate this property, even if the violation is slight.

Given the above situation, recent works began to consider *continuous* versions of function properties. Probably the most significant among these continuous versions so far are the submodularity ratio and the curvature. The submodularity ratio (originally defined by Das and Kempe [16]) is a parameter $\gamma \in [0, 1]$ replacing the binary submodularity property that a set function can either have or not have. A value of 1 corresponds to a fully submodular function, and lower values of $\gamma$ represent some violation of submodularity (the worse the violation, the lower $\gamma$). Similarly, the curvature (defined by Conforti and Cornuéjol [13]) is a parameter $c \in [0, 1]$ replacing the binary linearity property that a set function can either have or not have. A value of 1 corresponds to a fully linear function, and lower values of $c$ represent some violation of linearity.

A central conceptual contribution of Das and Kempe [16] was that they were able to demonstrate that continuous function properties further extend the usefulness of submodular maximization to new

36th Conference on Neural Information Processing Systems (NeurIPS 2022).

machine learning applications (such as subset selection for regression and dictionary selection). This has motivated a long list of works on such properties (see [3, 25, 26, 29, 34] for a few examples), including works that combine both the submodularity ratio and the curvature (see, e.g., [3]). However, to the best of our knowledge, no continuous version of the binary monotonicity property has been suggested so far.[1] See Appendix A for additional related work.

We note that the monotonicity property of set functions plays a central role in submodular maximization, and basically every problem in this field has been studied for both monotone and non-monotone objective functions. Naturally, monotone objective functions enjoy improved approximation guarantees compared to general functions, and it is natural to ask how much of this improvement applies also to functions that are almost monotone (in some sense). Since such functions often arise in machine learning applications when a diversity promoting component is added to a basic monotone objective, obtaining better guarantees for them should strongly enhance the usefulness of submodular maximization as a tool for many machine learning applications.

Formally, a non-negative set function $f \colon 2^{\mathcal{N}} \to \mathbb{R}_{\geq 0}$ over a ground set $\mathcal{N}$ is (increasingly) *monotone* if $f(S) \subseteq f(T)$ for every $S \subseteq T \subseteq \mathcal{N}$. Similarly, we define the *monotonicity ratio* of such a function $f$ as the maximum value $m \in [0, 1]$ such that $m \cdot f(S) \leq f(T)$ for every two sets $S \subseteq T \subseteq \mathcal{N}$. Equivalently, one can define the monotonicity ratio $m$ by $m \triangleq \min_{S \subseteq T \subseteq \mathcal{N}}[f(T)/f(S)]$, where the ratio $f(T)/f(S)$ is assumed to be 1 whenever $f(S) = 0$. Intuitively, the monotonicity ratio measures how much of the value of a set $S$ can be lost when additional elements are added to $S$. One can view $m$ as the distance of $f$ from monotonicity. In particular, $m = 1$ if and only if $f$ is monotone.

Our main contribution in this paper is demonstrating the usefulness of the monotonicity ratio in machine learning applications, which we do in two steps.

- First, we show (in Sections 3, 4 and 5) that for many standard submodular maximization algorithms one can prove new approximation guarantees that depend on the monotonicity ratio. These approximation guarantees interpolate between the known approximation ratios of these algorithms for monotone and non-monotone submodular functions.

- Then, using the above new approximation guarantees, we derive new approximation ratios for the standard applications of movie recommendation, quadratic programming, image summarization and ride-share optimization. Our guarantees improve over the state-of-the-art for most values of the problems' parameters. See Section 6 for more detail.

**Remark.** Computing the monotonicity ratio $m$ of a given function seems to be difficult. Thus, the algorithms we analyze avoid assuming access to $m$, and the value of $m$ is only used in the analyses of these algorithms. Nevertheless, in the context of particular applications, we are able to bound $m$, and plugging this bound into our general results yields our improved guarantees for these applications.

## 1.1 Our Results

Given a ground set $\mathcal{N}$, a set function $f \colon 2^{\mathcal{N}} \to \mathbb{R}$ is submodular if $f(S \cup \{u\}) - f(S) \geq f(T \cup \{u\}) - f(T)$ for every two sets $S \subseteq T \subseteq \mathcal{N}$ and element $u \in \mathcal{N} \setminus T$. Submodular maximization problems ask to maximize such functions subject to various constraints. To allow for multiplicative approximation guarantees for these problems, it is usually assumed that the objective function $f$ is non-negative. Accordingly, we consider in this paper the following three basic problems.

- Given a non-negative submodular function $f \colon 2^{\mathcal{N}} \to \mathbb{R}$, find a set $S \subseteq \mathcal{N}$ that (approximately) maximizes $f$. This problem is termed "unconstrained submodular maximization", and is studied in Section 3.

- Given a non-negative submodular function $f \colon 2^{\mathcal{N}} \to \mathbb{R}$ and an integer parameter $0 \leq k \leq |\mathcal{N}|$, find a set $S \subseteq \mathcal{N}$ of size at most $k$ that (approximately) maximizes $f$ among such sets. This problem is termed "maximizing a submodular function subject to a cardinality constraint", and is studied in Section 4.

- Given a non-negative submodular function $f \colon 2^{\mathcal{N}} \to \mathbb{R}$ and a matroid $\mathcal{M}$ over the same ground set, find a set $S \subseteq \mathcal{N}$ that is independent in $\mathcal{M}$ and (approximately) maximizes $f$

---

[1]Following the appearance of the pre-print version of this paper, we learned that Iyer defined in his Ph.D. thesis [30] such a property, which is identical to the one we define. However, Iyer only used this property to prove the result appearing below as Theorem 4.1; thus, our work is the first to systematically study this property.

among such sets. This problem is termed "maximizing a submodular function subject to a matroid constraint", and is studied in Section 5 (see Section 5 for the definition of matroids).

We present both algorithmic and inapproximability results for the above problems. Our algorithmic results reanalyze a few standard algorithms, and surprisingly show that almost all these algorithms guarantee an approximation ratio of $m \cdot \alpha_{\text{mon}} + (1 - m) \cdot \alpha_{\text{non-mon}}$, where $m$ is the monotonicity ratio, $\alpha_{\text{mon}}$ is the approximation ratio known for the algorithm when $f$ is monotone, and $\alpha_{\text{non-mon}}$ is the approximation ratio known for the algorithm when $f$ is a general non-negative submodular function.

While the above mentioned algorithmic results lead to our improved guarantees for applications, our inapproximability results represent our main technical contribution. In general, these results are based on the symmetry gap framework of Vondrák [52]. The original version of this framework is able to deal both with the case of general (not necessarily monotone) submodular functions, and with the case of monotone submodular functions; which in our terms correspond to the cases of $m \geq 0$ and $m \geq 1$, respectively. However, to prove our inapproximability results, we had to show that the framework extends to arbitrary lower bounds on $m$, which was challenging because the original proof of the framework is highly based on derivatives of continuous functions. From this point of view, submodularity is defined as having non-positive second-order derivatives, and monotonicity is defined as having non-negative first-order derivatives. However, the definition of the monotonicity ratio cannot be easily restated in terms of derivatives;[2] and thus, handling it required us to come up with a different proof approach.

Interestingly, our results for unconstrained submodular maximization proves that the optimal approximation ratio for this problem does not exhibit a linear dependence on $m$. Thus, the nice linear dependence demonstrated by almost all our algorithmic results is probably an artifact of looking at standard algorithms rather than representing the true nature of the monotonicity ratio, and we expect future algorithms tailored to take advantage of the monotonicty ratio to improve over this linear dependence. The reason that we concentrate in this work on reanalyzing standard submodular maximization algorithms rather than inventing new ones is that we want to stress the power obtained by using the new notion of monotonicity ratio, as opposed to power gained via new algorithmic innovations. This is in line with the research history of the submodularity ratio and the curvature. For both of these parameters, the original works concentrated on reanalyzed the standard greedy algorithm in view of the new suggested parameter; and the invention of algorithms tailored to the parameter was deferred to later works (see [49] and [12] for examples of such algorithms for the curvature and submodularity ratio, respectively).

Over the years, the standard submodular maximization algorithms have been extended and improved in various ways. Some works presented accelerated and/or parallelized versions of these algorithms, while other works generalized the algorithms beyond the realm of set functions (for example, to (DR-)submodular functions over lattices or continuous domains). Since our motivation in this paper is related to the monotonicity ratio, which is essentially independent of the extensions and improvements mentioned above, we mostly analyze the vanilla versions of all the algorithms considered. This keeps our analyses relatively simple. However, our experiments are based on more state-of-the-art versions of the algorithms. Similarly, many continuous properties (including the submodularity ratio) have weak versions that only depend on the behavior of the function for nearly feasible sets, and immediately enjoy most of the results that apply to the original strong property. The definition of such weak versions is useful for capturing additional application, but often add little from a theoretical perspective. Therefore, in the theoretical parts of this paper we consider only the monotonicity ratio as it is defined above; but for the sake of one of our applications we later define also the natural corresponding weak property.

## 2 Preliminaries and Basic Observations

In this section we define the notation used in this paper, and then state some useful basic observations. Given an element $u \in \mathcal{N}$ and a set $S \subseteq \mathcal{N}$, we use $S + u$ and $S - u$ as shorthands for $S \cup \{u\}$ and $S \setminus \{u\}$. Additionally, given a set function $f : 2^{\mathcal{N}} \to \mathbb{R}$, we define $f(u \mid S) \triangleq f(S + u) - f(S)$—this value is known as the marginal contribution of $u$ to $S$ with respect to $f$. Similarly, given an additional

---

[2]To see why that is the case, notice that a function can have a monotonicity ratio close to 1, even in the presence of very negative derivatives, as long as these derivatives do not occur over too long sections.

set $T \subseteq \mathcal{N}$, we define $f(T \mid S) \triangleq f(S \cup T) - f(S)$. We also use $\mathbf{1}_S$ to denote the characteristic vector of the set $S$, i.e., a vector in $[0, 1]^{\mathcal{N}}$ that has 1 in the coordinates corresponding to elements that appear in $S$ and 0 to the other coordinates. Finally, if $f$ is non-negative, we say that it is $m$-monotone if its monotonicity ratio is at least $m$; and given an event $\mathcal{E}$, we denote by $\mathbf{1}[\mathcal{E}]$ the indicator of this event, i.e., a random variable that takes the value 1 when the event happens, 0 otherwise .

Next, we present a well-known continuous extension of set functions. Given a set function $f \colon 2^{\mathcal{N}} \to \mathbb{R}$, its *multilinear extension* is a function $F \colon [0, 1]^{\mathcal{N}} \to \mathbb{R}$ defined as follows. For every vector $\mathbf{x} \in [0, 1]^{\mathcal{N}}$, let $\texttt{R}(\mathbf{x})$ to be a random subset of $\mathcal{N}$ that includes every element $u \in \mathcal{N}$ with probability $x_u$, independently. Then, $F(\mathbf{x}) = \mathbb{E}[f(\texttt{R}(\mathbf{x}))]$. Another extension of set functions is central to the proof of the next lemma, which generalizes Lemma 2.2 of [6]—see Appendix B for the proof.

**Lemma 2.1.** *Let $f \colon 2^{\mathcal{N}} \to \mathbb{R}_{\geq 0}$ be a non-negative $m$-monotone submodular function. For every deterministic set $O \subseteq \mathcal{N}$ and random set $D \subseteq \mathcal{N}$, $\mathbb{E}[f(O \cup D)] \geq (1 - (1 - m) \cdot \max_{u \in \mathcal{N}} \Pr[u \in D]) \cdot f(O)$.*

We conclude this section with the following observation, which we view as evidence that the class of non-negative $m$-monotone functions is a natural class for every $m \in [0, 1]$.

**Observation 2.2.** *For every two non-negative $m$-monotone functions $f, g \colon 2^{\mathcal{N}} \to \mathbb{R}_{\geq 0}$ and constant $c \geq 0$, the following functions are also $m$-monotone: (i) $h(S) = f(S) + g(S)$, (ii) $h(S) = f(S) + c$, and (iii) $h(S) = c \cdot f(S)$.*

## 3 Unconstrained Maximization

Recall that in the unconstrained submodular maximization problem, we are given a non-negative submodular function $f \colon 2^{\mathcal{N}} \to \mathbb{R}_{\geq 0}$, and the objective is to find a set $S \subseteq \mathcal{N}$ that (approximately) maximizes $f(S)$. Buchbinder et al. [7] gave the first $1/2$-approximation algorithm for this problem, known as the (randomized) double greedy algorithm. As its name suggests, double greedy maintains two solutions: one starting as the empty set, and one starting as the entire ground set. Then, it considers all elements, and greedily decides for each element either to add it to the originally empty set, or remove it from the other set. When the algorithm terminates, the two sets are identical, and their common value is the output of the algorithm. The $1/2$-approximation guarantee of double greedy is known to be optimal in general due to a matching inapproximability result due to Feige et al. [21]. Nevertheless, in this section we determine the extent to which one can improve over this guarantee as a function of the monotonicity ratio $m$ of $f$.

**Theorem 3.1.** *The double greedy algorithm of Buchinder et al. [7] guarantees $[1/(2 - m)]$-approximation for unconstrained submodular maximization, and no polynomial time algorithm obtains an approximation ratio of $1/(2 - m) + \varepsilon$ for any constant $\varepsilon > 0$.*[3]

Interestingly, Theorem 3.1 shows that the optimal approximation ratio for unconstrained submodular maximization does not have a linear dependence on $m$. The first part of Theorem 3.1 is proved in Appendix C.1. Below, we concentrate on proving the second part of Theorem 3.1. We do this using a generalization of the symmetry gap framework of Vondrák [52] that is informally stated as Theorem 3.2 (see Appendix C.2 for the formal statement of the theorem). The fractional solution mentioned in this informal statement is evaluate using the multilinear extension of the submodular objective function.

**Theorem 3.2.** *Consider a non-negative $m$-monotone submodular function $f$ and a collection $\mathcal{F} \subseteq 2^{\mathcal{N}}$ of feasible sets such that the problem $\max\{f(S) \mid S \in \mathcal{F}\}$ is symmetric with respect to some group $\mathcal{G}$ of permutations over $\mathcal{N}$. If the best fractional solution for this problem which is symmetric with respect to $\mathcal{G}$ is worse by a factor of $\gamma$ compared to the optimal solution, then we say that the problem has a symmetry gap of $\gamma$. In this case, exponentially many value oracle queries are required to obtain $(1 + \varepsilon)\gamma$-approximation for the class of problems $\max\{\tilde{f}(S) \mid S \in \tilde{\mathcal{F}}\}$ in which $\tilde{f}$ is a non-negative $m$-monotone submodular function, and $\tilde{\mathcal{F}}$ is some generalization of $\mathcal{F}$ (in particular, if $\mathcal{F}$ is a matroid/cardinality constraint, then so is $\tilde{\mathcal{F}}$).*

---

[3]In the second part of Theorem 3.1, like in all the other inapproximability results in this paper, we make the standard assumption that the objective function $f$ can be accessed only through a value oracle that given a set $S \subseteq \mathcal{N}$ returns $f(S)$.

To use Theorem 3.2, we need to define a submodular maximization problem with a significant symmetry gap. Let us choose $\mathcal{N} = \{u, v\}$, $f(S) = m \cdot \mathbf{1}[S \neq \varnothing] + (1 - m) \cdot (|S| \bmod 2)$ and $\mathcal{F} = 2^{\mathcal{N}}$, where $m$ is an arbitrary constant $m \in [0, 1]$. One can verify that $f$ is submodular and non-negative, that its monotonicity ratio is exactly $m$, and that the problem $\max\{f(S) \mid S \in \mathcal{F}\}$ is symmetric with respect to the group $\mathcal{G}$ of the two possible permutations of $\mathcal{N}$. The following lemma calculates the symmetry gap of this problem, and its proof can be found in Appendix C.3. The second part of Theorem 3.1 follows from this lemma and Theorem 3.2.

**Lemma 3.3.** *The problem* $\max\{f(S) \mid S \in \mathcal{F}\}$ *has a symmetry gap of* $\frac{1}{2-m}$.

# 4 Maximization with a Cardinality Constraint

In this section we consider the problem of maximizing a non-negative submodular function $f : 2^{\mathcal{N}} \to \mathbb{R}_{\geq 0}$ subject to a cardinality constraint. In other words, we are given an integer value $1 \leq k \leq |\mathcal{N}|$, and the objective is to output a set $S \subseteq \mathcal{N}$ of size at most $k$ (approximately) maximizing $f$ among such sets. A standard greedy algorithm for this problem starts with the empty set, and then iteratively adds elements to this set, choosing in each iteration the element whose addition increases the value of the set by the most. When the objective function $f$ is guaranteed to be monotone, it is long known that this greedy algorithm guarantees $(1 - 1/e)$-approximation for the above problem [44], and that this is essentially the best possible for any polynomial time algorithm [43]. However, the greedy algorithm has no constant approximation guarantee when the objective function is not guaranteed to be monotone (see [4] for an example demonstrating this). In Appendix D.1 we prove Theorem 4.1, which generalizes the result of [44], and proves an approximation guarantee for the greedy algorithm that deteriorates gracefully with the monotonicity ratio $m$.

**Theorem 4.1.** *The Greedy algorithm (Algorithm 2) has an approximation ratio of at least $m(1 - 1/e)$ for the problem of maximizing a non-negative $m$-monotone submodular function subject to a cardinality constraint.*

Following a long line of works [35, 22, 46, 52, 6, 20], the state-of-the-art approximation guarantee for the case in which the objective function $f$ is not guaranteed to be monotone is currently $0.385$ [5]. However, the algorithm obtaining this approximation ratio is quite involved, which limits its practicality. Arguably, the state-of-the-art approximation ratio obtained by a "simple" algorithm is the $1/e \approx 0.367$-approximation obtained by an algorithm called Random Greedy, which adds to its solution, in each iteration, a uniformly random element out of the $k$ elements that can (individually) add the most to the value of this solution. Random Greedy has the nice property that for monotone objective functions it recovers the optimal $1 - 1/e$ approximation guarantee. In Appendix D.2 we prove Theorem 4.2, which gives an approximation guarantee for Random Greedy that smoothly changes as a function of $m$ and recovers both the above mentioned $1/e$ and $1 - 1/e$ guarantees.

**Theorem 4.2.** *Random Greedy (Algorithm 3) has an approximation ratio of at least $m(1 - 1/e) + (1 - m) \cdot (1/e)$ for the problem of maximizing a non-negative $m$-monotone submodular function subject to a cardinality constraint.*

There is still a gap between the state-of-the-art $0.385$-approximation for non-monotone objectives and the state-of-the-art inapproximability result due to Oveis Gharan and Vondrák [46], which only shows that no polynomial time algorithm can guarantee a better than roughly $0.491$-approximation. In Appendix D.3 we give Theorem D.4, which uses Theorem 3.2 to prove an inapproximability result that smoothly depends on $m$ and recovers the above mentioned inapproximability results for $m = 0$ and $m = 1$.

To get an intuitive understanding of Theorem D.4, we numerically evaluated it for various values of $m$. The plot obtained in this way appears in Figure 1a. For context, this figure also includes all the other results proved in this section. As is evident from Figure 1a, Theorem D.4 improves over the $1 - 1/e$ inapproximability result of Nemhauser and Wolsey [43] only for $m$ that is smaller than roughly $0.56$. This is surprising since, intuitively, one would expect the best possible approximation ratio to be strictly worse than $1 - 1/e$ for any $m < 1$. However, we were unable to prove an inapproximability that is even slightly lower than $1 - 1/e$ for any $m > 0.56$. Understanding whether this is an artifact of our proof or a real phenomenon is an interesting question that we leave open.

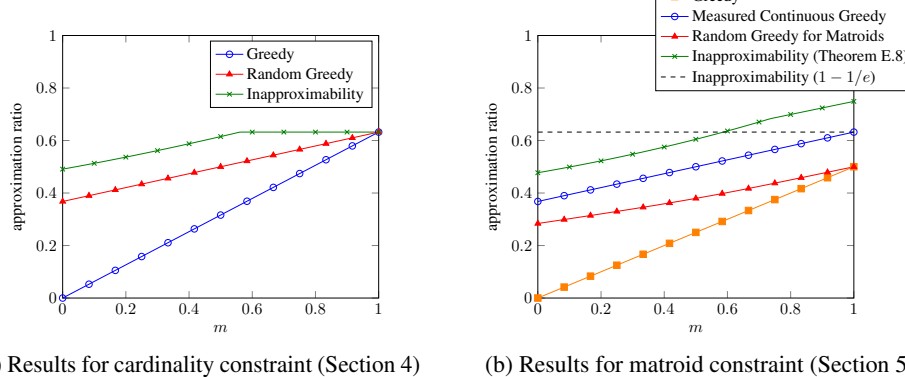

(a) Results for cardinality constraint (Section 4)     (b) Results for matroid constraint (Section 5)

Figure 1: Graphical representation of the results of Sections 4 and 5

## 5   Maximization with a Matroid Constraint

In this section we consider the problem of maximizing a non-negative submodular function subject to a matroid constraint. A matroid $\mathcal{M}$ over the ground set $\mathcal{N}$ is defined as a pair $\mathcal{M} = (\mathcal{N}, \mathcal{I})$, where $\mathcal{I}$ is a non-empty subset of $2^{\mathcal{N}}$ obeying two properties for every two sets $S, T \subseteq \mathcal{N}$: (i) if $S \subseteq T$ and $T \in \mathcal{I}$, then $S \in \mathcal{I}$; and (ii) if $S, T \in \mathcal{I}$ and $|S| < |T|$, then there exists an element $u \in T \setminus S$ such that $S + u \in \mathcal{I}$. A set $S \subseteq \mathcal{N}$ is called *independent* with respect to a matroid $\mathcal{M}$ if it belongs to $\mathcal{I}$ (otherwise, we say that $S$ is dependent with respect to $\mathcal{M}$); and the matroid constraint corresponding to a given matroid $\mathcal{M}$ allows only independent sets with respect to this matroid as feasible solutions. Hence, we can restate the problem we consider in this section in the following more formal way. Given a non-negative submodular function $f: 2^{\mathcal{N}} \to \mathbb{R}_{\geq 0}$ and a matroid $\mathcal{M} = (\mathcal{N}, \mathcal{I})$, output an independent set $S \in \mathcal{I}$ (approximately) maximizing $f$ among all such sets. It is also useful to note that an independent set $S \in \mathcal{I}$ is called a *base* of $\mathcal{M}$ if it is an inclusion-wise maximal independent set, i.e., $S$ is not a subset of any other independent set.

A standard greedy algorithm for the above problem starts with the empty set, and then iteratively adds to it elements, choosing in each iteration the element that increases the value of the solution by the most among the elements whose addition to the solution does not violate the matroid constraint. When the objective function $f$ is guaranteed to be monotone, this greedy algorithm guarantees $1/2$-approximation [24]. Our first result for this section (proved in Appendix E.1) shows how this approximation guarantee changes as a function of $m$ (the greedy algorithm has no constant guarantee for non-monotone functions in this case as well).

**Theorem 5.1.** *The Greedy algorithm (Algorithm 4) has an approximation ratio of at least $m/2$ for maximizing a non-negative $m$-monotone submodular function subject to a matroid constraint.*

The approximation ratio of the greedy algorithm was improved over by the seminal work of Călinescu et al. [9], who described the Continuous Greedy algorithm whose approximation ratio is $1 - 1/e$ when $f$ is monotone; matching the inapproximability result of Nemhauser and Wolsey [43]. In contrast, when $f$ is not guaranteed to be monotone, the approximability of the problem is less well-understood. On the one hand, after a long line of works [35, 22, 46, 52, 20], the state-of-the-art approximation ratio for the problem is $0.385$ [5], but on the other hand, it is only known that no polynomial time algorithm for the problem can guarantee $0.478$-approximation [46].

Unfortunately, the above mentioned state-of-the-art $0.385$-approximation algorithm is quite involved. Therefore, we chose to consider in this work two other algorithms. The first algorithm is Measure Continuous Greedy (due to [22]) which guarantees an approximation ratio of $1/e - o(1) \approx 0.367$. This algorithm performs only slightly worse than the above state-of-the-art, and is a central component of all the currently known algorithms achieving better than $1/e$-approximation. Measured Continuous Greedy is also known to guarantee $(1 - 1/e - o(1))$-approximation when the objective $f$ is monotone, and the next theorem (proved in Appendix E.2) shows that its approximation guarantee changes smoothly with the monotonicity ratio of $f$.

**Theorem 5.2.** *Measured Continuous Greedy (Algorithm 5) has an approximation ratio of at least $m(1 - 1/e) + (1 - m) \cdot (1/e) - o(1)$ for maximizing a non-negative $m$-monotone submodular function subject to a matroid constraint, where the $o(1)$ term diminishes with the ground set's size.*[4]

The other algorithm we consider is an algorithm called Random Greedy for Matroids (due to [6]). Unlike Measured Continuous Greedy and almost all the other algorithms suggested for non-monotone objectives to date, this algorithm is combinatorial, which makes it appealing in practice. It starts with a base solution consisting of dummy elements representing empty slots, and iteratively performs swaps on this base solution in the following way. In each iteration, the algorithm picks a base $M$ maximizing the (individual) marginal value of the elements within it with respect to the current base solution. For every element in $M$, the algorithm identifies a distinct element of the current base solution with which it can be swapped, and then for a uniformly random element of $M$ such a swap is indeed done. Buchbinder et al. [6] proved an approximation ratio of roughly $(1 + e^{-2})/4$ for Random Greedy for Matroids. The next theorem shows how this approximation guarantee improves as a function of the monotonicity ratio. In this theorem we refer to the rank $k$ of the matroid constraint $\mathcal{M}$, which is the size of the largest independent set with respect to this matroid. We also note that the algorithm we analyze is identical to the algorithm of [6] up to two modifications: our algorithm makes more iterations, and it updates the solution in an iteration only when this increases the solution's value.

**Theorem 5.3.** *For every $\varepsilon \in (0, 1)$, Random Greedy for Matroids (Algorithm 6) has an approximation ratio of at least $\frac{1 + m + e^{-2/(1-m)}}{4} - \varepsilon - o_k(1)$ for the problem of maximizing a non-negative $m$-monotone submodular function subject to a matroid constraint (except in the case of $m = 1$ in which the approximation ratio is $1/2 - \varepsilon - o_k(1)$), where $o_k(1)$ represents a term that diminishes with $k$.*

Theorem 5.3 is proved in Appendix E.3. Let us also mention Theorem E.8, which appears in Appendix E.4 and uses Theorem 3.2 to generalize the $0.478$ inapproximability result of Oveis Gharan and Vondrák [46]. To get an intuitive understanding of Theorem E.8, we numerically evaluated it for various values of $m$, and depict the results in Figure 1b. For context, this figure also includes all the other results proved in this section. Somewhat surprisingly, Figure 1b shows that Theorem E.8 does not generalize the $1 - 1/e$ inapproximability result of Nemhauser and Wolsey [43] for monotone functions despite the fact that this inapproximability result holds for every monotonicity ratio $m \in [0, 1]$. This resembles the inability of Theorem D.4 to improve over the same inapproximability result for large values of $m$.

## 6 Applications and Experiment Results

Many machine learning applications require optimization of non-monotone submodular functions subject to some constraint. Unfortunately, such functions enjoy relatively low approximation guarantees. Nevertheless, in many cases the non-monotone objective functions have a significant monotone component that can be captured by the monotonicity ratio. In this section, we discuss two concrete applications with non-monotone submodular objective functions. For each application we provide a lower bound on the monotonicity ratio $m$ of the objective function, which translates via our results from the previous sections into an improved approximation guarantee for the application.

To demonstrate the value of our improved guarantees in experiments, we took the following approach. The output of an approximation algorithm provides an upper bound on the value of the optimal solution for the problem (formally, this upper bound is the value of the output over the approximation ratio of the algorithm). Thus, we plot in each experiment the upper bound on the value of the optimal solution obtained with and without taking into account the monotonicity ratio, which gives a feeling of how the magnitude of our improvements compare to other values of interest (such as the gaps between the performances of the algorithms considered). In Appendix F we give a third application (Image summarization) that we study in the same way; and in Appendix G we lower bound the monotonicity ratio of a fourth application (Ride-Share Optimization).

---

[4]Technically, Measured Continuous Greedy is an algorithm for maximizing the multilinear extesnion of a non-negative submodular function subject to a general solvable down-closed convex body $P$ constraint, and we prove in Appendix E.2 that it guarantees the approximation ratio stated in Theorem 5.2 for this setting. However, this implies the result stated in Theorem 5.2 using a standard reduction (see Appendix E.2 for further detail).

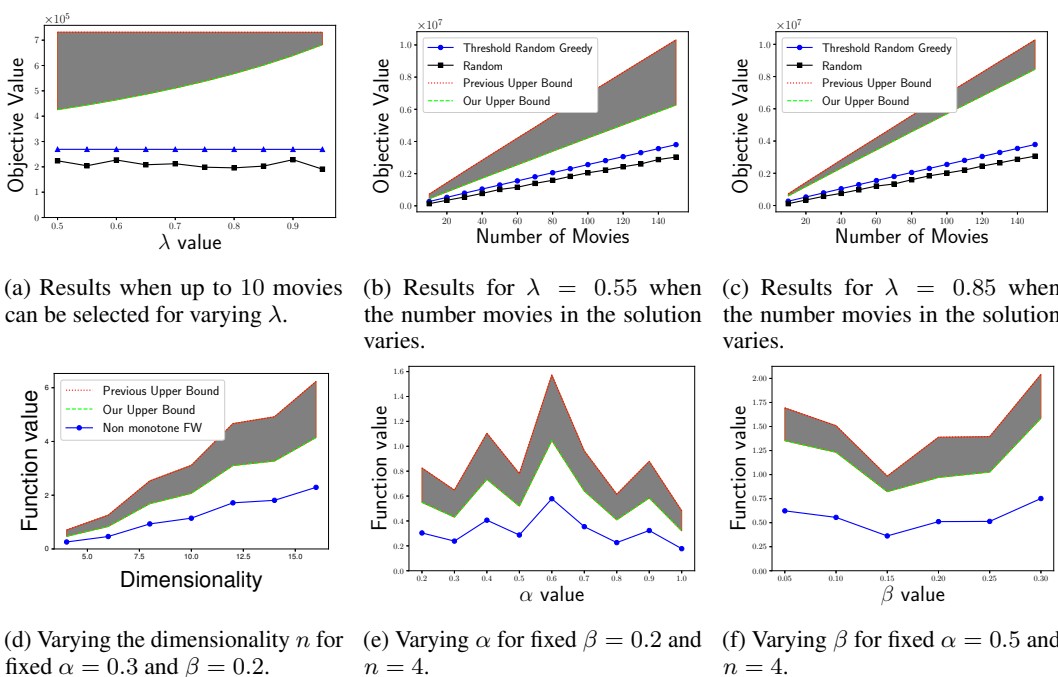

(a) Results when up to 10 movies can be selected for varying $\lambda$.

(b) Results for $\lambda = 0.55$ when the number movies in the solution varies.

(c) Results for $\lambda = 0.85$ when the number movies in the solution varies.

(d) Varying the dimensionality $n$ for fixed $\alpha = 0.3$ and $\beta = 0.2$.

(e) Varying $\alpha$ for fixed $\beta = 0.2$ and $n = 4$.

(f) Varying $\beta$ for fixed $\alpha = 0.5$ and $n = 4$.

Figure 2: Experimental results for Personalized Movie Recommendation (a–c) and Quadratic Programming (d–f). Each plot includes the output of the algorithms we consider as well the previous and improved upper bounds on the optimal value (the area between these two bounds is shaded).

## 6.1 Personalized Movie Recommendation

The first application we consider is Personalized Movie Recommendation. Consider a movie recommendation system where each user specifies what genres she is interested in, and the system has to provide a representative subset of movies from these genres. Assume that each movie is represented by a vector consisting of users' ratings for the corresponding movie. One challenge here is that each user does not necessarily rate all the movies, hence, the vectors representing the movies do not necessarily have similar sizes. To overcome this challenge, a low-rank matrix completion techniques [10] can be performed on the matrix with missing values in order to obtain a complete rating matrix. Formally, given few ratings from $k$ users to $n$ movies we obtain in this way a rating matrix $\mathbf{M}$ of size $k \times n$. Following [40], to score the quality of a selected subset of movies, we use the function $f(S) = \sum_{u \in \mathcal{N}} \sum_{v \in S} s_{u,v} - \lambda \sum_{u \in S} \sum_{v \in S} s_{u,v}$. Here, $\mathcal{N}$ is the set of $n$ movies, $\lambda \in [0, 1]$ is a parameter and $s_{u,v}$ denotes the similarity between movies $u$ and $v$ (the similarity $s_{u,v}$ can be calculated based on the matrix $\mathbf{M}$ in multiple ways: cosine similarity, inner product, etc). Note that the first term in $f$'s definition captures coverage, while the second term captures diversity. Thus, the parameter $\lambda$ denotes the importance of diversity in the returned subset.

One can verify that the above defined function $f$ is non-negative and submodular. The next theorem, proved in Appendix H.1, analyzes the monotonicity ratio of this function. In this theorem we assume that the similarity scores $s_{u,v}$ are non-negative and obey $s_{u,v} = s_{v,u}$ for every $u, v \in \mathcal{N}$. Note that the above mentioned ways to define these scores have these properties. Interestingly, it turns out that the function $f$ is monotone when $\lambda$ is small enough despite the fact that this function is traditionally treated as non-monotone (e.g., in [40, 23]). This is a nice unexpected result of the use of the monotonicity ratio, which required us to really understand the degree of non-monotonicity represented by the objective function.

**Theorem 6.1.** *The objective function $f$ is monotone for $0 \leq \lambda \leq 1/2$ and $2(1 - \lambda)$-monotone for $1/2 \leq \lambda \leq 1$.*

To demonstrate the value of our lower bound on the monotonicity ratio, we followed the experimental setup of [40] and used a subset of movies from the MovieLens data set [28] which includes 10,437

movies. Each movie in this data set is represented by a 25 dimensional feature vector calculated using user ratings, and we used inner products to obtain the similarity values $s_{i,j}$ based on these vectors.

In our experiment we employed accelerated versions of the algorithms analyzed in Section 4 for a cardinality constraint. Specifically, instead of the Greedy algorithm we used Threshold Greedy [1] and Sample Greedy [39]; and instead of Random Greedy we used a threshold based version of this algorithm due to [8] that we refer to as Threshold Random Greedy (Algorithm 6 in [8]). All three algorithms had almost identical performance in our experiments (see Appendix I), thus, to avoid confusion, in Figure 2 we draw only the output of Threshold Random Greedy.

Each plot of Figure 2 depicts the outputs of Threshold Random Greedy and a scarecrow algorithm called Random that simply outputs a random subset of movies of the required size. Each point in the plots represents the average value of the outputs of 10 executions of these algorithms. We also depict in each plot the upper bound on the value of the optimal solution based on the general approximation ratio of Random Greedy and the improved approximation ratio implied by Theorems 4.2 and 6.1—the area between the two upper bounds is shaded. In Figure 2a we plot these values for the case in which we asked the algorithms to pick at most 10 movies, and we vary the parameter $\lambda$. In Figures 2b amd 2c we plotted the same values for a fixed parameter $\lambda$, while varying the maximum cardinality (number of movies) allowed for the output set. Since the height of the shaded area is on the same order of magnitude as the values of the solutions produced by Threashold Random Greedy (especially when $\lambda$ is close to $1/2$), our results demonstrate that the improved upper bound we are able to prove is much tighter than the state-of-the-art. Furthermore, our improved upper bound shows that the gap between the empirical outputs of Threshold Random Greedy and Random is much more significant as a percentage of the value of the optimal solution than one could believe based on the weaker bound.

## 6.2 Quadratic Programming

Consider the function

$$F(\mathbf{x}) = \frac{1}{2}\mathbf{x}^T\mathbf{H}\mathbf{x} + \mathbf{h}^T\mathbf{x} + c \ . \tag{1}$$

By choosing appropriate matrix $\mathbf{H}$, vector $\mathbf{h}$ and scalar $c$, this function can be made to have various properties. Specifically, we would like to make it non-negative and DR-submodular (DR-submodularity is an extension of submodularity to continuous functions—see Appendix J for more detail). Our goal in this section is to maximize $F$ under a polytope constraint given by $P = \{\mathbf{x} \in \mathbb{R}^n_{\geq 0} \mid A\mathbf{x} \leq \mathbf{b}, \mathbf{x} \leq \mathbf{u}, A \in \mathbb{R}^{m \times n}_{\geq 0}, \mathbf{b} \in \mathbb{R}^m_{\geq 0}\}$ for some dimensions $n$ and $m$.

Following Bian et al. [2], we set $m = n$, choose the matrix $\mathbf{H} \in \mathbb{R}^{n \times n}$ to be a randomly generated symmetric matrix whose entries are drawn uniformly at random (and independently) from $[-1, 0]$, and choose $\mathbf{A} \in \mathbb{R}^{m \times n}$ to be a randomly generated matrix whose entries are drawn uniformly at random from $[v, v + 1]$ for $v = 0.01$ (this choice of $v$ guarantees that the entries of $A$ are strictly positive). We also set $\mathbf{b} = \bar{1}$ (i.e., $\mathbf{b}$ is the all ones vector), and $\mathbf{u}$ to be the upper bound on $P$ given by $u_j = \min_{j \in [m]} b_i / A_{i,j}$ for every $j \in [n]$. Finally, we set $\mathbf{h} = -\beta \cdot \mathbf{H}^T\mathbf{u}$ for a parameter $\beta > 0$.

The non-positivity of $\mathbf{H}$ guarantees that $f$ is DR-submodular. To make sure that $f$ is also non-negative, the value of $c$ should be at least $-\min_{\bar{0} \leq \mathbf{x} \leq \mathbf{u}} \frac{1}{2}\mathbf{x}^T\mathbf{H}\mathbf{x} + \mathbf{h}^T\mathbf{x}$ (where $\bar{0}$ is the all zeros vector). This value can be approximately obtained by using QUADPROGIP[5] [53]. Let the value of this minimum be $M$; then we set $c = -M + \alpha|M|$ for some parameter $\alpha > 0$.

The definition of the monotonicity ratio can be extend to the continuous setting we consider in this section using the formula $m = \inf_{\bar{0} \leq \mathbf{x} \leq \mathbf{y} \leq \mathbf{u}} \frac{F(\mathbf{y})}{F(\mathbf{x})}$, where the ratio $F(\mathbf{y})/F(\mathbf{x})$ is understood as 1 whenever $F(\mathbf{x}) = 0$. The following theorem analyzes the monotonicity ratio of the function $F$ given in Equation (1) based on this definition. The proof of this theorem can be found in Appendix H.2.

**Theorem 6.2.** *For $\beta \in (0, 1/2)$, the objective function $F$ given by Equation* (1) *is $\frac{(1-2\beta)\cdot\alpha}{1+\alpha}$-monotone. Furthermore, when $\min_{\bar{0} \leq x \leq \mathbf{u}}(\frac{1}{2}\mathbf{x}^T\mathbf{H}\mathbf{x} + \mathbf{h}\mathbf{x}) \geq 0$, $F$ is even $(1 - 2\beta)$-monotone.*

We applied the Non-monotone Frank-Wolfe algorithm of Bian et al. [2] to the above defined optimization problem (we refer the reader to Appendix J for further detail about this algorithm and

---

[5]We have used IBM CPLEX optimization studio `https://www.ibm.com/products/ilog-cplex-optimization-studio`.

its analysis). Figures 2d, 2e and 2f depict the results we obtained. Specifically, Figure 2d shows the value of the solution obtained by Non-monotone Frank-Wolfe for $\alpha = 0.3$ and $\beta = 0.2$ as the dimensionality $n$ varies. The shaded area is the area between the previous upper bound on the optimal value, and our upper bound that takes advantage on the monotonicity ratio bound given by Theorem 6.2. Figures 2e and 2f are similar, but they fix the dimensionality $n$ to be 4, and vary $\alpha$ or $\beta$ instead. Let us discuss now some properties of Figures 2d, 2e and 2f. (i) Each data point in these figures corresponds to a single instance drawn from the distribution described above. This implies that the plots in these figures vary for different runs of our experiment, but the plots that we give represent a (single) typical run. (ii) The size of the the shaded area depends on $\alpha$ and $\beta$, but also on the sign of $\min_{\bar{0} \le \mathbf{x} \le \mathbf{u}}(\frac{1}{2}\mathbf{x}^T\mathbf{H}\mathbf{x} + \mathbf{h}\mathbf{x})$. This is the reason that this size behaves somewhat non-continuously in Figure 2f. Interestingly, the sign of this minimum is mostly a function of $\beta$. In other words, there are values of $\beta$ for which the minimum is non-negative with high probability, and other values for which the minimum is negative with high probability. (iii) One can see that the use of the monotonicity ratio significantly improves the upper bound on the optimal value, especially when $\min_{\bar{0} \le \mathbf{x} \le \mathbf{u}}(\frac{1}{2}\mathbf{x}^T\mathbf{H}\mathbf{x} + \mathbf{h}\mathbf{x})$ happens to be non-negative.

## 7    Conclusion

In this paper we have defined the monotonicity ratio, analyzed how the approximation ratios of standard submodular maximization algorithms depend on this ratio, and then demonstrated that this leads to improved approximation guarantees for the applications of movie recommendation, image summarization and quadratic programming. We believe that the monotonicity ratio is a natural parameter of submodular maximization problems, refining the binary distinction between monotone and non-monotone objective functions and improving the power of submodular maximization tools in machine learning applications. Thus, we hope to see future work towards understanding the optimal dependence on $m$ of the approximation ratios of various submodular maximization problems.

An important take-home message from our work is that, at least in the unconstrained submodular maximization case, the optimal algorithm has an approximation ratio whose dependence on $m$ is non-linear. Such algorithms are rarely obtained using current techniques, which might be one of the reasons why these techniques have so far failed to obtain tight approximation guarantees for constrained non-monotone submodular maximization.

### Funding Transparency Statement

**Funding in direct support of this work:**   This research was supported in part by Israel Science Foundation (ISF) grant no. 459/20.

**Additional revenues related to this work:**   None

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
