# A   Additional Related Work

Lin and Bilmes [37] described an algorithm that takes advantages of a continuous partial monotonicity property, but unlike the monotonicity ratio, their property was defined in terms of the particular submodular objective they were interested in. More recently, Cui et al. [14] considered a weaker, but still binary, version of monotonicity called weak-monotonicity.

# B   Proof of Lemma 2.1

In this section we prove Lemma 2.1.

**Lemma 2.1.** *Let $f\colon 2^{\mathcal{N}} \to \mathbb{R}_{\geq 0}$ be a non-negative $m$-monotone submodular function. For every deterministic set $O \subseteq \mathcal{N}$ and random set $D \subseteq \mathcal{N}$, $\mathbb{E}[f(O \cup D)] \geq (1 - (1 - m) \cdot \max_{u \in \mathcal{N}} \Pr[u \in D]) \cdot f(O)$.*

To prove this lemma we first have to define the Lovász extension of set functions. The *Lovász extension* of a set function $f\colon 2^{\mathcal{N}} \to \mathbb{R}$ is a function $\hat{f}\colon [0,1]^{\mathcal{N}} \to \mathbb{R}$ defined as follows. For every vector $\mathbf{x} \in [0,1]^{\mathcal{N}}$,

$$\hat{f}(\mathbf{x}) = \int_0^1 f(T_\lambda(\mathbf{x}))d\lambda \ ,$$

where $T_\lambda(\mathbf{x}) \triangleq \{u \in \mathcal{N} \mid x_u \geq \lambda\}$. The Lovász extension of a submodular function is known to be convex. More important for us is the following known lemma regarding this extension. This lemma stems from an equality, proved by Lovász [38], between the Lovász extension of a submodular function and another extension known as the convex closure.

**Lemma B.1.** *Let $f\colon 2^{\mathcal{N}} \to \mathbb{R}$ be a submodular function, and let $\hat{f}$ be its Lovász extension. For every $\mathbf{x} \in [0,1]^{\mathcal{N}}$ and random set $D_{\mathbf{x}} \subseteq \mathcal{N}$ obeying $\Pr[u \in D_{\mathbf{x}}] = x_u$ for every $u \in \mathcal{N}$ (i.e., the marginals of $D_{\mathbf{x}}$ agree with $\mathbf{x}$), $\hat{f}(\mathbf{x}) \leq \mathbb{E}[f(D_{\mathbf{x}})]$.*

Using the last lemma, we can now prove Lemma 2.1.

*Proof of Lemma 2.1.* Let $\mathbf{x}$ be the vector of marginals of $O \cup D$, i.e., $x_u = \Pr[u \in O \cup D]$ for every $u \in \mathcal{N}$. Then, by Lemma B.1,

$$
\begin{aligned}
\mathbb{E}[f(O \cup D)] \geq \hat{f}(\mathbf{x}) &= \int_0^1 f(T_\lambda(\mathbf{x}))d\lambda \\
&= \int_0^{\max_{u \in \mathcal{N}} \Pr[u \in D]} f(T_\lambda(\mathbf{x}))d\lambda + \int_{\max_{u \in \mathcal{N}} \Pr[u \in D]}^1 f(T_\lambda(\mathbf{x}))d\lambda \\
&= \int_0^{\max_{u \in \mathcal{N}} \Pr[u \in D]} f(O \cup T_\lambda(\mathbf{x}))d\lambda + (1 - \max_{u \in \mathcal{N}} \Pr[u \in D]) \cdot f(O) \ ,
\end{aligned}
$$

where the last equality holds since the elements of $O$ appear in $T_\lambda(\mathbf{x})$ for every $\lambda \in [0,1]$, and no other element appears in $T_\lambda(\mathbf{x})$ when $\lambda > \Pr[u \in D]$. Using the definition of the monotonicity ratio, the expression $f(O \cup T_\lambda(\mathbf{x}))$ on the rightmost side of the previous equation can be lower bounded by $m \cdot f(O)$, which yields

$$
\begin{aligned}
\mathbb{E}[f(O \cup D)] &\geq \int_0^{\max_{u \in \mathcal{N}} \Pr[u \in D]} m \cdot f(O)d\lambda + (1 - \max_{u \in \mathcal{N}} \Pr[u \in D]) \cdot f(O) \\
&= m \cdot \max_{u \in \mathcal{N}} \Pr[u \in D] \cdot f(O) + (1 - \max_{u \in \mathcal{N}} \Pr[u \in D]) \cdot f(O) \\
&= (1 - (1 - m) \cdot \max_{u \in \mathcal{N}} \Pr[u \in D]) \cdot f(O) \ . \qquad \square
\end{aligned}
$$

# C   Proofs of Section 3

In this section we give the proofs of Section 3.

## C.1 Proof of the first part of Theorem 3.1

In this section we prove the first part of Theorem 3.1, which is restated by the following theorem.

**Theorem C.1.** *The double greedy algorithm of Buchinder et al. [7] guarantees $[1/(2-m)]$- approximation for unconstrained submodular maximization.*

Let $f \colon 2^{\mathcal{N}} \to \mathbb{R}_{\geq 0}$ be an arbitrary non-negative $m$-monotone submodular function over the ground set $\mathcal{N}$. To prove Theorem C.1, we need to show that given $f$, the double greedy algorithm of Buchinder et al. [7] outputs a (random) set $S$ obeying $\mathbb{E}[f(S)] \geq f(OPT)/(2-m)$, where $OPT$ is some subset of $\mathcal{N}$ maximizing $f$. Therefore, we start by looking at the approximation guarantees that are known for double greedy when ignoring the monotonicity ratio.

Buchinder et al. [7] proved that the output $S$ of double greedy always obeys

$$\mathbb{E}[f(S)] \geq \frac{2f(OPT) + f(\varnothing) + f(\mathcal{N})}{4} \ .$$

However, it turns out that this guarantee is only a special case of a more general guarantee that can be proved. Specifically, we prove below the following guarantee.

**Proposition C.2.** *The (random) output set $S$ of double greedy obeys*

$$\mathbb{E}[f(S)] \geq \frac{2r}{(r+1)^2} \cdot f(OPT) + \frac{1}{(r+1)^2} \cdot f(\varnothing) + \frac{r^2}{(r+1)^2} \cdot f(\mathcal{N})$$

*for every value $r > 0$, simultaneously (i.e., the algorithm need not know $r$).*

Proposition C.2 is based on ideas first used by Buchbinder et al. [6] in the context of an algorithm that is related to double greedy. Recently, Qi [47] observed that these ideas are useful also in the context of the double greedy algorithm, and used them to derive an improved result for a related problem termed "Regularized Unconstrained Submodular Maximization". Proposition C.2 is another consequence of the application of these ideas to double greedy.

Before getting to the proof of Proposition C.2, let us show that it implies Theorem C.1.

*Proof of Theorem C.1.* Since $f$ is $m$ monotone and $OPT \subseteq \mathcal{N}$, $f(N) \geq m \cdot f(OPT)$. Additionally, the non-negativity of $f$ guarantees $f(\varnothing) \geq 0$. Plugging both these observations into the guarantee of Proposition C.2, we get

$$\mathbb{E}[f(S)] \geq \frac{2r}{(r+1)^2} \cdot f(OPT) + \frac{1}{(r+1)^2} \cdot f(\varnothing) + \frac{r^2}{(r+1)^2} \cdot f(\mathcal{N})$$

$$\geq \frac{2r}{(r+1)^2} \cdot f(OPT) + \frac{r^2}{(r+1)^2} \cdot [m \cdot f(OPT)] = \frac{2r + r^2 m}{(r+1)^2} \cdot f(OPT) \ .$$

Since the above inequality holds for every $r > 0$, we can choose $r = 1/(1-m)$, and get

$$\mathbb{E}[f(S)] \geq \frac{2r + r^2 m}{(r+1)^2} \cdot f(OPT) = \frac{2/r + m}{(1 + 1/r)^2} \cdot f(OPT)$$

$$= \frac{2(1-m) + m}{(1 + (1-m))^2} \cdot f(OPT) = \frac{2-m}{(2-m)^2} \cdot f(OPT) = \frac{1}{2-m} \cdot f(OPT) \ . \qquad \square$$

The rest of this section is devoted to proving Proposition C.2. To prove this proposition, we first need to describe the double greedy algorithm that it analyzes, which appears as Algorithm 1. In a nutshell, this algorithm maintains two solutions $X$ and $Y$ that are originally the empty set and entire ground set, respectively. In every iteration, the algorithm considers a different element of the ground set, and either adds it to $X$, or removes it from $Y$. Once all the elements have been considered, the sets $X$ and $Y$ become identical, and they are the output of the algorithm. Note also that Algorithm 1 uses $n$ to denote the size of the ground set $\mathcal{N}$.

The heart of the analysis of Algorithm 1 is Lemma C.3, which was proved by Qi [47], and we prove here in more detail for completeness. To state Lemma C.3, we need to define for every integer $0 \leq i \leq n$ the set $OPT_i = (OPT \cup X_i) \cap Y_i$. Notice that $OPT_i$ agrees with $X_i$ and $Y_i$ on all the elements on which these two sets agree (i.e., elements that Algorithm 1 considered in its first $i$ iterations). On the remaining elements, $OPT_i$ agrees with $OPT$. Thus, as $i$ increases, $OPT_i$ evolves from being equal to $OPT$ to being equal to the output set $X_n = Y_n$ of Algorithm 1.

---

**Algorithm 1:** `Double-Greedy`

---

1  Denote the elements of $\mathcal{N}$ by $u_1, u_2, \ldots, u_n$ in an arbitrary order.
2  Let $X_0 \leftarrow \varnothing$ and $Y_0 \leftarrow \varnothing$.
3  **for** $i = 1$ **to** $n$ **do**
4  $\quad$ Let $a_i \leftarrow f(u_i \mid X_{i-1})$ and $b_i \leftarrow -f(u_i \mid Y_{i-1} - u_i)$.
5  $\quad$ **if** $b_i \leq 0$ **then** Let $X_i \leftarrow X_{i-1} + u_i$ and $Y_i \leftarrow Y_{i-1}$.
6  $\quad$ **else if** $a_i \leq 0$ **then** Let $X_i \leftarrow X_{i-1}$ and $Y_i \leftarrow Y_{i-1} - u_i$.
7  $\quad$ **else**
8  $\quad\quad$ **with** *probability* $\frac{a_i}{a_i+b_i}$ **do** Let $X_i \leftarrow X_{i-1} + u_i$ and $Y_i \leftarrow Y_{i-1}$.
9  $\quad\quad$ **otherwise** Let $X_i \leftarrow X_{i-1}$ and $Y_i \leftarrow Y_{i-1} - u_i$. // Occurs with prob. $\frac{b_i}{a_i+b_i}$.
10 $\quad$ **return** $X_n (= Y_n)$.

---

**Lemma C.3.** *For every integer $1 \leq i \leq n$ and $r > 0$,*

$$\mathbb{E}[f(OPT_{i-1}) - f(OPT_i)] \leq \frac{1}{2}\mathbb{E}[r^{-1}(f(X_i) - f(X_{i-1})) + r(f(Y_i) - f(Y_{i-1}))] .$$

*Proof.* By the law of total expectation, it suffices to prove the lemma conditioned on any particular choice for the random bits tossed in the first $i - 1$ iterations of Algorithm 1. Notice that once these random bits are fixed, $OPT_{i-1}$, $X_{i-1}$ and $Y_{i-1}$ become deterministic sets, and so do the numbers $a_i$ and $b_i$ calculated by Algorithm 1. We now need to consider three cases based on the values of these numbers.

The first case is the case of $b_i \leq 0$. In this case Algorithm 1 deterministically set $X_i \leftarrow X_{i-1} + u_i$ and $Y_i \leftarrow Y_{i-1}$, which reduces the inequality that we need to prove to

$$f(OPT_{i-1}) - f(OPT_{i-1} + u_i) \leq \frac{1}{2r}[f(X_{i-1} + u_i) - f(X_{i-1})] . \tag{2}$$

Observe that the description of Algorithm 1 implies $X_{i-1} \subseteq Y_{i-1}$, $u_i \notin X_{i-1}$ and $u_i \in Y_{i-1}$ (see Buchinder et al. [7] for a formal proof of these properties). Given these properties, the submodularity of $f$ shows that the right hand side of Inequality (2) is non-negative because

$$
\begin{aligned}
f(X_{i-1} + u_i) - f(X_{i-1}) &\geq [f((X_{i-1} + u_i) \cup (Y_{i-1} - u_i)) \\
&\quad + f((X_{i-1} + u_i) \cap (Y_{i-1} - u_i)) - f(Y_{i-1} - u_i)] - f(X_{i-1}) \\
&= f(Y_{i-1}) + f(X_{i-1}) - f(Y_{i-1} - u_i) - f(X_{i-1}) = -b_i \geq 0 .
\end{aligned}
$$

To complete the proof of the first case, it remains to show that the left hand side of Inequality (2) is non-positive. If $u_i \in OPT_{i-1}$, then this left hand side is trivially 0. Otherwise, since $OPT_{i-1} \subseteq Y_{i-1}$ by definition, the submodularity of $f$ shows that this left hand side is non-positive because

$$f(OPT_{i-1}) - f(OPT_{i-1} + u_i) \leq f(Y_{i-1} - u_i) - f(Y_{i-1}) = b_i \leq 0 .$$

The second case that we need to consider is the case of $b_i > 0$ and $a_i \leq 0$. However, since the analysis of this case is analogous to the analysis of the previous case, we omit it. Thus, we are left with the case in which both $a_i$ and $b_i$ are positive. The analysis of this case consists of two sub-cases depending on whether $u_i \in OPT$ or not. Since the proofs of these sub-cases are analogous to each other, we assume from this point on that $u_i \notin OPT$.

In the case we consider, Algorithm 1 sets $X_i \leftarrow X_{i-1} + u_i$ and $Y_i \leftarrow Y_{i-1}$ with probability $a_i/(a_i + b_i)$, and otherwise it sets $X_i \leftarrow X_{i-1}$ and $Y_i \leftarrow Y_{i-1} - u_i$. Thus, the law of total expectation implies

$$
\begin{aligned}
\mathbb{E}[f(OPT_{i-1}) - f(OPT_i)] &= \frac{a_i}{a_i + b_i} \cdot [f(OPT_{i-1}) - f(OPT_{i-1} + u_i)] \\
&\quad + \frac{b_i}{a_i + b_i} \cdot [f(OPT_{i-1}) - f(OPT_{i-1})] \\
&= \frac{a_i}{a_i + b_i} \cdot [f(OPT_{i-1}) - f(OPT_{i-1} + u_i)] ,
\end{aligned}
$$

and

$$\mathbb{E}[r^{-1}(f(X_i) - f(X_{i-1})) + r(f(Y_i) - f(Y_{i-1}))]$$

$$= \frac{a_i}{a_i + b_i} \cdot [r^{-1}(f(X_{i-1} + u_i) - f(X_{i-1})) + r(f(Y_{i-1}) - f(Y_{i-1}))]$$

$$+ \frac{b_i}{a_i + b_i} \cdot [r^{-1}(f(X_{i-1}) - f(X_{i-1})) + r(f(Y_{i-1} - u_i) - f(Y_{i-1}))]$$

$$= \frac{r^{-1} a_i^2}{a_i + b_i} + \frac{r b_i^2}{a_i + b_i} \quad .$$

Plugging both these equalities into the inequality that we need to prove, we get that this inequality is equivalent to

$$\frac{a_i}{a_i + b_i} \cdot [f(OPT_{i-1}) - f(OPT_{i-1} + u_i)] \le \frac{1}{2}\left[\frac{r^{-1} a_i^2}{a_i + b_i} + \frac{r b_i^2}{a_i + b_i}\right] \quad .$$

Furthermore, like in the first case, we have $f(OPT_{i-1}) - f(OPT_{i-1} + u_i) \le b_i$, and therefore, it suffices to prove the inequality

$$\frac{a_i b_i}{a_i + b_i} \le \frac{1}{2}\left[\frac{r^{-1} a_i^2}{a_i + b_i} + \frac{r b_i^2}{a_i + b_i}\right] \quad ,$$

which holds since

$$r^{-1} a_i^2 + r b_i^2 = (a_i/\sqrt{r} - b_i\sqrt{r})^2 + 2(a_i/\sqrt{r})(b_i\sqrt{r}) \ge 2a_i b_i \quad . \qquad \square$$

Using Lemma C.3, we can now prove Proposition C.2.

*Proof of Proposition C.2.* Throughout this proof, $r$ is an arbitrary positive number. Summing up the guarantees of Lemma C.3 for all integers $1 \le i \le n$ yields

$$\sum_{i=1}^{n} \mathbb{E}[f(OPT_{i-1}) - f(OPT_i)] \le \frac{1}{2}\mathbb{E}\left[r^{-1} \cdot \sum_{i=1}^{n}(f(X_i) - f(X_{i-1})) + r \cdot \sum_{i=1}^{n}(f(Y_i) - f(Y_{i-1}))\right].$$

Using the linearity of expectation, we can collapse the telescopic sums in the last inequality, and get

$$\mathbb{E}[f(OPT_0) - f(OPT_n)] \le \frac{1}{2}\mathbb{E}[r^{-1}(f(X_n) - f(X_0)) + r(f(Y_n) - f(Y_0))] \quad .$$

As explained above, $OPT_0 = OPT$ and $OPT_n = X_n = Y_n$. Additionally, $X_0$ and $Y_0$ are set by Algorithm 1 to $\varnothing$ and $\mathcal{N}$, respectively. Plugging all these equalities into the previous inequality reduces it to

$$\mathbb{E}[f(OPT) - f(X_n)] \le \frac{1}{2}\mathbb{E}[r^{-1}(f(X_n) - f(\varnothing)) + r(f(X_n) - f(\mathcal{N}))] \quad .$$

The proposition now follows by rearranging the above inequality, and observing that $X_n$ is the output set $S$ mentioned in the statement of the proposition. $\qquad \square$

## C.2   Proof of Theorem 3.2

In this section, we show how the proof of the symmetry gap technique due to Vondrák [52] can be adapted to prove Theorem 3.2. Let us begin the section by stating some definitions that are required in order to formally state Theorem 3.2.

**Definition C.4.** [Strong symmetry] Consider a non-negative submodular function $f$ and a collection $\mathcal{F} \subseteq 2^{\mathcal{N}}$ of feasible sets. The problem $\max\{f(S) \mid S \in \mathcal{F}\}$ is strongly symmetric with respect to a group of permutations $\mathcal{G}$ on $\mathcal{N}$, if (1) $f(S) = f(\sigma(S))$ for all $S \subseteq \mathcal{N}$ and $\sigma \in \mathcal{G}$, and (2) $S \in \mathcal{F} \iff S' \in \mathcal{F}$ whenever $\mathbb{E}_{\sigma \in \mathcal{G}}[\mathbf{1}_{\sigma(S)}] = \mathbb{E}_{\sigma \in \mathcal{G}}[\mathbf{1}_{\sigma(S')}]$, where $\mathbb{E}_{\sigma \in \mathcal{G}}$ represents the expectation over picking $\sigma$ uniformly at random out of $\mathcal{G}$.

**Definition C.5.** [Symmetry gap] Consider a non-negative submodular function $f$ and a collection $\mathcal{F} \subseteq 2^{\mathcal{N}}$ of feasible sets. Let $F(x)$ be the multilinear extension of $f$ and $P(\mathcal{F}) \subseteq [0,1]^{\mathcal{N}}$ be the convex hull of $\mathcal{F}$. Then, if the problem $\max\{f(S) \mid S \in \mathcal{F}\}$ is strongly symmetric with respect to a group $\mathcal{G}$ of permutation, then its symmetry is defined as

$$\frac{\max\{F(\bar{\mathbf{x}}) \mid \mathbf{x} \in P(\mathcal{F})\}}{\max\{F(\mathbf{x}) \mid \mathbf{x} \in P(\mathcal{F})\}} \; ,$$

where $\bar{\mathbf{x}} \triangleq \mathbb{E}_{\sigma \in \mathcal{G}}[\sigma(\mathbf{x})]$.

**Definition C.6.** [Refinement] Consider a set $\mathcal{F} \subseteq 2^{\mathcal{N}}$, and let $X$ be some set. We say that $\tilde{\mathcal{F}} \subseteq 2^{\mathcal{N} \times X}$ is a refinement of $\mathcal{F}$ if

$$\tilde{F} = \left\{ S \subseteq \mathcal{N} \times X \;\middle|\; \mathbf{x} \in P(\mathcal{F}), \text{ where } x_u = \tfrac{|S \cap (\{u\} \times X)|}{|X|} \text{ for every } u \in \mathcal{N} \right\} \; .$$

Using the above definitions, we can now formally state the theorem that we want to prove.

**Theorem 3.2.** *Consider a non-negative $m$-monotone submodular function $f$ and a collection $\mathcal{F} \subseteq 2^{\mathcal{N}}$ of feasible sets such that the problem $\max\{f(S) \mid S \in \mathcal{F}\}$ is strongly symmetric with respect to some group $\mathcal{G}$ of permutations over $\mathcal{N}$ and has a symmetry gap $\gamma$. Let $\mathcal{C}$ be the class of problems $\max\{\tilde{f}(S) \mid S \in \tilde{F}\}$ in which $\tilde{f}$ is a non-negative $m$-monotone submodular function, and $\tilde{F}$ is a refinement of $F$. Then, for every $\varepsilon > 0$, any (even randomized) $(1 + \varepsilon)\gamma$-approximation algorithm for the class $\mathcal{C}$ would require exponentially many value queries to $\tilde{f}$.*

The crux of the symmetry gap technique is two lemmata due to [52] that we restate below. Lemma C.7 shows that given a non-negative set function $f$, one can obtain from it two continuous versions: a continuous version $\hat{F}$ that resembles $f$ itself, and a continuous version $\hat{G}$ that resembles a symmetrized version of $f$. Distinguishing between $\hat{F}$ and $\hat{G}$ is difficult, however, this does not translate into an hardness for discrete problems since $\hat{F}$ and $\hat{G}$ are continuous. Therefore, Vondrák [52] proved also Lemma C.8, which shows how these continuous functions can be translated back into set functions with appropriate properties.

**Lemma C.7** (Lemma 3.2 of [52]). *Consider a function $f \colon 2^{\mathcal{N}} \to \mathbb{R}_{\geq 0}$ invariant under a group of permutations $\mathcal{G}$ on the ground set $\mathcal{N}$. Let $F(\mathbf{x})$ be the multilinear extension of $F$, define $\bar{x} = \mathbb{E}_{\sigma \in \mathcal{G}}[\mathbf{1}_{\sigma(\mathbf{x})}]$ and fix any $\varepsilon > 0$. Then, there is $\delta > 0$ and functions $\hat{F}, \hat{G} \colon [0,1]^{\mathcal{N}} \to \mathbb{R}_{\geq 0}$ (which are also symmetric with respect to $\mathcal{G}$), satisfying the following:*

1. *For all $\mathbf{x} \in [0,1]^{\mathcal{N}}$, $\hat{G}(\mathbf{x}) = \hat{F}(\bar{\mathbf{x}})$.*

2. *For all $\mathbf{x} \in [0,1]^{\mathcal{N}}$, $|\hat{F}(\mathbf{x}) - F(\mathbf{x})| \leq \varepsilon$.*

3. *Whenever $\|\mathbf{x} - \bar{\mathbf{x}}\|_2 \leq \delta$, $\hat{F}(\mathbf{x}) = \hat{G}(\mathbf{x})$ and the value depends only on $\bar{\mathbf{x}}$.*

4. *The first partial derivatives of $\hat{F}$ and $\hat{G}$ are absolutely continuous.*

5. *If $f$ is monotone, then, for every element $u \in \mathcal{N}$, $\frac{\partial \hat{F}}{\partial x_u} \geq 0$ and $\frac{\partial \hat{G}}{\partial x_u} \geq 0$ everywhere.*

6. *If $f$ is submodular then, for every two elements $u, v \in \mathcal{N}$, $\frac{\partial^2 \hat{F}}{\partial x_u \partial x_v} \leq 0$ and $\frac{\partial^2 \hat{G}}{\partial x_u \partial x_v} \leq 0$ almost everywhere.*

**Lemma C.8** (Lemma 3.1 of [52]). *Let $n$ be a positive integer, and let $F \colon [0,1]^{\mathcal{N}} \to \mathbb{R}$ and $X = [n]$. If we define $f \colon 2^{\mathcal{N} \times X} \to \mathbb{R}_{\geq 0}$ so that $f(S) = F(\mathbf{x})$, where $x_u = \frac{1}{n}|S \cap (\{u\} \times X)|$. Then,*

1. *if $\frac{\partial F}{\partial x_u} \geq 0$ everywhere for each element $u \in \mathcal{N}$, then $f$ is monotone,*

2. *and if the first partial derivatives of $F$ are absolutely continuous and $\frac{\partial^2 F}{\partial x_u \partial x_v} \leq 0$ almost everywhere for all elements $u, v \in \mathcal{N}$, then $f$ is submodular.*

One can note that the above lemmata have the property that if the function $f$ plugged into Lemma C.7 is monotone, then the discrete functions obtained by applying Lemma C.8 to the functions $\hat{F}$ and $\hat{G}$ are also monotone. This is the reason that the framework of [52] applies to monotone functions

(as well as general, not necessarily monotone, functions). Therefore, to get the proof of [52] to yield Theorem 3.2, it suffices to prove the following two modified versions of Lemmata C.7 and C.8. These modified versions preserve $m$-monotonicity for any $m \in [0, 1]$, rather than just standard monotonicity.

**Lemma C.9** (modified version of Lemma C.7). *Consider a function $f\colon 2^{\mathcal{N}} \to \mathbb{R}_{\geq 0}$ that is $m$-monotone and invariant under a group of permutations $\mathcal{G}$ on the ground set $\mathcal{N}$. Let $F(\mathbf{x})$ be the multilinear extension of $F$, define $\bar{x} = \mathbb{E}_{\sigma \in \mathcal{G}}[\mathbf{1}_{\sigma(\mathbf{x})}]$ and fix any $\varepsilon > 0$. Then, there is $\delta > 0$ and functions $\hat{F}, \hat{G}\colon [0, 1]^{\mathcal{N}} \to \mathbb{R}_{\geq 0}$ (which are also symmetric with respect to $\mathcal{G}$), satisfying the following:*

1. *For all $\mathbf{x} \in [0, 1]^{\mathcal{N}}$, $\hat{G}(\mathbf{x}) = \hat{F}(\bar{\mathbf{x}})$.*

2. *For all $\mathbf{x} \in [0, 1]^{\mathcal{N}}$, $|\hat{F}(\mathbf{x}) - F(\mathbf{x})| \leq \varepsilon$.*

3. *Whenever $\|\mathbf{x} - \bar{\mathbf{x}}\|_2 \leq \delta$, $\hat{F}(\mathbf{x}) = \hat{G}(\mathbf{x})$ and the value depends only on $\bar{\mathbf{x}}$.*

4. *For every two vectors $\mathbf{x}, \mathbf{y} \in [0, 1]^{\mathcal{N}}$ obeying $\mathbf{x} \leq \mathbf{y}$, $m \cdot F(\mathbf{x}) \leq F(\mathbf{y})$.*

5. *If $f$ is submodular then, for every two elements $u, v \in \mathcal{N}$, $\frac{\partial^2 \hat{F}}{\partial x_u \partial x_v} \leq 0$ and $\frac{\partial^2 \hat{G}}{\partial x_u \partial x_v} \leq 0$ almost everywhere.*

**Lemma C.10** (modified version of Lemma C.8). *Let $n$ be a positive integer, and let $F\colon [0, 1]^{\mathcal{N}} \to \mathbb{R}$ and $X = [n]$. If we define $f\colon 2^{\mathcal{N} \times X} \to \mathbb{R}_{\geq 0}$ so that $f(S) = F(\mathbf{x})$, where $x_u = \frac{1}{n}|S \cap (\{u\} \times X)|$. Then,*

1. *if for some value $m \in [0, 1]$ the inequality $m \cdot F(\mathbf{x}) \leq F(\mathbf{y})$ holds for any two vectors $\mathbf{x}, \mathbf{y} \in [0, 1]^{\mathcal{N}}$ that obey $\mathbf{x} \leq \mathbf{y}$, then $f$ is $m$-monotone,*

2. *and if the first partial derivatives of $F$ are absolutely continuous and $\frac{\partial^2 F}{\partial x_u \partial x_v} \leq 0$ almost everywhere for all elements $u, v \in \mathcal{N}$, then $f$ is submodular.*

The proof of Lemma C.9 is quite long and appears below. However, before getting to this proof, let first give the much simpler proof of Lemma C.10.

*Proof of Lemma C.10.* The second point in Lemma C.10 follows immediately from Lemma C.8, so we concentrate on proving the first point. In other words, we assume that $m \cdot F(\mathbf{x}) \leq F(\mathbf{y})$ for every two vectors $\mathbf{x}, \mathbf{y} \in [0, 1]^{\mathcal{N}}$ obeying $\mathbf{x} \leq \mathbf{y}$, and we need to show that $m \cdot f(S) \leq f(T)$ for every two sets $S \subseteq T \subseteq \mathcal{N}$.

Let us define two vectors $\mathbf{x}^{(S)}, \mathbf{x}^{(T)} \subseteq [0, 1]^{\mathcal{N}}$ as follows. For every $u \in \mathcal{N}$,

$$x_u^{(S)} = \frac{1}{n}|S \cap (\{u\} \times X)| \quad \text{and} \quad x_u^{(T)} = \frac{1}{n}|T \cap (\{u\} \times X)| \ .$$

Since $S \subseteq T$, we get $\mathbf{x}^{(S)} \leq \mathbf{x}^{(T)}$, which implies $m \cdot F(\mathbf{x}^{(S)}) \leq F(\mathbf{x}^{(T)})$; and the last inequality proves the lemma since $f(S) = F(\mathbf{x}^{(S)})$ and $f(T) = F(\mathbf{x}^{(T)})$ by the definition of $f$. $\qquad\square$

We now get to the proof of Lemma C.9. We use in this proof functions $\hat{F}$ and $\hat{G}$ that are similar to the ones constructed by Vondrák [52] in the proof of Lemma C.7. Specifically, like in the proof of [52], we define

$$\hat{G}(\mathbf{x}) = G(\mathbf{x}) + 256M|\mathcal{N}|\alpha J(\mathbf{x}) \ ,$$

where $M$ is the maximum value that the function $f$ takes on any set, $G$ is a symmetrized version of the multilinear extension $F$ of $f$ defined as $G(\mathbf{x}) = F(\bar{\mathbf{x}})$, $J(\mathbf{x}) \triangleq |\mathcal{N}|^2 + 3|\mathcal{N}| \cdot \sum_{u \in \mathcal{N}} x_u - \left(\sum_{u \in \mathcal{N}} x_u\right)^2$, and $\alpha$ is a positive value that is independent of $\mathbf{x}$. Similarly, the function $\hat{F}$ was defined by Vondrák [52] as

$$\hat{F}(\mathbf{x}) = \tilde{F}(\mathbf{x}) + 256M|\mathcal{N}|\alpha J(\mathbf{x}) \ ,$$

where the function $\tilde{F}$ interpolates between the multilinear extension $F$ of $f$ and its symmetrized version $G$, and is given by

$$\tilde{F}(\mathbf{x}) = (1 - \phi(D(\mathbf{x}))) \cdot F(\mathbf{x}) + \phi(D(\mathbf{x})) \cdot G(\mathbf{x}) \ .$$

Here, $D(\mathbf{x}) \triangleq \|\mathbf{x} - \bar{\mathbf{x}}\|_2^2$, and $\phi \colon \mathbb{R}_{\geq 0} \to [0, 1]$ is a function which is defined using the following lemma.

**Lemma C.11** (Lemma 3.7 of [52]). *For any $\alpha, \beta > 0$, there is $\delta > (0, \beta)$ and a function $\phi \colon \mathbb{R}_{\geq 0} \to [0, 1]$ with an absolutely continuous first derivative such that*

- *For $t \in [0, \delta]$, $\phi(t) = 1$.*

- *For $t \geq \beta$, $\phi(t) < e^{-1/\alpha}$.*

- *For all $t \geq 0$, $|t\phi'(t)| \leq 4\alpha$.*

- *For almost all $t \geq 0$, $|t^2\phi''(t)| \leq 10\alpha$.*

Vondrák [52] proved that the above functions $\hat{F}$ and $\hat{G}$ have all the properties guaranteed by Lemma C.7 for the $\delta$ whose existence is guaranteed by Lemma C.11 when the values of $\alpha$ and $\beta$ are set to be $\alpha = \frac{\varepsilon}{2000M|\mathcal{N}|^3}$ and $\beta = \frac{\varepsilon}{16M|\mathcal{N}|}$. Moreover, the proof of [52] continues to work as long as $\alpha \leq \frac{\varepsilon}{2000M|\mathcal{N}|^3}$ and $\beta \leq \frac{\varepsilon}{16M|\mathcal{N}|}$. Therefore, we assume below that $\alpha = \min\{1, \frac{\varepsilon}{2000M|\mathcal{N}|^3}\}$ and $\beta = \min\{\alpha^2, \frac{\varepsilon}{16M|\mathcal{N}|}\}$, and we prove only the part of Lemma C.9 that is not stated in the guarantees of Lemma C.7, which is Property 4 of the lemma. We begin by showing that the function $\hat{G}$ indeed has this property.

**Lemma C.12.** *For every two vectors $\mathbf{x}, \mathbf{y} \in [0, 1]^{\mathcal{N}}$ obeying $\mathbf{x} \leq \mathbf{y}$, $m \cdot \hat{G}(\mathbf{x}) \leq \hat{G}(\mathbf{y})$.*

*Proof.* Consider the random sets $\mathtt{R}(\bar{\mathbf{x}})$ and $\mathtt{R}(\bar{\mathbf{y}})$. Since $\bar{\mathbf{x}} \leq \bar{\mathbf{y}}$, the set $\mathtt{R}(\bar{\mathbf{y}})$ stochastically dominates $\mathtt{R}(\bar{\mathbf{x}})$. In other words, one can correlate the randomness of these sets in a way that does not alter their distributions, but guarantees that the inclusion $\mathtt{R}(\bar{\mathbf{x}}) \subseteq \mathtt{R}(\bar{\mathbf{y}})$ holds deterministically. Assuming this done, we get

$$m \cdot G(\mathbf{x}) = m \cdot F(\bar{\mathbf{x}}) = m \cdot \mathbb{E}[f(\mathtt{R}(\bar{\mathbf{x}}))] \leq \mathbb{E}[f(\mathtt{R}(\bar{\mathbf{y}}))] = F(\bar{\mathbf{y}}) = G(\mathbf{y}) \ , \tag{3}$$

where the inequality follows from the linearity of the expectation and the $m$-monotonicity of $f$.

Observe now that for every element $u \in \mathcal{N}$, the partial derivative of $J$ with respect to $z_u$ at any point $\mathbf{z} \in [0, 1]^{\mathcal{N}}$ is

$$\frac{\partial J(\mathbf{z})}{\partial z_u} = 3|\mathcal{N}| - 2\sum_{v \in \mathcal{N}} z_v \geq |\mathcal{N}| \geq 0 \ .$$

Hence, the inequality $\mathbf{x} \leq \mathbf{y}$ implies $m \cdot J(\mathbf{x}) \leq J(\mathbf{x}) \leq J(\mathbf{y})$. Together with Inequality (3), this implies the lemma. $\square$

One can observe that the arguments used to prove Inequality(3) in the proof of the last lemma also show that $m \cdot F(\mathbf{x}) \leq F(\mathbf{y})$, which is a fact that we use below. However, proving that $\hat{F}$ also has this property (and therefore, obeys Property 4 of Lemma C.9) is more involved. As a first step towards this goal, we bound the gradient of

$$\tilde{F}(\mathbf{x}) - F(\mathbf{x}) = \phi(D(\mathbf{x})) \cdot [G(\mathbf{x}) - F(\mathbf{x})] \ .$$

The following lemma does that in the regime in which $D(\mathbf{x})$ is small, and the next lemma handles the other regime.

**Lemma C.13.** *For every element $u \in \mathcal{N}$ and vector $\mathbf{x} \in [0, 1]^{\mathcal{N}}$ obeying $D(\mathbf{x}) \leq \beta$, the absolute value of the partial derivative $\frac{\partial\{\phi(D(\mathbf{x}))\cdot[G(\mathbf{x})-F(\mathbf{x})]\}}{\partial x_u}$ is at most $72\sqrt{\beta}M|\mathcal{N}| \leq 72\alpha M|\mathcal{N}|$.*

*Proof.* Observe that

$$\frac{\partial\{\phi(D(\mathbf{x})) \cdot [G(\mathbf{x}) - F(\mathbf{x})]\}}{\partial x_u} = \phi'(D(\mathbf{x})) \cdot \frac{\partial D(\mathbf{x})}{\partial x_u} \cdot [G(\mathbf{x}) - F(\mathbf{x})]$$

$$+ \phi(D(\mathbf{x})) \cdot \left[\frac{\partial G(\mathbf{x})}{\partial x_u} - \frac{\partial F(\mathbf{x})}{\partial x_u}\right] \ .$$

To use this equation to bound the absolute value of the left hand side, we need to make some observations. First, Lemma 3.6 of [52] shows that $\|\nabla D(\mathbf{x})\|_2 = 2\sqrt{D(\mathbf{x})}$, which implies

$$\frac{\partial D(\mathbf{x})}{\partial x_u} \le \|\nabla D(\mathbf{x})\|_2 = 2\sqrt{D(\mathbf{x})} \ .$$

Additionally, Lemma 3.5 of [52] shows that $|G(\mathbf{x}) - F(\mathbf{x})| \le 8M|\mathcal{N}| \cdot D(\mathbf{x})$, and therefore,

$$
\begin{aligned}
\left| \phi'(D(\mathbf{x})) \cdot \frac{\partial D(\mathbf{x})}{\partial x_u} \cdot [G(\mathbf{x}) - F(\mathbf{x})] \right| &\le |\phi'(D(\mathbf{x}))| \cdot \left| \frac{\partial D(\mathbf{x})}{\partial x_u} \right| \cdot |G(\mathbf{x}) - F(\mathbf{x})| \\
&\le |\phi'(D(\mathbf{x}))| \cdot 2\sqrt{D(\mathbf{x})} \cdot 8M|\mathcal{N}| \cdot D(\mathbf{x}) \\
&= |D(\mathbf{x}) \cdot \phi'(D(\mathbf{x}))| \cdot 16M|\mathcal{N}| \cdot \sqrt{D(\mathbf{x})} \\
&\le 64\alpha\sqrt{\beta}M|\mathcal{N}| \ ,
\end{aligned}
$$

where the second inequality follows from Lemma C.11 and our assumption that $D(\mathbf{x}) \le \beta$.

We now observe that

$$
\left| \phi(D(\mathbf{x})) \cdot \left[ \frac{\partial G(\mathbf{x})}{\partial x_u} - \frac{\partial F(\mathbf{x})}{\partial x_u} \right] \right| = \phi(D(\mathbf{x})) \cdot \left| \frac{\partial G(\mathbf{x})}{\partial x_u} - \frac{\partial F(\mathbf{x})}{\partial x_u} \right|
$$

$$
\le \phi(D(\mathbf{x})) \cdot \|\nabla G(\mathbf{x}) - \nabla F(\mathbf{x})\|_2 \le \phi(D(\mathbf{x})) \cdot 8M|\mathcal{N}| \cdot \sqrt{D(\mathbf{x})} \le 8\sqrt{\beta}M|\mathcal{N}| \ ,
$$

where the second inequality holds since Lemma 3.5 of [52] shows that $\|\nabla G(\mathbf{x}) - F(\mathbf{x})\|_2 \le 8M|\mathcal{N}| \cdot \sqrt{D(\mathbf{x})}$; and the last inequality holds by our assumption that $D(\mathbf{x}) \le \beta$ and by recalling that the range of $\phi$ is $[0, 1]$.

Combining all the above yields

$$
\left| \frac{\partial \{\phi(D(\mathbf{x})) \cdot [G(\mathbf{x}) - F(\mathbf{x})]\}}{\partial x_u} \right| \le 64\alpha\sqrt{\beta}M|\mathcal{N}| + 8\sqrt{\beta}M|\mathcal{N}| \le 72\sqrt{\beta}M|\mathcal{N}| \ ,
$$

where the second inequality holds since $\alpha \le 1$. $\qquad\square$

**Lemma C.14.** *For every element $u \in \mathcal{N}$ and vector $\mathbf{x} \in [0, 1]^{\mathcal{N}}$ obeying $D(\mathbf{x}) \ge \beta$, the absolute value of the partial derivative $\frac{\partial \{\phi(D(\mathbf{x})) \cdot [G(\mathbf{x}) - F(\mathbf{x})]\}}{\partial x_u}$ is at most $72\alpha M|\mathcal{N}|^{3/2}$.*

*Proof.* Repeating the proof of Lemma C.13, except for the use of the inequality $D(\mathbf{x}) \le \beta$ (which does not hold in the current lemma) and the inequality $\phi(\mathbf{x}) \le 1$ (which too weak for our current purpose), we get

$$
\left| \phi(D(\mathbf{x})) \cdot \left[ \frac{\partial G(\mathbf{x})}{\partial x_u} - \frac{\partial F(\mathbf{x})}{\partial x_u} \right] \right| \le 64\alpha M|\mathcal{N}| \cdot \sqrt{D(\mathbf{x})} + |\phi(D(\mathbf{x}))| \cdot 8M|\mathcal{N}| \cdot \sqrt{D(\mathbf{x})} \ .
$$

The expression $\phi(D(\mathbf{x}))$ can be upper bounded by $e^{-1/\alpha} \le \alpha$ by Lemma C.11. Also, $D(\mathbf{x}) = \|\mathbf{x} - \bar{\mathbf{x}}\|_2^2 \le |\mathcal{N}|$. The lemma now follows by plugging these two upper bounds into the previous inequality. $\qquad\square$

**Corollary C.15.** *For every element $u \in \mathcal{N}$ and vector $\mathbf{x} \in [0, 1]^{\mathcal{N}}$, the absolute value of the partial derivative $\frac{\partial \{\phi(D(\mathbf{x})) \cdot [G(\mathbf{x}) - F(\mathbf{x})]\}}{\partial x_u}$ is at most $72\alpha M|\mathcal{N}|^{3/2}$.*

The last corollary implies that $\tilde{F}$ can be presented as the sum of $F$ and a component that changes slowly. Therefore, if we add to $\tilde{F}$ a function that increases quickly enough (as is done to define $\hat{F}$), then we should get a function that can be represented as $F$ plus a monotone component. This is the intuition formalized in the proof of the next lemma.

**Lemma C.16.** *The function $\hat{F}(\mathbf{x}) - F(\mathbf{x})$ has non-negative partial derivatives for every $\mathbf{x} \in [0, 1]^{\mathcal{N}}$.*

*Proof.* By the definition of $\hat{F}(\mathbf{x})$,

$$\hat{F}(\mathbf{x}) - F(\mathbf{x}) = \tilde{F}(\mathbf{x}) - F(\mathbf{x}) + 256M|\mathcal{N}|\alpha J(\mathbf{x}) \ .$$

By Corollary C.15 and the observation that all the partial derivatives of $J(\mathbf{x})$ are at least $|\mathcal{N}|$ (see the proof of Lemma C.12), the last equality implies, for every element $u \in \mathcal{N}$,

$$\frac{\partial [\hat{F}(\mathbf{x}) - F(\mathbf{x})]}{\partial x_u} \ge -72\alpha M|\mathcal{N}|^{3/2} + 256\alpha M|\mathcal{N}|^2 \ge 0 \ . \qquad\square$$

We are now ready to show that $\hat{F}$ obeys Property 4 of Lemma C.9.

**Lemma C.17.** *For every two vectors* $\mathbf{x}, \mathbf{y} \in [0,1]^{\mathcal{N}}$ *obeying* $\mathbf{x} \leq \mathbf{y}$, $m \cdot \hat{F}(\mathbf{x}) \leq \hat{F}(\mathbf{y})$.

*Proof.* Note that $\overline{\mathbf{1}_\varnothing} = \mathbf{1}_\varnothing$, which implies that $G(\mathbf{1}_\varnothing) = F(\mathbf{1}_\varnothing)$, and therefore,

$$\tilde{F}(\mathbf{1}_\varnothing) - F(\mathbf{1}_\varnothing) = \phi(D(\mathbf{1}_\varnothing)) \cdot [G(\mathbf{1}_\varnothing) - F(\mathbf{1}_\varnothing)] = 0 \ .$$

Plugging this observation into the definition of $\hat{F}$ now gives

$$\hat{F}(\mathbf{1}_\varnothing) - F(\mathbf{1}_\varnothing) = \tilde{F}(\mathbf{1}_\varnothing) - F(\mathbf{1}_\varnothing) + 256 M |\mathcal{N}| \alpha J(\mathbf{1}_\varnothing) = 256 M |\mathcal{N}| \alpha J(\mathbf{1}_\varnothing) \ .$$

Since all the first partial derivatives of $\hat{F}(\mathbf{z}) - F(\mathbf{z})$ are non-negative by Lemma C.16, the last inequality implies

$$\hat{F}(\mathbf{y}) - F(\mathbf{y}) \geq \hat{F}(\mathbf{x}) - F(\mathbf{x}) \geq 256 M |\mathcal{N}| \alpha J(\mathbf{x}) \geq 0 \ .$$

Hence,

$$m \cdot \hat{F}(\mathbf{x}) \leq m \cdot [F(\mathbf{x}) + \hat{F}(\mathbf{y}) - F(\mathbf{y})] \leq F(\mathbf{y}) + m \cdot [\hat{F}(\mathbf{y}) - F(\mathbf{y})] \leq F(\mathbf{y}) + [\hat{F}(\mathbf{y}) - F(\mathbf{y})] = \hat{F}(\mathbf{y}) \ ,$$

where the second inequality holds by the discussion immediately after the proof of Lemma C.12, and the last inequality holds since $m \leq 1$ and $\hat{F}(\mathbf{y}) - F(\mathbf{y}) \geq 0$. $\qquad \square$

### C.3 Proof of Lemma 3.3

**Lemma 3.3.** *The problem* $\max\{f(S) \mid S \in \mathcal{F}\}$ *has a symmetry gap of* $\frac{1}{2-m}$.

*Proof.* Observe that our definition of $\mathcal{F}$ implies that $P(\mathcal{F}) = [0,1]^{\mathcal{N}}$. Therefore,

$$\max\{F(\mathbf{x}) \mid \mathbf{x} \in P(\mathcal{F})\} = \max\{F(\mathbf{x}) \mid \mathbf{x} \in [0,1]^{\mathcal{N}}\} = \max\{f(S) \mid S \subseteq \mathcal{N}\} = 1 \ , \quad (4)$$

where the second equality holds since, for every vector $\mathbf{x}$, $F(\mathbf{x})$ is a convex combination of values of $f$ for subsets of $\mathcal{N}$; and on the other hand, for every set $S \subseteq \mathcal{N}$, $f(S) = F(\mathbf{1}_S)$.

Observe now that the definition of $f$ implies that

$$F(\mathbf{x}) = m[1 - (1 - x_u)(1 - x_v)] + (1 - m) \cdot [x_u(1 - x_v) + x_v(1 - x_u)]$$
$$= x_u + x_v - x_u x_v (2 - m) \ .$$

Since $\bar{\mathbf{x}}$ is a vector that has the value $(x_u + x_v)/2$ in both its coordinates, if we we use the shorthand $y = (x_u + x_v)/2$, then we get

$$F(\bar{\mathbf{x}}) = 2y - (2 - m)y^2 \ .$$

This expression is maximized for $y = 1/(2 - m)$, and the maximum attained for this $y$ is

$$\frac{2}{2 - m} - \frac{(2 - m)}{(2 - m)^2} = \frac{1}{2 - m} \ .$$

Since the value $y = 1/(2 - m)$ is obtained, for example, when $\mathbf{x} = (y, y) \in [0,1]^{\mathcal{N}}$, the above implies

$$\max\{F(\bar{\mathbf{x}}) \mid \mathbf{x} \in P(\mathcal{F})\} = \frac{1}{2 - m} \ .$$

Together with Equation (4), this implies the lemma. $\qquad \square$

## D  Inapproximability and Proofs of Section 4

In this section we state and analyze the algorithms used to prove the results given in Section 4. We also state and prove in Section D.3 the inapproximability result mentioned in Section 4.

## D.1 Analysis of the Greedy Algorithm

In this section we prove Theorem 4.1, which we repeat here for convenience.

**Theorem 4.1.** *The Greedy algorithm (Algorithm 2) has an approximation ratio of at least $m(1 - 1/e)$ for the problem of maximizing a non-negative $m$-monotone submodular function subject to a cardinality constraint.*

The greedy algorithm starts with an empty solution, and then augments this solution in $k$ iterations (recall that $k$ is the maximum cardinality allowed for a feasible solution). Specifically, in iteration $i$, the algorithm adds to the current solution the element $u_i$ with the best (largest) marginal contribution with respect to the current solution—but only if this addition does not decrease the value of the solution. A formal description of the greedy algorithm appears as Algorithm 2. Note that in this description the solution of the algorithm after $i$ iterations, for every integer $0 \leq i \leq n$, is denoted by $A_i$.

---

**Algorithm 2:** The Greedy Algorithm $(f, k)$

---
1 Let $A_0 \leftarrow \varnothing$.
2 **for** $i = 1$ **to** $k$ **do**
3     Let $u_i$ be the element of $\mathcal{N} \setminus A_{i-1}$ maximizing $f(u_i \mid A_{i-1})$.
4     **if** $f(u_i \mid A_{i-1}) \geq 0$ **then** Let $A_i \leftarrow A_{i-1} + u_i$.
5     **else** Let $A_i \leftarrow A_{i-1}$.
6 **return** $A_k$.

---

Our first step towards proving Theorem 4.1 is the following lemma, which lower bounds the increase in the value of $f(A_i)$ as a function of $i$. Specifically, the lemma shows that this increase is significant as long as there is a significant gap between between $f(A_{i-1})$ and $m \cdot f(OPT)$, where $OPT$ is an arbitrary optimal solution.

**Lemma D.1.** *For every integer $1 \leq i \leq k$, $f(A_i) - f(A_{i-1}) \geq k^{-1}[m \cdot f(OPT) - f(A_{i-1})]$.*

*Proof.* We need to distinguish between two cases. Consider first the case in which $f(u_i \mid A_{i-1}) \geq 0$. In this case,

$$
\begin{aligned}
f(A_i) - f(A_{i-1}) = f(u_i \mid A_{i-1}) &\geq \frac{|OPT \setminus A_{i-1}|}{k} \cdot f(u_i \mid A_{i-1}) \\
&\geq \frac{|OPT \setminus A_{i-1}|}{k} \cdot \max_{u \in OPT \setminus A_{i-1}} f(u \mid A_{i-1}) \geq \frac{\sum_{u \in OPT \setminus A_{i-1}} f(u \mid A_{i-1})}{k} \\
&\geq \frac{f(OPT \cup A_{i-1}) - f(A_{i-1})}{k} \geq \frac{m \cdot f(OPT) - f(A_{i-1})}{k} \quad,
\end{aligned}
$$

where the first inequality holds since $|OPT \setminus A_{i-1}| \leq |OPT| \leq k$ because $OPT$ is a feasible solution, the second inequality is due to the way used by the greedy algorithm to choose the element $u_i$, the penulatimate inequality follows from the submodularity of $f$, and the last inequality holds since $f$ is $m$-monotone.

Consider now the case in which $f(u_i \mid A_{i-1}) < 0$. In this case, $f(A_i) - f(A_{i-1}) = 0$ because $A_i = A_{i-1}$. Furthermore, repeating the arguments used to prove the above inequality yields

$$
\begin{aligned}
m \cdot f(OPT) - f(A_{i-1}) &\leq |OPT \setminus A_{i-1}| \cdot \max_{u \in OPT \setminus A_{i-1}} f(u \mid A_{i-1}) \\
&\leq |OPT \setminus A_{i-1}| \cdot \max_{u \in \mathcal{N} \setminus A_{i-1}} f(u \mid A_{i-1}) \leq 0 \quad. \qquad \square
\end{aligned}
$$

Rearranging the last lemma, we get the following inequality.

$$
m \cdot f(OPT) - f(A_i) \leq (1 - 1/k) \cdot [m \cdot f(OPT) - f(A_{i-1})] \quad. \tag{5}
$$

This inequality bounds the rate in which the gap between $m \cdot f(OPT)$ reduces as a function of $i$. This allows us to prove Theorem 4.1.

*Proof of Theorem 4.1.* Combining Inequality (5) for every integer $1 \le i \le k$ yields

$$m \cdot f(OPT) - f(A_k) \le (1 - 1/k)^k \cdot [m \cdot f(OPT) - f(A_0)] \ .$$

Rearranging this inequality, we get

$$f(A_k) \ge m \cdot f(OPT) - m \cdot (1 - 1/k)^k \cdot [f(OPT) - f(A_0)] \ge m \cdot \left(1 - \frac{1}{e}\right) \cdot f(OPT) \ ,$$

where the last inequality follows from the non-negativity of $f$ and the inequality $(1 - 1/k)^k \le \frac{1}{e}$. $\quad\square$

## D.2   Analysis of Random Greedy

In this section we prove Theorem 4.2, which we repeat here for convenience.

**Theorem 4.2.** *Random Greedy (Algorithm 3) has an approximation ratio of at least $m(1 - 1/e) + (1 - m) \cdot (1/e)$ for the problem of maximizing a non-negative $m$-monotone submodular function subject to a cardinality constraint.*

Like the standard greedy algorithm from Section D.1, the Random Greedy algorithm starts with an empty solution, and then augments it in $k$ iterations. Specifically, in iteration $i$ the algorithm finds a set $M_i$ of at most $k$ elements whose total marginal contribution with respect to the current solution is maximal. Then, at most one element of $M_i$ is added to the algorithm's current solution in a random way guaranteeing that every element of $M_i$ is added to the solution with probability exactly $1/k$. A formal presentation of the Random Greedy algorithm appears as Algorithm 3. Note that in this presentation the solution of the algorithm after $i$ iterations is denoted by $A_i$.

---

**Algorithm 3:** Random Greedy $(f, k)$

1 Let $A_0 \leftarrow \varnothing$.
2 **for** $i = 1$ **to** $k$ **do**
3 $\quad$ Let $M_i \leftarrow \arg\max_{B \subseteq \mathcal{N} \setminus A_{i-1}, |B| \le k} \{\sum_{u \in B} f(u \mid A_{i-1})\}$.
4 $\quad$ **with** *probability* $(1 - |M_i|/k)$ **do**
5 $\quad\quad \lfloor\ A_i \leftarrow A_{i-1}$.
6 $\quad$ **otherwise**
7 $\quad\quad$ Let $u_i$ be a uniformly random element of $M_i$.
8 $\quad\quad$ Set $A_i \leftarrow A_{i-1} + u_i$.

9 **return** $A_k$.

---

We start the analysis of the Random Greedy algorithm with the following lemma.

**Lemma D.2.** *For every integer $0 \le i \le k$ and element $u \in \mathcal{N}$, $\Pr[u \in A_i] \le 1 - (1 - 1/k)^i$.*

*Proof.* Note that in each iteration $i$ of Algorithm 3, any element $u \in \mathcal{N} \setminus A_{i-1}$ is added to the current solution with probability of at most $1/k$. Hence,

$$\Pr[u \in A_i] = 1 - \Pr[u \notin A_i] = 1 - \prod_{j=1}^{i} \Pr[u \notin A_j \mid u \notin A_{j-1}] \le 1 - (1 - 1/k)^i \ . \quad\square$$

Plugging the guarantee of the last lemma into Lemma 2.1 yields the following lower bound on the expected value of $A_i \cup OPT$.

**Corollary D.3.** *For every integer $0 \le i \le k$, $\mathbb{E}[f(A_i \cup OPT)] \ge [1 - (1 - m) \cdot (1 - (1 - \frac{1}{k})^i)] \cdot f(OPT) = m \cdot f(OPT) + (1 - m)(1 - \frac{1}{k})^i \cdot f(OPT)$.*

Using the last corollary we are now ready to prove Theorem 4.2.

*Proof of Theorem 4.2.* Let $\mathcal{E}_{i-1}$ be an arbitrary possible choice for the random decisions of Random Greedy during its first $i-1$ iterations. Observe that, conditioned on $\mathcal{E}_{i-1}$ happening,

$$\mathbb{E}[f(A_i) - f(A_{i-1})] = \frac{\sum_{u \in M_i} f(u \mid A_{i-1})}{k}$$

$$\geq \frac{\sum_{u \in OPT \setminus A_{i-1}} f(u \mid A_{i-1})}{k} \geq \frac{f(A_{i-1} \cup OPT) - f(A_{i-1})}{k} \ ,$$

where the first inequality follows from the choice of $M_i$ by the algorithm, and the second inequality follows from submodularity. Taking now expectation over the choice $\mathcal{E}_{i-1}$ that realized, the last inequality yields

$$\mathbb{E}[f(A_i) - f(A_{i-1})] \geq \frac{\mathbb{E}[f(A_{i-1} \cup OPT)] - \mathbb{E}[f(A_{i-1})]}{k} \tag{6}$$

$$\geq \frac{m \cdot f(OPT) + (1-m)(1-\frac{1}{k})^{i-1} \cdot f(OPT) - \mathbb{E}[f(A_{i-1})]}{k} \ ,$$

where the second inequality is due to Corollary D.3.

The last inequality lower bounds the expected increase in the value of the solution of Random Greedy in every iteration. This implies also a lower bound on the expected value of $f(A_i)$. To complete the proof of the theorem, we need to prove a closed form for this implied lower bound, which we do by induction. Specifically, let us prove by induction on $i$ that

$$\mathbb{E}[f(A_i)] \geq \left[ m \cdot \left(1 - \left(1 - \frac{1}{k}\right)^i\right) + (1-m) \cdot \frac{i}{k} \cdot \left(1 - \frac{1}{k}\right)^{i-1} \right] \cdot f(OPT) \tag{7}$$

for every integer $0 \leq i \leq k$, which implies the theorem by plugging $i = k$ because $(1 - 1/k)^k \leq 1/e \leq (1 - 1/k)^{k-1}$.

For $i = 0$, Inequality (7) holds since the non-negativity of $f$ guarantees that $f(A_0) \geq 0 = [(1-m) \cdot (\frac{0}{k}) \cdot (1 - \frac{1}{k})^{-1} + m \cdot (1 - (1 - \frac{1}{k})^0)] \cdot f(OPT)$. Consider now some integer $0 < i \leq k$, and let us prove Inequality (7) for this value of $i$ assuming that its holds for $i - 1$. By Inequality (6),

$$\mathbb{E}[f(A_i)] = \mathbb{E}[f(A_{i-1})] + \mathbb{E}[f(A_i) - f(A_{i-1})]$$

$$\geq \mathbb{E}[f(A_{i-1})] + \frac{m \cdot f(OPT) + (1-m)(1-\frac{1}{k})^{i-1} \cdot f(OPT) - \mathbb{E}[f(A_{i-1})]}{k}$$

$$= \left(1 - \frac{1}{k}\right) \cdot \mathbb{E}[f(A_{i-1})] + \frac{m + (1-m)(1-\frac{1}{k})^{i-1}}{k} \cdot f(OPT) \ .$$

Plugging the induction hypothesis into the last inequality, we get

$$\mathbb{E}[f(A_i)] \geq \left(1 - \frac{1}{k}\right) \cdot \left[ m \cdot \left(1 - \left(1 - \frac{1}{k}\right)^{i-1}\right) + (1-m) \cdot \frac{i-1}{k} \cdot \left(1 - \frac{1}{k}\right)^{i-2} \right] \cdot f(OPT)$$

$$+ \frac{m + (1-m)(1-\frac{1}{k})^{i-1}}{k} \cdot f(OPT)$$

$$= \left[ m \left(1 - \left(1 - \frac{1}{k}\right)^i\right) + (1-m) \cdot \frac{i}{k} \cdot \left(1 - \frac{1}{k}\right)^{i-1} \right] \cdot f(OPT) \ . \qquad \square$$

## D.3 Inapproximability for a Cardinality Constraint

In this section we state and prove the inapproximability result stated in Section 4.

**Theorem D.4.** *For any constant $\varepsilon > 0$, no polynomial time algorithm can obtain an approximation ratio of*

$$\min_{\alpha \in [0,1]} \frac{\max_{x \in [0,1]} \{\alpha(mx^2 + 2x - 2x^2) + 2(1-\alpha)(1 - e^{x-1})(1 - (1-m)x)\}}{\max\{1, 2(1-\alpha)\}} + \varepsilon$$

*for the problem of maximizing a non-negative $m$-monotone submodular function subject to a cardinality constraint.*

We prove Theorem D.4 using the symmetry gap technique, and specifically, via our extension of this technique proved in Theorem 3.2. To use this theorem, we need to construct an instance of our problem in which there is a large gap between the values of the best (general) solution and the best symmetric solution. Our instance is based on an instance constructed by Oveis Gharan and Vondrák [46]. However, the objective function in the original instance of [46] is not $m$-monotone for any $m > 0$, and therefore, we need to modify it so that it becomes $m$-monotone for a value $m \in [0, 1]$ of our choosing.

Fix some positive integer value $r$ to be determined later and some value $\alpha \in [0, 1]$. The ground set of the instance we construct is $\mathcal{N} = \{a, b\} \cup \{a_i, b_i \mid i \in [r]\}$, and the constraint of the instance is a cardinality constraint allowing a feasible solution to include up to 2 elements. The objective function of our instance is the function $f \colon 2^{\mathcal{N}} \to \mathbb{R}_{\geq 0}$ defined by $f(S) = \alpha \cdot f_1(S) + (1 - \alpha)[f_2(S) + f_3(S)]$, where

$$f_1(S) = m \cdot \mathbf{1}[S \cap \{a, b\} \neq \varnothing] + (1 - m) \cdot (|S \cap \{a, b\}| \bmod 2) \ ,$$
$$f_2(S) = \mathbf{1}[S \cap \{a_i \mid i \in [r]\} \neq \varnothing] \cdot (1 - (1 - m) \cdot \mathbf{1}[a \in S])$$

and

$$f_3(S) = \mathbf{1}[S \cap \{b_i \mid i \in [r]\} \neq \varnothing] \cdot (1 - (1 - m) \cdot \mathbf{1}[b \in S]) \ .$$

Let us denote the above described instance of submodular maximization subject to a cardinality constraint by $\mathcal{I}$. We begin the analysis of $\mathcal{I}$ by proving some properties of its objective function.

**Lemma D.5.** *The objective function $f$ of $\mathcal{I}$ is non-negative, $m$-monotone and submodular.*

*Proof.* We prove below that the functions $f_1$, $f_2$ and $f_3$ have the properties stated in the lemma. This implies that $f$ also has these properties by Observation 2.2 and the well-known closure of the class of submodular functions to multiplication by a non-negative constant and addition (see, e.g., Lemma 1.2 of [4]). The function $f_1$ is identical to the function proved in Section 3 to have the properties stated in the lemma, and the functions $f_2$ and $f_3$ are identical to each other up to switching the roles of $a$ with $b$ and $a_i$ with $b_i$. Therefore, to prove that both $f_2$ and $f_3$ have the properties stated by the lemma it suffices to show that $f_2$ has these properties, which we do in the rest of this proof.

Clearly, $f_2$ is non-negative. To see that $f_2$ is a submodular function, note that

- For every set $S \subseteq \mathcal{N} - a$, $f_2(a \mid S) = -\mathbf{1}[S \cap \{a_i \mid i \in [r]\} \neq \varnothing] \cdot (1 - m)$.

- For every integer $1 \leq i \leq r$ and set $S \subseteq \mathcal{N} - a_i$, $f_2(a_i \mid S) = \mathbf{1}[S \cap \{a_i \mid i \in [r]\} = \varnothing] \cdot (1 - (1 - m) \cdot \mathbf{1}[a \in S])$.

- For every element $u \in (\mathcal{N} - a) \setminus \{a_i \mid i \in [r]\}$ and set $S \subseteq \mathcal{N} - u$, $f_2(u \mid S) = 0$.

Since all the above marginal contributions are down-monotone functions of $S$ (i.e., functions whose value can only decrease when elements are added to $S$), the function $f_2$ is submodular.

It remains to argue why $f_2$ is $m$-monotone. Consider any two sets $S \subseteq T \subseteq \mathcal{N}$. If $f_2(S) = 0$, then the inequality $m \cdot f(S) \leq f(T)$ follows from the non-negativity of $f_2$. Therefore, consider the case in which $f_2(S) > 0$, which implies that $S \cap \{a_i \mid i \in [r]\} \neq \varnothing$; and therefore, $f_2(S) = (1 - (1 - m) \cdot \mathbf{1}[a \in S]) \leq 1$. Since $S$ is a subset of $T$, we also get $f_2(T) = (1 - (1 - m) \cdot \mathbf{1}[a \in T]) \geq m$, and hence, $m \cdot f_2(S) \leq m \cdot 1 = m \leq f_2(T)$. $\qquad\square$

A cardinality constraint is symmetric in the sense that the feasibility of a set depends only on the number of elements in it, and is completely independent of the identity of these elements. Let us now denote by $\mathcal{G}$ the group of permutations of $\mathcal{N}$ that are equivalent to applying any number of the following two steps: (1) switching $a$ with $b$ and $a_i$ with $b_i$ for every $i \in [r]$, or (2) switching $a_i$ with $a_j$ for two integers $i, j \in [r]$. The first step preserves the value of $f$ because it simply switches the values of $f_2$ and $f_3$, while leaving the value of $f_1$ unaffected; and the second step preserves the value of $f$ since it deals with elements that both $f_1$ and $f_3$ ignore, and $f_2$ treats in the same way. Hence, for every set $S \subseteq \mathcal{N}$ and permutation $\sigma \in \mathcal{G}$, we have $f(S) = f(\sigma(S))$, which implies the following observation.

**Observation D.6.** *The instance $\mathcal{I}$ is strongly symmetric with respect to $\mathcal{G}$.*

To use Theorem 3.2, we still need to bound the symmetry gap of $\mathcal{I}$, which we do next.

**Lemma D.7.** *The symmetry gap of $\mathcal{I}$ is at most*

$$\frac{\max_{x \in [0,1]}\{\alpha(mx^2 + 2x - 2x^2) + 2(1-\alpha)[1 - (1 - (1-x)/r)^r](1 - (1-m)x)\}}{\max\{1, 2(1-\alpha)\}}$$

$$\leq \frac{\max_{x \in [0,1]}\{\alpha(mx^2 + 2x - 2x^2) + 2(1-\alpha)(1 - e^{x-1})(1 - (1-m)x)\}}{\max\{1, 2(1-\alpha)\}} + 2/r \enspace .$$

*Proof.* Two possible feasible solutions for $\mathcal{I}$ are the sets $\{a, b_1\}$ and $\{a_1, b_1\}$ whose values according to $f$ are 1 and $2(1-\alpha)$, respectively. Therefore, the value of the optimal solution for $\mathcal{I}$ is at least $\max\{1, 2(1-\alpha)\}$. Since the symmetry gap is the ratio between the value of the best symmetric solution and the value of the best solution, to prove the lemma it remains to argue that the best symmetric solution for $\mathcal{I}$ has a value of $\max_{x \in [0,1]}\{\alpha(mx^2 + 2x - 2x^2) + 2(1-\alpha)[1 - (1 - (1-x)/r)^r](1 - (1-m)x)\}$.

We remind the reader that a symmetric solution for $\mathcal{I}$ is $\bar{\mathbf{y}} = \mathbb{E}_{\sigma \in \mathcal{G}}[\mathbf{y}]$ for some vector $\mathbf{y} \in [0,1]^{\mathcal{N}}$ obeying $\|\mathbf{y}\|_1 \leq 2$. Since $a$ and $b$ can be exchanged with each other by the permutations of $\mathcal{G}$, the values of the coordinates of $a$ and $b$ in $\bar{\mathbf{y}}$ must be equal to each other. Similarly, every two elements of $\{a_i, b_i \in i \in [r]\}$ can be exchanged by the permutations of $\mathcal{G}$, and therefore, the values of the coordinates of these elements in $\bar{\mathbf{y}}$ must all be identical. Thus, any symmetric solution $\bar{\mathbf{y}}$ can be represented as

$$\bar{y}_u = \begin{cases} x & \text{if } u = a \text{ or } u = b \enspace , \\ z & \text{if } u \in \{a_i, b_i \mid i \in [r]\} \end{cases}$$

for some values $x, z \in [0,1]$ obeying $2x + 2rz \leq 2$ (or equivalently, $z \leq (1-x)/r$). The value of this solution (according to the multilinear extension $F$ of $f$) is

$$\alpha[m(1 - (1-x)^2) + 2(1-m)x(1-x)] + 2(1-\alpha)(1 - (1-z)^r)(1 - (1-m)x)$$

$$= \alpha(mx^2 + 2x - 2x^2) + 2(1-\alpha)(1 - (1-z)^r)(1 - (1-m)x) \enspace .$$

Since this expression is a non-decreasing function of $z$, the maximum value of any symmetry solution for $\mathcal{I}$ is

$$\max_{\substack{x,z \in [0,1] \\ z \leq (1-x)/r}} \{\alpha(mx^2 + 2x - 2x^2) + 2(1-\alpha)(1 - (1-z)^r)(1 - (1-m)x)\}$$

$$= \max_{x \in [0,1]} \{\alpha(mx^2 + 2x - 2x^2) + 2(1-\alpha)[1 - (1 - (1-x)/r)^r](1 - (1-m)x)\} \enspace . \qquad \square$$

Since any refinement of a cardinality constraint is a cardinality constraint over a larger ground set, plugging Lemma D.5, Observation D.6 and Lemma D.7 into Theorem 3.2 yields the following corollary.

**Corollary D.8.** *For every constant $\varepsilon' > 0$, no polynomial time algorithm for maximizing a non-negative $m$-monotone submodular function subject to a cardinality contraint obtains an approximation ratio of*

$$\frac{\max_{x \in [0,1]}\{\alpha(mx^2 + 2x - 2x^2) + 2(1-\alpha)(1 - e^{x-1})(1 - (1-m)x)\}}{\max\{1, 2(1-\alpha)\}} + 2/r + \varepsilon' \enspace .$$

Theorem D.4 now follows from the last corollary by choosing $\varepsilon' = \varepsilon/2$, $r = \lceil 4/\varepsilon \rceil$ and

$$\alpha = \arg\min_{\alpha' \in [0,1]} \frac{\max_{x \in [0,1]}\{\alpha'(mx^2 + 2x - 2x^2) + 2(1-\alpha')(1 - e^{x-1})(1 - (1-m)x)\}}{\max\{1, 2(1-\alpha')\}} \enspace .$$

# E  Inapproximability and Proofs of Section 5

In this section we state and analyze the algorithms used to prove the results given in Section 5. We also state and prove in Section E.4 the inapproximability result mentioned in Section 5.

### E.1 Analysis of the Greedy algorithm

A version of the greedy algorithm designed for matroid constraints appears as Algorithm 4. This algorithm starts with an empty solution, and then iteratively adds elements to this solution, where the element added in each iteration is the element with the largest marginal contrition with respect to the current solution among all the elements whose addition to the solution does not violate feasibility. The algorithm terminates when no additional elements can be added to the solution without decreasing its value.

---

**Algorithm 4:** The Greedy Algorithm (for a Matroid Constraint) $(f, \mathcal{M} = (\mathcal{N}, \mathcal{I}))$

---
**1** Let $A_0 \leftarrow \varnothing$, and $i \leftarrow 0$.
**2** **while** *true* **do**
**3** $\quad$ Let $u_{i+1}$ be the element of $\{v \in \mathcal{N} \setminus A_i \mid A_i + v \in \mathcal{I}\}$ maximizing $f(u_{i+1} \mid A_i)$.
**4** $\quad$ **if** $f(u_{i+1} \mid A_i) \geq 0$ **then** Let $A_{i+1} \leftarrow A_i + u_{i+1}$, and then, increase $i$ by 1.
**5** $\quad$ **else return** $A_i$.

---

**Theorem 5.1.** *The Greedy algorithm (Algorithm 4) has an approximation ratio of at least $m/2$ for maximizing a non-negative $m$-monotone submodular function subject to a matroid constraint.*

*Proof.* Lemma 3.2 of [27] shows that the greedy algorithm outputs a solution $S$ of value at least $f(S \cup OPT)/2$ for the problem of maximizing a non-negative submodular function $f$ subject to a matroid constraint, where $OPT$ is an optimal solution for the problem.[6] The theorem now follows since for an $m$-monotone function $f$ we are guaranteed to have $f(S \cup OPT) \geq m \cdot f(OPT)$. $\quad\square$

### E.2 Analysis of Measured Continuous Greedy

In this section, we reanalyze the Measured Continuous Greedy algorithm of [22] in view of the monotonicity ratio. Given a non-negative submodular function $f\colon 2^{\mathcal{N}} \to \mathbb{R}_{\geq 0}$ and a down-closed solvable[7] convex body $P \subseteq [0,1]^{\mathcal{N}}$, Measured Continuous Greedy is an algorithm designed to find a vector $\mathbf{x} \in P$ that approximately maximize $F(\mathbf{x})$, where $F$ is the multilinear extension of $f$. Specifically, we prove the following theorem.

**Theorem E.1.** *Given a non-negative $m$-monotone submodular function $f\colon 2^{\mathcal{N}} \to \mathbb{R}_{\geq 0}$, a solvable down-close convex body $P \subseteq [0,1]^{\mathcal{N}}$ and a parameter $T \geq 0$, Measured Continuous Greedy outputs a vector $\mathbf{x} \in [0,1]^{\mathcal{N}}$ obeying $F(\mathbf{x}) \geq [m(1 - e^{-T}) + (1 - m)Te^{-T}] \cdot f(OPT)$, where $F$ is the multilinear extension of $f$ and $OPT$ is the set maximizing $f$ among all sets whose characteristic vectors belong to $P$. Furthermore, $\mathbf{x} \in P$ whenever $T \in [0,1]$.*

We note that Feldman et al. [22] discussed conditions that guarantee that $\mathbf{x}$ belongs to $P$ also for some values of $T$ that are larger than 1. However, the above stated form of Theorem E.1 already suffices to prove Theorem 5.2. Let us explain why this is the case. When $P$ is the matroid polytope $P_{\mathcal{M}}$ of a matroid $\mathcal{M}$, there are algorithms called Pipage Rounding [9] and Swap Rounding [11] that, given a vector $\mathbf{x} \in P$ produce a set $S$ that is independent in $\mathcal{M}$ and also obeys $\mathbb{E}[f(S)] \geq F(\mathbf{x}) - o(1) \cdot f(OPT)$. Therefore, one can obtain an algorithm for maximizing $f$ subject to the matroid $\mathcal{M}$ by executing Measured Continuous Greedy with $P = P_{\mathcal{M}}$ and $T = 1$, and then applying either Pipage Rounding or Swap Rounding to the resulting vector; which yields an algorithm with the properties specified by Theorem 5.2.

We now describe the version of Measured Continuous Greedy that we analyze (given as Algorithm 5). For simplicity, we chose to analyze a continuous version of this algorithm that assumes direct access to the multilinear extension $F$ of the objective function rather than just to the objective function itself. We refer the reader to [22] for details about discretizing the algorithm and avoiding the assumption of direct access to $F$. We also note that the $o(1)$ error term in the approximation guarantee stated in

---

[6]In fact, Lemma 3.2 of [27] proves a more general result for $p$-set systems, but it implies the stated result since matroids are 1-set systems.

[7]A body $P \subseteq [0,1]^{\mathcal{N}}$ is *solvable* if one can efficiently optimize linear functions subject to it, and is *down-closed* if $\mathbf{y} \in P$ implies $\mathbf{x} \in P$ for every vector $\mathbf{x} \in [0,1]^{\mathcal{N}}$ obeying $\mathbf{x} \leq \mathbf{y}$ (this inequality should be understood to hold coordinate-wise).

Theorem E.1 is due to these issues. Our description of Measured Continuous Greedy requires some additional notation, namely, given two vectors $\mathbf{x}$ and $\mathbf{y}$, we denote by $\mathbf{x} \vee \mathbf{y}$ their coordinate-wise maximum and by $\mathbf{x} \odot \mathbf{y}$ their coordinate-wise multiplication.

Measured Continuous Greedy starts at "time" $0$ with the empty solution, and improves this solution during the time interval $[0, T]$. We denote the solution of the algorithm at time $t$ by $\mathbf{y}(t)$. At every time $t \in [0, T]$, the algorithm calculates a vector $\mathbf{w}$ whose $u$-coordinate is the gain that can be obtained by increasing this coordinates in the solution $\mathbf{y}(t)$ to be 1 (i.e., $w_u(t) = F(\mathbf{y}(t) \vee 1_{\{u\}}) - F(\mathbf{y}(t))$). Then, the algorithm finds a vector $\mathbf{x}(t) \in P$ that maximizes the objective function $\mathbf{w}(t) \cdot \mathbf{x}(t)$, and adds to the solution $\mathbf{y}(t)$ an infinitesimal part of $(1_{\mathcal{N}} - \mathbf{y}(t)) \odot \mathbf{x}(t)$ (to understand where the last expression comes from, we note that when $\mathbf{x}$ is integral, fully adding $(1_{\mathcal{N}} - \mathbf{y}(t)) \odot \mathbf{x}(t)$ to $\mathbf{y}(t)$ sets to 1 all the coordinates that are 1 in $\mathbf{x}(t)$, which matches the "spirit" of the definition of $\mathbf{w}$).

---

**Algorithm 5:** Measured Continuous Greedy($f, P, T$)

1   Let $\mathbf{y}(0) \leftarrow 1_{\varnothing}$.
2   **foreach** $t \in [0, T]$ **do**
3      For each $u \in \mathcal{N}$, let $w_u(t) \leftarrow F(\mathbf{y}(t) \vee 1_{\{u\}}) - F(\mathbf{y}(t))$.
4      Let $\mathbf{x}(t) \leftarrow \arg\max_{\mathbf{x} \in P}\{\mathbf{w}(t) \cdot \mathbf{x}\}$.
5      Increase $\mathbf{y}(t)$ at a rate of $\frac{d\mathbf{y}(t)}{dt} = (1_{\mathcal{N}} - \mathbf{y}(t)) \odot \mathbf{x}(t)$.
6   **return** $y(T)$.

---

The first step in the analysis of Measured Continuous Greedy is bounding the maximum value of the coordinates of the solution $\mathbf{y}(t)$.

**Lemma E.2.** *For every* $t \in [0, T]$, $\|\mathbf{y}(t)\|_{\infty} \leq 1 - e^{-t}$.

*Proof.* Fix an arbitrary element $u \in \mathcal{N}$, and let us explain why $y_u(t) \leq 1 - e^{-t}$. By Line 5 of Algorithm 5, $y_u(t)$ obeys the differential inequality

$$\frac{dy_u(t)}{dt} = (1 - y_u(t)) \cdot x_u(t) \leq 1 - y_u(t) \ ,$$

and the solution of this differential inequality for the initial condition $y_u = 0$ is

$$y_u(t) \leq 1 - e^{-t} \ . \qquad \square$$

We are now ready to prove Theorem E.1

*Proof of E.1.* Recall that $\mathbf{x}(t)$ is a vector inside $P$ for every time $t \in [0, T]$, and since $P$ is down-closed, $(1_{\mathcal{N}} - \mathbf{y}(t)) \odot \mathbf{x}(t)$ and $1_{\varnothing}$ both belong to $P$ as well. This means that for $T \leq 1$ the vector $\mathbf{y}(T) = (1 - T) \cdot 1_{\varnothing} + \int_0^T (1_{\mathcal{N}} - y(t)) \odot \mathbf{x}(t)dt$ is a convex combination of vectors in $P$, and therefore belongs to $P$ by the convexity of $P$.

It remains to lower bound the value of $F(\mathbf{y}(T))$. By the chain rule,

$$\frac{dF(\mathbf{y}(t))}{dt} = \sum_{u \in \mathcal{N}} \left( \frac{dy_u(t)}{dt} \cdot \frac{\partial F(\mathbf{y})}{\partial y_u}\Big|_{\mathbf{y}=\mathbf{y}(t)} \right) = \sum_{u \in \mathcal{N}} \left( (1 - y_u(t)) \cdot x_u(t) \cdot \frac{\partial F(\mathbf{y})}{\partial y_u}\Big|_{\mathbf{y}=\mathbf{y}(t)} \right) \ .$$

Since $F$ is multilinear, its partial derivative with respect to a single coordinate is equal to the difference between the value of the function for two different values of this coordinate over the difference between these values. Plugging this observation into the previous inequality yields

$$\frac{dF(\mathbf{y}(t))}{dt} = \sum_{u \in \mathcal{N}} \left( (1 - y_u(t)) \cdot x_u(t) \cdot \frac{F(\mathbf{y}(t) \vee 1_{\{u\}}) - F(\mathbf{y}(t))}{1 - y_u(t)} \right) = \mathbf{x}(t) \cdot \mathbf{w}(t) \ .$$

One possible candidate to be $\mathbf{x}(t)$ is $\mathbf{1}_{OPT}$. Hence, by the definition of $\mathbf{x}(t)$, $\mathbf{x}(t) \cdot \mathbf{w}(t) \geq \mathbf{1}_{OPT} \cdot \mathbf{w}(t)$. Combining this inequality with the previous one, we get

$$
\begin{aligned}
\frac{dF(\mathbf{y}(t))}{dt} &\geq \mathbf{1}_{OPT} \cdot \mathbf{w}(t) = \sum_{u \in OPT} \left[ F(\mathbf{y}(t) \vee 1_{\{u\}}) - F(\mathbf{y}(t)) \right] \\
&\geq F(\mathbf{y}(t) \vee \mathbf{1}_{OPT}) - F(\mathbf{y}(t)) \geq [1 - (1-m) \cdot \|\mathbf{y}(t)\|_\infty] \cdot f(OPT) - F(\mathbf{y}(t)) \\
&\geq [1 - (1-m)(1 - e^{-t})] \cdot f(OPT) - F(\mathbf{y}(t)) \\
&= [m + (1-m)e^{-t}] \cdot f(OPT) - F(\mathbf{y}(t)) \ ,
\end{aligned}
$$

where the second inequality holds by the submodularity of $f$, the penultimate inequality holds by Lemma 2.1, and the last inequality follows from Lemma E.2.

Solving the differential inequality that we got for the initial condition $F(\mathbf{y}(0)) \geq 0$ (which holds by the non-negativity of $f$) yields

$$
F(y(t)) \geq \left[ m(1 - e^{-t}) + (1-m)te^{-t} \right] \cdot f(OPT) \ ,
$$

and the theorem now follows by plugging $t = T$. $\qquad\qquad\square$

### E.3    Analysis of Random Greedy for Matroids

In this section we prove Theorem 5.3, which we repeat here for convenience.

**Theorem 5.3.** *For every $\varepsilon \in (0,1)$, Random Greedy for Matroids (Algorithm 6) has an approximation ratio of at least $\frac{1+m+e^{-2/(1-m)}}{4} - \varepsilon - o_k(1)$ for the problem of maximizing a non-negative $m$-monotone submodular function subject to a matroid constraint (except in the case of $m = 1$ in which the approximation ratio is $1/2 - \varepsilon - o_k(1)$), where $o_k(1)$ represents a term that diminishes with $k$.*

To prove the theorem, we first need to state the algorithm it refers to. Towards this goal, let us assume that the ground set $\mathcal{N}$ contains a set $D$ of $2k$ "dummy" elements that are known to the algorithm and have the following two properties.

- $f(S) = (S \setminus D)$ for every set $S \subseteq \mathcal{N}$.
- $S \in \mathcal{I}$ if and only if $S \setminus D \in \mathcal{I}$ and $|S| \leq k$.

This assumption is useful since it allows us to assume that the optimal solution $OPT$ is a base of $\mathcal{M}$, and thus, simplifies the description of our algorithm (Random Greedy for Matroids). We can justify our assumption using the following procedure: (i) add $2k$ dummy elements to the ground set, (ii) extend $f$ and $\mathcal{I}$ according to the above properties, (iii) execute Random Greedy for Matroids on the resulting instance, and (iv) remove from the output of the algorithm any dummy elements that end up in it. This procedure guarantees that any approximation guarantee obtained by Random Greedy for Matroids using our assumption can be obtained also without the assumption.

Our version of the Random Greedy for Matroids algorithm is given as Algorithm 6. Like the original version of the algorithm (due to [6]), our version starts with a base of $\mathcal{M}$ consisting only of dummy elements, and then modifies it in a series of iterations. In each iteration $i$, the algorithm starts with a solution $S_{i-1}$, and then identifies a base $M_i$ of $\mathcal{M}$ whose elements have the largest total marginal contribution with respect to $S_{i-1}$ ($M_i$ is also required to be disjoint from $S_{i-1}$). The algorithm then picks a uniformly random element $u_i \in S_{i-1}$, and adds it to the solution $S_{i-1}$ at the expense of an element $g_i(u_i)$ of $S_{i-1}$ given by a function $g_i$ that is chosen carefully (the existence of such a function follows, for example, from Corollary 39.12a of [48]).

As mentioned above, our version of Random Greedy for Matroids differs compared to the version of [6] in two respects. The first modification is in the number of iterations that the algorithm makes. To get the result of Buchbinder et al. [6], it suffices to use $k$ iterations. However, the optimal number of iterations increases with $m$, and therefore, our version of the algorithm uses $k/\varepsilon$ iterations for some parameter $\varepsilon \in (0,1)$ (we assume without loss of generality that $k/\varepsilon$ is integral; otherwise, we can replace $\varepsilon$ with a value which is smaller than $\varepsilon$ by at most a factor of 2 and has this property). Furthermore, since we do not want to assume knowledge of $m$ in the algorithm, we use a number of iterations that is appropriate for $m = 1$, which requires us to make the second modification to the algorithm; namely, we check whether replacing $g(u_i)$ with $u_i$ is beneficial, and make the swap only

---

**Algorithm 6:** Random Greedy for Matroids($f, \mathcal{M} = (\mathcal{N}, \mathcal{I}), \varepsilon$)

---

**1** Initialize $S_0$ to be an arbitrary base containing only elements of $D$.
**2 for** $i = 1$ **to** $k/\varepsilon$ **do**
**3**     Let $M_i \subseteq N$ be a base of $\mathcal{M}$ that contains only elements of $\mathcal{N} \setminus S_{i-1}$ and maximizes $\sum_{u \in M_i} f(u \mid S_{i-1})$ among all such bases.
**4**     Let $g_i$ be a function mapping each element of $M_i$ to an element of $S_{i-1}$ obeying $S_{i-1} - g_i(u) + u \in \mathcal{I}$ for every $u \in S_{i-1}$.
**5**     Let $u_i$ be a uniformly random element from $M_i$. **if** $f(S_{i-1} - g_i(u_i) + u_i) > f(S_{i-1})$
    **then** Let $S_i \leftarrow S_{i-1} - g_i(u_i) + u_i$.
**6**     **else** Let $S_i \leftarrow S_{i-1}$.
**7 return** $S_{k/\varepsilon}$.

---

if this is indeed the case. This guarantees that doing more iterations can never decrease the value of the algorithm's solution.

Since Theorem 5.3 is trivial for a constant $k$, we can assume in the analysis of Algorithm 6 that $k$ is larger than any given constant. The first step in this analysis is proving the following lower bound on the expected value of $OPT \cup S_i$.

**Observation E.3.** *For every integer $0 \leq i \leq k/\varepsilon$, $\mathbb{E}[f(OPT \cup S_i)] \geq \frac{1}{2}(1 + m + (1 - m)(1 - 2/k)^i) \cdot f(OPT)$.*

*Proof.* For every integer $0 \leq i \leq k/\varepsilon$ and element $u \in \mathcal{N} \setminus D$, let $p_{u,i}$ denote the probability $u$ belongs to $S_i$. We would like to argue that when $i > 0$, we have $p_{u,i} \leq p_{u,i-1}(1 - 2/k) + 1/k$. To see why this is the case, note that $u$ belongs to $S_i$ only if one of the following happens: (i) $u$ belongs to $S_{i-1}$ and is not removed from the solution (happens with probability $p_{u,i-1}(1 - 1/k)$ since $g_i(u_i)$ is a uniformly random element of $S_{i-1}$), or (ii) $u$ belongs to $M_{i-1}$ and is chosen as $u_i$ (happens with probability at most $(1 - p_{u,i})/k$). Therefore,

$$p_{u,i} \leq p_{u,i-1} \cdot (1 - 1/k) + (1 - p_{u,i-1})/k = p_{u,i-1} \cdot (1 - 2/k) + 1/k \ . \tag{8}$$

Next, we aim to prove by induction that $p_{u,i} \leq \frac{1}{2}(1 - (1 - 2/k)^i)$ for every integer $0 \leq i \leq k/\varepsilon$. For $i = 0$, this is true since $u \in \mathcal{N} \setminus D$ implies that $p_{u,0} = 0 = \frac{1}{2}(1 - (1 - 2/k)^0)$. Assume now that the claim holds for $i - 1$, and let us prove it for $i \geq 1$. By the induction hypothesis and Inequality (8),

$$p_{u,i} \leq p_{u,i-1}(1 - 2/k) + 1/k \leq \tfrac{1}{2}(1 - (1 - 2/k)^{i-1})(1 - 2/k) + 1/k = \tfrac{1}{2}(1 - (1 - 2/k)^i) \ .$$

The observation now follows since Lemma 2.1 guarantees that $\mathbb{E}[f(OPT \cup S_i)] = \mathbb{E}[f(OPT \cup (S_i \setminus D))] \geq (1 - (1 - m) \cdot \max_{u \in \mathcal{N} \setminus D} p_{i,u}) \cdot f(OPT)$. $\qquad\square$

Below we prove a lower bound on the value of the solution of Algorithm 6 after any number of iterations. However, to prove this lower bound we first need to prove the following technical observation.

**Observation E.4.** *For every positive integer $i$,*

$$\left(1 - \frac{2}{k}\right)^{i-1} \geq e^{-\frac{2i}{k}} - \frac{k}{i} \cdot o_k(1) \ .$$

*Proof.* Note that

$$e^{-\frac{2i}{k}} = \left(e^{-\frac{2}{k}}\right)^i \leq \left(1 - \frac{2}{k} + \frac{4}{k^2}\right)^i \leq \left(1 - \frac{2}{k}\right)^i + \frac{4}{k^2} \cdot i\left(1 - \frac{2}{k} + \frac{4}{k^2}\right)^{i-1}$$

$$\leq \left(1 - \frac{2}{k}\right)^i + \frac{4i}{k^2}\left(1 - \frac{1}{k}\right)^{i-1} \leq \left(1 - \frac{2}{k}\right)^i + \frac{4i}{k^2} \cdot e^{-\frac{i-1}{k}} \ ,$$

where the third inequality holds for $k \geq 4$, and the second inequality holds since the derivative of the function $(1 - 2/k + x)^i$ is $i(1 - 2/k + x)^{i-1}$, which implies

$$
\left(1 - \frac{2}{k} + \frac{4}{k^2}\right)^i = \left(1 - \frac{2}{k}\right)^i + \int_0^{4/k^2} i(1 - 2/k + x)^{i-1} dx
$$

$$
\leq \left(1 - \frac{2}{k}\right)^i + \frac{4i}{k^2}(1 - 2/k + 4/k^2)^{i-1} dx \ .
$$

To complete the proof of the observation, it remains to note that, since the maximum of the function $x^2 e^{-x}$ for $x \geq 0$ is $4e^{-2}$,

$$
\frac{4i}{k^2} \cdot e^{-\frac{i-1}{k}} \leq \frac{16e^{-2}}{i} \cdot e^{\frac{1}{k}} = \frac{k}{i} \cdot o_k(1) \ . \qquad \square
$$

We are now ready to prove the promised lower bound on the value of the solution $S_i$ of Algorithm 6 after any number of iterations.

**Lemma E.5.** *For every integer $0 \leq i \leq k/\varepsilon$,*

$$
\mathbb{E}[f(S_i)] \geq \left[\frac{1+m}{4} \cdot \left(1 - e^{-\frac{2i}{k}}\right) + \frac{(1-m)i}{2k} \cdot e^{-\frac{2i}{k}} - o_k(1)\right] \cdot f(OPT) \ .
$$

*Proof.* For $i = 0$ the lemma follows from the non-negativity of $f$ since the right hand side of the inequality that we need to prove is non-positive for $i = 0$. Together with Observation E.4, this implies that it suffices to prove the following inequality

$$
\mathbb{E}[f(S_i)] \geq \left[\frac{1+m}{4} \cdot \left(1 - \left(1 - \frac{2}{k}\right)^i\right) + \frac{(1-m)i}{2k} \cdot \left(1 - \frac{2}{k}\right)^{i-1}\right] \cdot f(OPT) \ , \qquad (9)
$$

and the rest of the proof is devoted to this goal.

Fix an arbitrary integer $1 \leq i \leq k/\varepsilon$. We would like to derive a lower bound on the expected marginal contribution of the element $u_i$ to the set $S_{i-1}$, and an upper bound on the expected marginal contribution of the element $g(u_i)$ to the set $S_{i-1} \setminus g(u_i)$. Let $A_{i-1}$ be an event fixing all random choices of Algorithm 6 up to iteration $i - 1$ (including), and let $\mathcal{A}_{i-1}$ be the set of all possible $A_{i-1}$ events. Conditioned on any event $A_{i-1} \in \mathcal{A}_{i-1}$, the sets $S_{i-1}$ and $M_i$ becomes deterministic, and we can define $M'_i$ as a set containing the elements of $OPT \setminus S_{i-1}$ plus enough dummy elements of $D \setminus S_{i-1}$ to make the size of $M'_i$ exactly $k$. Then,

$$
\mathbb{E}[f(u \mid S_{i-1}) \mid A_{i-1}] = \frac{\sum_{u \in M_i} f(u \mid S_{i-1})}{k} \geq \frac{\sum_{u \in M'_i} f(u \mid S_{i-1})}{k}
$$

$$
= \frac{\sum_{u \in OPT \setminus S_{i-1}} f(u \mid S_{i-1})}{k} \geq \frac{f(OPT \cup S_{i-1}) - f(S_{i-1})}{k} \ ,
$$

where $S_i$, $M_i$ and $M'_i$ represent here their values conditioned on $A_i$, the first inequality follows from the definition of $M_i$ and the second inequality holds by the submodularity of $f$. Similarly,

$$
\mathbb{E}[f(g(u_i) \mid S_{i-1} - g(u_i)) \mid A_{i-1}] = \frac{\sum_{u \in M_i} f(g(u_i) \mid S_{i-1} - g(u))}{k}
$$

$$
\leq \frac{f(S_{i-1}) - f(\varnothing)}{k} \leq \frac{f(S_{i-1})}{k} \ ,
$$

where the first inequality follows from the submodularity of $f$. Taking expectation over the event $A_{i-1}$, we get

$$
\mathbb{E}[f(u_i \mid S_{i-1})] \geq \frac{\mathbb{E}[f(OPT \cup S_{i-1})] - \mathbb{E}[f(S_{i-1})]}{k}
$$

$$
\geq \frac{\frac{1}{2}(1 + m + (1 - m)(1 - 2/k)^{i-1}) \cdot f(OPT) - \mathbb{E}[f(S_{i-1})]}{k} \ ,
$$

where the last inequality is due to Observation E.3, and

$$\mathbb{E}[f(g(u_i) \mid S_{i-1} - g(u_i))] \le \frac{\mathbb{E}[f(S_{i-1})]}{k} \quad .$$

Combing the last two inequalities now yields

$$
\begin{aligned}
\mathbb{E}[f(S_i)] &\ge \mathbb{E}[f(S_{i-1} - g(u_i) + u_i)] \qquad\qquad\qquad\qquad\qquad\qquad (10)\\
&= \mathbb{E}[f(S_{i-1})] + \mathbb{E}[f(u_i \mid S_{i-1} - g(u_i))] - \mathbb{E}[f(g(u_i) \mid S_{i-1} - g(u_i))]\\
&\ge \mathbb{E}[f(S_{i-1})] + \mathbb{E}[f(u_i \mid S_{i-1})] - \mathbb{E}[f(g(u_i) \mid S_{i-1} - g(u_i))]\\
&\ge \left(1 - \frac{2}{k}\right) \cdot \mathbb{E}[f(S_{i-1})] + \frac{\frac{1}{2}(1 + m + (1-m)(1-2/k)^{i-1}) \cdot f(OPT)}{k} \quad,
\end{aligned}
$$

where the first inequality follows from the submodularity of $f$ since $g(u_i) \neq u_i$ because $g(u_i) \in S_{i-1}$ and $u_i \in M_i$.

Since Inequality (10) holds for every integer $1 \le i \le k/\varepsilon$, we can use it repeatedly to get, for every integer $0 \le i \le k/\varepsilon$,

$$
\begin{aligned}
\mathbb{E}[f(S_i)] \ge \frac{1}{2k}&\left[(1+m)\sum_{j=1}^{i}\left(1 - \frac{2}{k}\right)^{i-j} + (1-m)\sum_{j=1}^{i}\left(1 - \frac{2}{k}\right)^{i-1}\right] \cdot f(OPT)\\
&+ \left(1 - \frac{2}{k}\right)^{i} \cdot f(S_0) \quad.
\end{aligned}
$$

Since the non-negativity of $f$ guarantees that $f(S_0) \ge 0$, the last inequality implies Inequality (9), and therefore, completes the proof of the lemma. $\qquad\square$

One can show that the lower bound for $f(S_i)$ proved by Lemma E.5 is maximized when $i = k/(1-m)$. Unfortunately, we cannot simply plug this $i$ value into the lower bound due to two issues: this value of $i$ might not be integral, and this value of $i$ might be larger than the number $k/\varepsilon$ of iterations. The following two lemmata prove the approximation guarantee of Theorem 5.3 despite these issues, and together they complete the proof of the theorem.

**Lemma E.6.** *When $m \le 1 - \varepsilon$, the approximation ratio of Algorithm 6 is at least*

$$\frac{1 + m + e^{-2/(1-m)}}{4} - o_k(1) \quad.$$

*Proof.* Let $i' = \lfloor k/(1-m) \rfloor$. Due to the condition of the lemma, Algorithm 6 makes at least $i'$ iterations. Furthermore, since Algorithm 6 makes a swap in its solution only when this swap is beneficial, the expected value of the output of the algorithm is at least

$$
\begin{aligned}
\mathbb{E}[f(S_{i'})] &\ge \left[\frac{1+m}{4} \cdot \left(1 - e^{-\frac{2i'}{k}}\right) + \frac{(1-m)i'}{2k} \cdot e^{-\frac{2i'}{k}} - o_k(1)\right] \cdot f(OPT)\\
&\ge \left[\frac{1+m}{4} \cdot \left(1 - e^{\frac{2}{k} - \frac{2}{1-m}}\right) + \frac{k-1}{2k} \cdot e^{-\frac{2}{1-m}} - o_k(1)\right] \cdot f(OPT)\\
&\ge \left[\frac{1+m}{4} \cdot \left(1 - e^{-\frac{2}{1-m}}\right) - \frac{e^{\frac{2}{k}}-1}{2} + \frac{1}{2} \cdot e^{-\frac{2}{1-m}} - \frac{1}{2k} - o_k(1)\right] \cdot f(OPT) \quad,
\end{aligned}
$$

where the first inequality follows from Lemma E.5, and the second inequality holds since $k/(1 - m) - 1 \le i' \le k/(1-m)$. Since the terms $\frac{e^{2/k}-1}{2}$ and $\frac{1}{2k}$ are both diminishing with $k$ (and therefore, can be replaced with $o_k(1)$), the last inequality implies the lemma. $\qquad\square$

**Lemma E.7.** *When $1 - \varepsilon \le m < 1$, the approximation ratio of Algorithm 6 is at least*

$$\frac{1 + m + e^{-2/(1-m)}}{4} - \varepsilon - o_k(1) \quad,$$

*and when $m = 1$ the approximation ratio of this algorithm is at least $1/2 - \varepsilon - o_k(1)$.*

*Proof.* The output set of Algorithm 6 is $f(S_{k/\varepsilon})$. By Lemma E.5, the expected value of this set is at least

$$\mathbb{E}[f(S_{k/\varepsilon})] \geq \left[\frac{1+m}{4} \cdot \left(1 - e^{-\frac{2}{\varepsilon}}\right) + \frac{1-m}{2\varepsilon} \cdot e^{-\frac{2}{\varepsilon}} - o_1(k)\right] \cdot f(OPT)$$

$$\geq \left[\frac{1+m}{4} \cdot \left(1 - e^{-\frac{2}{\varepsilon}}\right) - o_k(1)\right] \cdot f(OPT)$$

$$\geq \left[\frac{1+m}{4} \cdot \left(1 - \frac{1}{1+2/\varepsilon}\right) - o_k(1)\right] \cdot f(OPT)$$

$$= \left[\frac{1+m}{2(\varepsilon+2)} - o_k(1)\right] \cdot f(OPT) \geq \left[\frac{1+m}{4} - \frac{\varepsilon}{4} - o_k(1)\right] \cdot f(OPT) \ ,$$

where the third inequality holds since for every $x \geq 0$, $\ln(1/(1+x)) = \ln(1 - x/(1+x)) \geq -\frac{x/(1+x)}{1-x/(1+x)} = -x$.

The above inequality completes the proof for the case of $m = 1$. To complete the proof also for the case of $1 - \varepsilon \leq m < 1$, it suffice to observe that in this case

$$e^{-2/(1-m)} \leq e^{-2/\varepsilon} \leq \frac{1}{1+2/\varepsilon} \leq \frac{\varepsilon}{2} \ . \qquad \square$$

### E.4 Inapproximability for a Matroid Constraints

In this section we state and prove the inapproximability result mentioned in Section 5.

**Theorem E.8.** *For any constant $\varepsilon > 0$, no polynomial time algorithm can obtain an approximation ratio of*

$$\min_{\alpha \in [0,1]} \max_{x \in [0,1/2]} \{\alpha(mx^2 + 2x - 2x^2) + 2(1-\alpha)(1 - e^{-1/2})(1 - (1-m)x)\} + \varepsilon$$

*for the problem of maximizing a non-negative $m$-monotone submodular function subject to a matroid constraint.*

The proof of Theorem E.8 is very similar to the proof of Theorem D.4. Recall that in Section D.3, we proved Theorem D.4 by constructing an instance $\mathcal{I}$ of submodular maximization subject to a cardinality constraint, and then applying Theorem 3.2 to this instance. The proof of Theorem E.8 is based on an instance $\mathcal{I}'$ of submoduar maximization subject to a matroid constrained that is identical to $\mathcal{I}$ except for the following difference. In $\mathcal{I}$, the constraint is a cardinality constraint allowing the selection of up to 2 elements from the ground set $\mathcal{N} = \{a, b\} \cup \{a_i, b_i \mid i \in [r]\}$. In $\mathcal{I}'$, we have instead a (simplified) partition matroid constraint allowing the selection of up to 1 element from $\{a, b\}$ and up to 1 element from $\{a_i, b_i \mid i \in [r]\}$.

Since the instances $\mathcal{I}$ and $\mathcal{I}'$ have the same objective function, the properties of this function stated in Lemma D.5 apply to both of them. Furthermore, one can verify that $\mathcal{I}'$ is strongly symmetric with respect to the group $\mathcal{G}$ of permutation defined in Section D.3. Therefore, we concentrate on analyzing the symmetry gap of $\mathcal{I}'$.

**Lemma E.9.** *The symmetry gap of $\mathcal{I}'$ is at most*

$$\max_{x \in [0,1/2]} \{\alpha(mx^2 + 2x - 2x^2) + 2(1-\alpha)[1 - (1 - 1/(2r))^r](1 - (1-m)x)\}$$

$$\leq \max_{x \in [0,1/2]} \{\alpha(mx^2 + 2x - 2x^2) + 2(1-\alpha)(1 - e^{-1/2})(1 - (1-m)x)\} + 1/(2r) \ .$$

*Proof.* One possible feasible solution for $\mathcal{I}'$ is the set $\{a, b_1\}$ whose value according to $f$ is 1. Therefore, the value of the optimal solution for $\mathcal{I}'$ is at least 1. Since the symmetry gap is the ratio between the value of the best symmetric solution and the value of the best solution, to prove the lemma it remains to argue that the best symmetric solution for $\mathcal{I}$ has a value of at most $\max_{x \in [0,1/2]} \{\alpha(mx^2 + 2x - 2x^2) + 2(1-\alpha)[1 - (1 - 1/(2r))^r](1 - (1-m)x)\}$.

We remind the reader that a symmetric solution for $\mathcal{I}'$ is $\bar{\mathbf{y}} = \mathbb{E}_{\sigma \in \mathcal{G}}[\mathbf{y}]$ for some vector $\mathbf{y} \in [0,1]^{\mathcal{N}}$ obeying $y_a + y_b \leq 1$ and $\sum_{i=1}^r y_{a_i} + y_{b_i} \leq 1$. Since $a$ and $b$ can be exchanged with each other

by the permutations of $\mathcal{G}$, the values of the coordinates of $a$ and $b$ in $\bar{\mathbf{y}}$ must be equal to each other. Similarly, every two elements of $\{a_i, b_i \in i \in [r]\}$ can be exchanged by the permutations of $\mathcal{G}$, and therefore, the values of the coordinates of all these elements in $\bar{\mathbf{y}}$ must be all identical. Thus, any symmetric solution $\bar{\mathbf{y}}$ can be represented as

$$\bar{y}_u = \begin{cases} x & \text{if } u = a \text{ or } u = b \ , \\ z & \text{if } u \in \{a_i, b_i \mid i \in [r]\} \end{cases}$$

for some values $x \in [0, 1/2]$ and $z \in [0, 1/(2r)]$. The value of this solution (according to the multilinear extension $F$ of $f$) is

$$\begin{aligned} &\alpha[m(1 - (1-x)^2) + 2(1-m)x(1-x)] + 2(1-\alpha)(1 - (1-z)^r)(1 - (1-m)x) \\ =\ &\alpha(mx^2 + 2x - 2x^2) + 2(1-\alpha)(1 - (1-z)^r)(1 - (1-m)x) \\ \leq\ &\alpha(mx^2 + 2x - 2x^2) + 2(1-\alpha)(1 - (1 - 1/(2r))^r)(1 - (1-m)x) \ . \end{aligned}$$

Therefore, one can obtain an upper bound on the value of the best symmetric solution for $\mathcal{I}'$ by taking the maximum of the last expression over all the values that $x$ can take, which completes the proof of the lemma. $\qquad\square$

Since any refinement of a (simplified) partition matroid constraint is a (generalized) partition matroid constraint on its own right, plugging Lemmata D.5 and Lemma E.9 into Theorem 3.2 yields the following corollary.

**Corollary E.10.** *For every constant $\varepsilon' > 0$, no polynomial time algorithm for maximizing a non-negative $m$-monotone submodular function subject to a matroid contraint obtains an approximation ratio of*

$$\max_{x \in [0, 1/2]} \{\alpha(mx^2 + 2x - 2x^2) + 2(1-\alpha)(1 - e^{-1/2})(1 - (1-m)x)\} + 1/(2r) + \varepsilon' \ .$$

Theorem E.8 now follows from the last corollary by choosing $\varepsilon' = \varepsilon/2$, $r = \lceil \varepsilon^{-1} \rceil$ and

$$\alpha = \arg\min_{\alpha' \in [0,1]} \max_{x \in [0, 1/2]} \{\alpha'(mx^2 + 2x - 2x^2) + 2(1-\alpha')(1 - e^{-1/2})(1 - (1-m)x)\} \ .$$

# F  Personalized Image Summarization

Consider a setting in which we get as input a collection $\mathcal{N}$ of images from $\ell$ disjoint categories (e.g., birds, dogs, cats) and the user specifies $r \in [\ell]$ categories, and then demands a subset of the images in these categories that summarize all the images of the categories. Following [40] again, to evaluate a given subset of images we use the function $f(S) = \sum_{u \in \mathcal{N}} \max_{v \in S} s_{u,v} - \frac{1}{|\mathcal{N}|} \sum_{u \in S} \sum_{v \in S} s_{u,v}$, where $s_{u,v}$ is a non-negative similarity between $u$ and $v$.

One can verify that the above function $f$ is non-negative and submodular. Unfortunately, this function can have a very low monotonicity ratio. To compensate for this, we observe that most the analyses we described in the previous sections use the monotonicity ratio only to show that $f(S \cup T) \geq m \cdot f(S)$ for sets $S$ and $T$ that are feasible. This motivates the following weak version of the monotonicity ratio. We note that many continuous properties of set functions have such weak versions. For example, the original paper presenting the submodularity-ratio [16] presented in fact the weak version of this property, and the non-weak version was only formulated at a later point.

**Definition F.1.** Consider maximization of a non-negative function $f$ subject to some constraint. In the context of this constraint, we say that $f$ is *$m$-weakly monotone* if $f(S \cup T) \geq m \cdot f(S)$ holds for every two feasible sets $S$ and $T$.

**Theorem F.2.** *The objective function $f$ of personalized image summarization is $1 - \frac{2k}{|\mathcal{N}|}$-weakly monotone when the size of feasible solutions is at most $k$ for some $1 \leq k \leq |\mathcal{N}|$.*

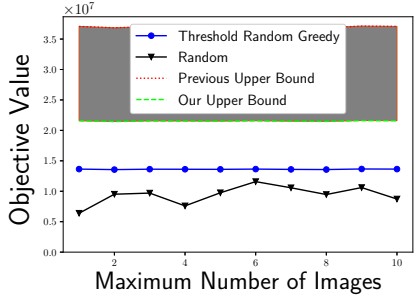
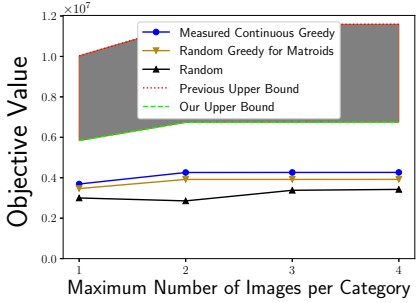

(a) Results for Personalized Image Summarization with a cardinality constraint for varying number of images in the summary produced.

(b) Results for Personalized Image Summarization with a matroid constraint. The $x$-axis is the number $k$ of images allowed from each category.

Figure 3: Personalized Image Summerization Results

*Proof.* When $k \geq |\mathcal{N}|/2$, the theorem is trivial. Thus, we can assume below $k < |\mathcal{N}|/2$. Consider two feasible sets $S, T \in \mathcal{N}$, and let us lower bound $f(S \cup T)$.

$$
\begin{aligned}
f(S \cup T) &= \sum_{u \in \mathcal{N}} \max_{v \in S \cup T} s_{u,v} - \frac{1}{|\mathcal{N}|} \sum_{u \in S \cup T} \sum_{v \in S \cup T} s_{u,v} \\
&\geq \sum_{u \in \mathcal{N}} \max_{v \in S \cup T} s_{u,v} - \frac{|S \cup T|}{|\mathcal{N}|} \sum_{u \in S \cup T} \max_{v \in S \cup T} s_{u,v} \\
&\geq \sum_{u \in \mathcal{N}} \max_{v \in S \cup T} s_{u,v} - \frac{2k}{|\mathcal{N}|} \sum_{u \in S \cup T} \max_{v \in S \cup T} s_{u,v} \\
&= \left(1 - \frac{2k}{|\mathcal{N}|}\right) \sum_{u \in \mathcal{N}} \max_{v \in S \cup T} s_{u,v} \geq \left(1 - \frac{2k}{|\mathcal{N}|}\right) \sum_{u \in \mathcal{N}} \max_{v \in S} s_{u,v} \ .
\end{aligned}
$$

Using this lower bound, we now get

$$
f(S) = \sum_{u \in E} \max_{v \in S} s_{u,v} - \frac{1}{|\mathcal{N}|} \sum_{u \in S} \sum_{v \in S} s_{u,v} \leq \sum_{u \in E} \max_{v \in S} s_{u,v} \leq \frac{f(S \cup T)}{1 - 2k/|\mathcal{N}|} \ ,
$$

which completes the proof of the theorem since $S$ and $T$ have been chosen as arbitrary feasible sets. □

Our experiments for this setting are based on a subset of the CIFAR-10 dataset [33] including 10,000 Tiny Images. These images belong to 10 classes, with 1000 images per class. Each image consists of $32 \times 32$ RGB pixels represented by a 3072 dimensional vector. To compute the similarity $s_{u,v}$ between images, we used the dot product.

In our first experiment, we simply looked for a summary consisting of a limited number of images. Since this is a cardinality constraint, we again used the scarecrow algorithm Random and the accelerated versions mentioned in Section 6.1 of the algorithms from Section 4. In Figure 3a we depict the outputs of Threshold Random Greedy and Random for various limits on the number of images in the summary (like in Section 6.1 we omit the other non-scarecrow algorithms since their performance is essentially identical to the one of Threshold Random Greedy, and we refer the reader to Appendix I for more detail). Figure 3a also includes the upper bounds on the optimal solution obtained via the previous approximation ratio for Random Greedy and our improved approximation ratio (the area between the two upper bounds is shaded). We can see that the upper bound obtained via our improved approximation ratio is much tighter, and this upper bound also demonstrates that the gap between the non-scarecrow and the scarecrow algorithms is significant compared to the optimal solution.

In our second experiment, we looked for a summary containing up to $k$ images from each category selected by the user for some parameter $k$ (we assumed in the experiment that the user chose the

categories: "airplane", "automobile" and "bird"). Since this is a (generalized partition) matroid constraint, in this experiment we used versions of the algorithms from Section 5. Specifically, we used Random Greedy for Matroids and an accelerated version of Measured Continuous Greedy based on the acceleration technique underlying the Accelerated Continuous Greedy of [1]. Additionally, we used in this experiment a scarecrow algorithm called Random that outputs a set containing a random selection of $k$ images from each one of the chosen categories. The values of the outputs of all these algorithms are depicted in Figure 3b (values shown are averaged over 10 executions).

Figure 3b also includes upper bounds on the value of the optimal solution. The previous upper bound is computed based on the previously known approximation ratios of the algorithms, and our upper bound is computed based on the approximation ratios proved in Theorems 5.2 and 5.3 and the weak monotonicity ratio proved in Theorem F.2.[8] As is evident from the similarity between Figures 3a and 3b, our observations from the first experiment extend also the more general constraint considered in the current experiment.

## G    Ride-Share Optimization

In this application, given a set $R$ of possible customer locations specified as (latitude, longitude) coordinate pairs, we aim to find a subset of these locations that will serve as waiting locations for drivers and minimizes the distance from each costumer to her closest driver. This problem can be modeled using the classical facility location problem, whose objective is known to be monotone and submodular. More formally, Mitrovic et al. [41] defined for every set $T$ of locations the objective value $f(T)$ as

$$f(T) = \sum_{a \in R} \max_{b \in T} c(a, b) \ ,$$

where $c(a, b)$ is a convenience score defined by $c(a, b) \triangleq 2 - \frac{2}{1 + e^{-200d(a,b)}}$, and $d(a, b) = |x_a - x_b| + |y_a - y_b|$ is the Manhattan distance between the points $a$ and $b$.

One drawback of the above objective function is that it does not promote diversity in the set of chosen locations. For example, imagine a scenario where, due to congestion or road maintenance in a specific area, traffic in and out of this area is slow or completely blocked. If all the selected waiting locations happen to be inside the affected area (i.e., there is no diversity in the selected locations), it will be difficult for the drivers to move between the waiting locations and the customers. To avoid such unfavorable scenarios, a diversity component should be added to the objective function. However, when a diversity component is added, the function becomes non-monotone (but still submodular), making the approximation guarantees of state-of-the-art algorithms much lower, as is discussed above.

Using the monotonicity ratio, the effect of the diversity component on the approximation guarantee can be significantly reduced. For example, a natural way to add a diversity component is demonstrated by the next objective function.

$$f(T) = \sum_{a \in R} \max_{b \in T} c(a, b) - \frac{1}{|R|} \sum_{x \in T} \sum_{y \in T} c(x, y) \ .$$

One can note that the last function has the same form as the function discussed in Appendix F. Hence, by Theorem F.2, the previous function is $(1 - \frac{2k}{|R|})$-weakly monotone, where $1 \leq k \leq |R|$ is the maximum size of a feasible solution.

## H    Proofs of Section 6

In this section we prove the theorems from Section 6.

---

[8]From a purely formal point of view this upper bound is not fully justified since Measured Continuous Greedy is a rare example of an algorithm whose analysis cannot use in a black box fashion the weak monotonicity ratio instead of the monotonicity ratio. However, due to probabilistic concentration, we expect the upper bound to still hold up to at most a small error.

## H.1 Proof of Theorem 6.1

**Theorem 6.1.** *The objective function $f$ is monotone for $0 \leq \lambda \leq 1/2$ and $2(1-\lambda)$-monotone for $1/2 \leq \lambda \leq 1$.*

*Proof.* We first prove the first part of the theorem. Thus, we assume $\lambda \leq 1/2$, and we need to show that for arbitrary set $S \subseteq \mathcal{N}$ and element $u \in \mathcal{N} \setminus S$ the marginal contribution $f(u \mid S)$ is non-negative. This holds because

$$
f(u \mid S) = \sum_{v \in \mathcal{N}} s_{v,u} - \lambda \left[ \sum_{v \in S} s_{u,v} + \sum_{v \in S} s_{v,u} + s_{u,u} \right]
$$

$$
= \sum_{v \in \mathcal{N}} s_{v,u} - \lambda \left[ 2 \sum_{v \in S} s_{v,u} + s_{u,u} \right] \geq \sum_{v \in \mathcal{N}} s_{v,u} - \sum_{v \in S} s_{v,u} - s_{u,u} \geq 0 \; ,
$$

where the second equality holds because $s_{u,v} = s_{v,u}$, and the first inequality holds since $\lambda \leq 1/2$ in the case we consider and the $s_{u,v}$ values are non-negative.

It remains to prove the second part of the theorem. Thus, we assume from now on $\lambda \in [1/2, 1]$, and we consider two sets $S \subseteq T \subseteq \mathcal{N}$. To prove the theorem we need to show that $f(T) \geq 2(1-\lambda) \cdot f(S)$. The first step towards showing this is to prove the following lower bound on $f(S)$.

$$
f(S) = 2(1-\lambda) \cdot f(S) + (2\lambda - 1) \cdot \left[ \sum_{u \in \mathcal{N}} \sum_{v \in S} s_{u,v} - \lambda \sum_{u \in S} \sum_{v \in S} s_{u,v} \right] \tag{11}
$$

$$
\geq 2(1-\lambda) \cdot f(S) + (2\lambda - 1) \cdot \sum_{u \in T \setminus S} \sum_{v \in S} s_{u,v}
$$

$$
= 2(1-\lambda) \cdot f(S) + (2\lambda - 1) \cdot \sum_{u \in S} \sum_{v \in T \setminus S} s_{u,v} \; , \tag{12}
$$

where the inequality holds since $\lambda \leq 1$, and the second equality holds since $s_{u,v} = s_{v,u}$. Using this lower bound, we now get

$$
f(T) = f(S) + \sum_{u \in \mathcal{N}} \sum_{v \in T \setminus S} s_{u,v} - \lambda \left[ \sum_{u \in S} \sum_{v \in T \setminus S} s_{u,v} + \sum_{u \in T \setminus S} \sum_{v \in S} s_{u,v} + \sum_{u \in T \setminus S} \sum_{v \in T \setminus S} s_{u,v} \right]
$$

$$
= f(S) + \sum_{u \in \mathcal{N}} \sum_{v \in T \setminus S} s_{u,v} - \lambda \left[ 2 \sum_{u \in S} \sum_{v \in T \setminus S} s_{u,v} + \sum_{u \in T \setminus S} \sum_{v \in T \setminus S} s_{u,v} \right]
$$

$$
\geq f(S) + (1 - 2\lambda) \cdot \sum_{u \in S} \sum_{v \in T \setminus S} s_{u,v} \geq 2(1-\lambda) \cdot f(S) \; ,
$$

where the first inequality holds since $\lambda \leq 1$, and the second inequality holds by Inequality (11). $\quad\square$

## H.2 Proof of Theorem 6.2

**Theorem 6.2.** *For $\beta \in (0, 1/2)$, the objective function $F$ given by Equation (1) is $\frac{(1-2\beta) \cdot \alpha}{1+\alpha}$-monotone. Furthermore, when $\min_{\bar{0} \leq x \leq u}(\frac{1}{2} x^T H x + h x) \geq 0$, $F$ is even $(1 - 2\beta)$-monotone.*

*Proof.* Fix two vectors $\bar{0} \leq \mathbf{x} \leq \mathbf{y} \leq \mathbf{u}$. We begin this proof by providing a lower bound on $F(\mathbf{y})$ and an upper bound on $F(\mathbf{x})$. The lower bound on $F(\mathbf{y})$ is as following.

$$
F(\mathbf{y}) = \frac{1}{2} \mathbf{y}^T \mathbf{H} \mathbf{y} + \mathbf{h}^T \mathbf{y} + c \geq \min_{\bar{0} \leq \mathbf{x} \leq \mathbf{u}} \left( \frac{1}{2} \mathbf{x}^T \mathbf{H} \mathbf{x} + \mathbf{h} \mathbf{x} \right) + c \; .
$$

To get the upper bound on $F(\mathbf{x})$, we first need to prove an upper bound on $c$.

$$
c \geq - \min_{\bar{0} \leq \mathbf{x} \leq \mathbf{u}} \left( \frac{1}{2} \mathbf{x}^T \mathbf{H} \mathbf{x} + \mathbf{h}^T \mathbf{x} \right) = - \min_{\bar{0} \leq \mathbf{x} \leq \mathbf{u}} \left( \frac{1}{2} \mathbf{x}^T \mathbf{H} \mathbf{x} - \beta \mathbf{u}^T \mathbf{H} \mathbf{x} \right) \geq - \left( \frac{1}{2} - \beta \right) \mathbf{u}^T \mathbf{H} \mathbf{u} \; .
$$

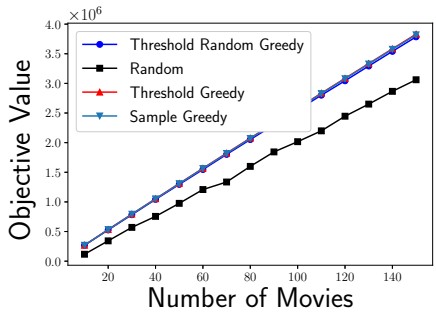
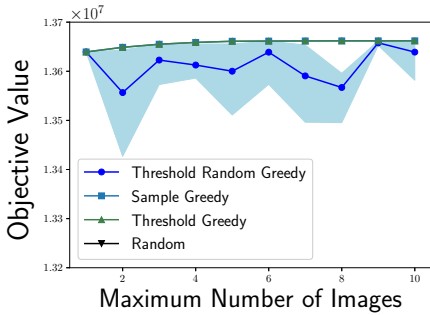

(a) Performance of the various algorithms in the movie recommendation setting (for $\lambda = 0.75$).

(b) Performance of the non-scarecrow algorithms in the image summarization setting with a cardinality constraint. The shaded area represents the standard deviation of Threshold Random Greedy.

Figure 4: Comparing the performance of algorithms for a cardinality constraint in our experiments.

The promised upper bound on $F(\mathbf{x})$ now follows.

$$F(\mathbf{x}) = \frac{1}{2}\mathbf{x}^T\mathbf{H}\mathbf{x} + \mathbf{h}^T\mathbf{x} + c \leq \mathbf{h}^T\mathbf{x} + c \leq \mathbf{h}^T\mathbf{u} + c = -\beta\mathbf{u}^T\mathbf{H}\mathbf{u} + c \leq \frac{\beta c}{1/2 - \beta} + c = \frac{c}{1 - 2\beta} \ ,$$

where the first inequality holds since $\mathbf{H}$ is non-positive, and the second inequality holds since $\mathbf{h}$ is non-negative.

Recall now that $c = -M + \alpha|M|$, which implies

$$\min_{0 \leq \mathbf{x} \leq \mathbf{u}} \left( \frac{1}{2}\mathbf{x}^T\mathbf{H}\mathbf{x} + \mathbf{h}^T\mathbf{x} \right) = M \geq -\frac{c}{1 + \alpha} \ ,$$

and therefore,

$$F(\mathbf{y}) \geq -\frac{c}{1 + \alpha} + c = \frac{c\alpha}{1 + \alpha} \geq \frac{(1 - 2\beta)\alpha}{1 + \alpha} \cdot F(\mathbf{x}) \ .$$

It remains to consider the case in which $\min_{\bar{0} \leq \mathbf{x} \leq \mathbf{u}} \left( \frac{1}{2}\mathbf{x}^T\mathbf{H}\mathbf{x} + \mathbf{h}\mathbf{x} \right) \geq 0$. In this case

$$F(\mathbf{y}) \geq c \geq (1 - 2\beta) \cdot F(\mathbf{x}) \ . \qquad \square$$

## I  Additional Plots for Section 6

As discussed in Section 6, the various algorithms we use in the context of a cardinality constraint have very similar empirical performance. Figure 4a presents the performance of all these algorithms in the movie recommendation setting with the number of movies in the summery varying. One can observe that the lines of the three non-scarecrow algorithms almost overlap. Figure 4b presents the performance of the non-scarecrow algorithms in the image summarization setting. In this figure we had to ignore the scarecrow algorithm Random because otherwise the lines of the three non-scarecrows algorithms are indistinguishable. Furthermore, we had to zoom in on a very small range of $y$-axis values. Despite these steps, the lines of Sample Greedy and Threshold Greedy still completely overlap, but the large zoom allows us to see that Threshold Random Greedy is marginally worse.

## J  Maximizating DR-submodular Functions subject to a Polytope Constraint

There are (at least) two natural ways in which the notion of submodularity can be extended from set functions to continuous functions. The more restrictive of these is known as DR-submodularity (first defined by [3]). Given a domain $\mathcal{X} = \prod_{i=1}^n \mathcal{X}_i$, where $\mathcal{X}_i$ is a closed range in $\mathbb{R}$ for every $i \in [n]$,

a function $F\colon \mathcal{X} \to \mathbb{R}$ is *DR-submodular* if for every two vectors $\mathbf{a}, \mathbf{b} \in \mathcal{X}$, positive value $k$ and coordinate $i \in [n]$ the inequality

$$F(\mathbf{a} + k\mathbf{e}_i) - F(\mathbf{a}) \geq F(\mathbf{b} + k\mathbf{e}_i) - F(\mathbf{b})$$

holds whenever $\mathbf{a} \leq \mathbf{b}$ and $\mathbf{b} + k\mathbf{e}_i \in \mathcal{X}$ (here and throughout the section $\mathbf{e}_i$ denotes the standard $i$-th basis vector, and comparison between two vectors should be understood to hold coordinate-wise). If $F$ is continuously differentiable, then the above definition of DR-submodulrity is equivalent to $\nabla F$ being an antitone mapping from $\mathcal{X}$ to $\mathbb{R}^n$ (i.e., $\nabla F(\mathbf{a}) \geq \nabla F(\mathbf{b})$ for every two vectors $\mathbf{a}, \mathbf{b} \in \mathcal{X}$ that obey $\mathbf{a} \leq \mathbf{b}$). Moreover, when $F$ is twice differentiable, it is DR-submodular if and only if its Hessian is non-positive at every vector $\mathbf{x} \in \mathcal{X}$.

In this section we consider the problem of maximizing a non-negative DR-submodular function $F\colon 2^{\mathcal{N}} \to \mathbb{R}_{\geq 0}$ subject to a solvable down-closed[9] convex body $P \subseteq \mathcal{X}$ (usually polytope) constraint. As is standard when dealing with problems of this kind, we assume that $F$ is $L$-smooth, i.e., for every two vectors $\mathbf{x}, \mathbf{y} \in \mathcal{X}$ it obeys

$$\|\nabla F(\mathbf{x}) - \nabla F(\mathbf{y})\|_2 \leq L\|\mathbf{x} - \mathbf{y}\|_2$$

for some non-negative parameter $L$. Additionally, for simplicity, we assume that $\mathcal{X} = [0,1]^n$. This assumption is without loss of generality because the natural mapping from $\mathcal{X}$ to $[0,1]^n$ preserves all our results.

We analyze a variant of the Frank-Wolfe algorithm for the above problem due to [2] called Non-monotone Frank-Wolfe. This variant was motivated by the Measured Continuous Greedy algorithm studied in Section E.2, and its assumes access to the first order derivatives of $F$. The details of the algorithm we consider appear as Algorithm 7. This algorithm gets a quality control parameter $\varepsilon \in (0,1)$, and it is assumed that $\varepsilon^{-1}$ is an integer (if this is not the case, one can fix that by reducing $\varepsilon$ by at most a factor 2). Algorithm 7 and its analysis also employ the notation defined in Section E.2, namely, given two vectors $\mathbf{x}, \mathbf{y}$, their coordinate-wise multiplication is denoted by $\mathbf{x} \odot \mathbf{y}$. Additionally, we denote by $\bar{0}$ and $\bar{1}$ the all zeros and all ones vectors, respectively.

---

**Algorithm 7:** Non-monotone Frank-Wolfe($\varepsilon$)

---

1 Let $\mathbf{y}^{(0)} \leftarrow \bar{0}$ and $t = 0$.
2 **while** $t \leq 1$ **do**
3     $\mathbf{s}^{(t)} \leftarrow \arg\max_{\mathbf{x} \in P} \mathbf{x} \cdot ((\bar{1} - \mathbf{y}^{(t)}) \odot \nabla F(\mathbf{y}^{(t)}))$.
4     $\mathbf{y}^{(t+\varepsilon)} \leftarrow \mathbf{y}^{(t)} + \varepsilon \cdot (\bar{1} - \mathbf{y}^{(t)}) \odot \mathbf{s}^{(t)}$.
5     $t \leftarrow t + \varepsilon$.
6 **return** $\mathbf{y}^{(1)}$.

---

To analyze Algorithm 7 we need to define two additional parameters. The first parameter is the diameter $D = \max_{x \in P} \|\mathbf{x}\|_2$ of $P$, which is a standard parameter. The other parameter is the monotonicity ratio of $F$, which can be extended to the continuous setting we study in the following natural way.[10]

$$m = \inf_{\substack{\mathbf{x}, \mathbf{y} \in \mathcal{X} \\ \mathbf{x} \leq \mathbf{y}}} \frac{F(\mathbf{y})}{F(\mathbf{x})} \quad,$$

where the ratio $F(\mathbf{y})/F(\mathbf{x})$ should be understood to have a value of 1 whenever $F(\mathbf{x}) = 0$. Additionally, let us denote by $\mathbf{o}$ an arbitrary optimal solution for the problem described above. Using these definitions, we are now ready to state the result that we prove for Algorithm 7.

**Theorem J.1.** *When given a non-negative $m$-monotone DR-submodular function $F\colon \mathcal{X} \to \mathbb{R}_{\geq 0}$ and a down-closed solvable convex body $P \subseteq \mathcal{X}$, the Measured Greedy Frank-Wolfe algorithm (Algorithm 7) outputs a solution $\mathbf{y} \in P$ such that $F(\mathbf{y}) \geq [m(1-1/e)+(1-m)\cdot(1/e)]\cdot F(\mathbf{o})-\varepsilon LD^2$.*

---

[9]In Section E.2, down-closeness of was defined for the special case of $P \subseteq [0,1]^{\mathcal{N}}$. More generally, a body $P \subseteq \mathcal{X}$ is down-closed if $\mathbf{b} \in P$ implies $\mathbf{a} \in P$ for every vector $\mathbf{a} \in \mathcal{X}$ obeying $\mathbf{a} \leq \mathbf{b}$.

[10]In Appendix 6.2 we showed how the monotonicity ratio can be extended to the particular continuous setting studied in that section. The definition of Appendix 6.2 is obtained from the more general definition we give here by setting $\mathcal{X} = \prod_{i=1}^n [0, u_i]$.

Our first objective towards proving Theorem J.1 is to lower bound the expression $F(\mathbf{o}+\mathbf{y}^{(t)}\cdot(\bar{1}-\mathbf{o}))$, which we do in the next two lemmata.

**Lemma J.2.** *For every integer $i \in [0, \varepsilon^{-1}]$, $\mathbf{y}^{(\varepsilon i)} \geq \bar{0}$ and $\|\mathbf{y}^{(\varepsilon i)}\|_\infty \leq 1 - (1-\varepsilon)^{-i}$.*

*Proof.* We prove the lemma by induction on $i$. For $i = 0$, the lemma follows directly from the initialization $\mathbf{y}^{(0)} = \bar{0}$ because $1 - (1-\varepsilon)^{-0} = 0$. Assume now that the lemma holds for $i - 1$, and let us prove it for an integer $0 < i \leq 1$. Observe that, for every $j \in [n]$,

$$y_j^{\varepsilon i} = y_j^{\varepsilon(i-1)} + \varepsilon \cdot \left(1 - y_j^{\varepsilon(i-1)}\right) \cdot s_j^{\varepsilon(i-1)} \geq y_j^{\varepsilon(i-1)} \geq 0 \ ,$$

where the first inequality holds since $y_j^{\varepsilon(i-1)} \leq 1$ by the induction hypothesis and the value of $s_j^{(\varepsilon(i-1))}$ is non-negative by definition. Moreover,

$$y_j^{\varepsilon i} = y_j^{\varepsilon(i-1)} + \varepsilon \cdot \left(1 - y_j^{\varepsilon(i-1)}\right) \cdot s_j^{\varepsilon(i-1)} \leq y_j^{\varepsilon(i-1)} + \varepsilon \cdot \left(1 - y_j^{\varepsilon(i-1)}\right)$$

$$= \varepsilon + (1-\varepsilon) \cdot y_j^{\varepsilon(i-1)} \leq \varepsilon + (1-\varepsilon) \cdot \left[1 - (1-\varepsilon)^{(i-1)}\right] = 1 - (1-\varepsilon)^i \ ,$$

where again the first inequality holds since $s^{(\varepsilon(i-1))} \in \mathcal{X}$, which implies $s_j^i \leq 1$; and the second inequality holds by the induction hypothesis. $\square$

**Lemma J.3.** *For every integer $i \in [0, \varepsilon^{-1}]$, $F(\mathbf{o}+\mathbf{y}^{(\varepsilon i)}\cdot(\bar{1}-\mathbf{o})) \geq \left[(1 - (1-m)(1-(1-\varepsilon)^i)\right] \cdot F(\mathbf{o}) = \left[m + (1-m)(1-\varepsilon)^i\right] \cdot F(\mathbf{o})$.*

*Proof.* Observe that

$$F(\mathbf{o} + \mathbf{y}^{(\varepsilon i)} \cdot (\bar{1} - \mathbf{o})) \geq \left(1 - \|\mathbf{y}^{(\varepsilon i)}\|_\infty\right) \cdot F(\mathbf{o}) + \|\mathbf{y}^{(\varepsilon i)}\|_\infty \cdot F\left(\mathbf{o} + \frac{\mathbf{y}^{(\varepsilon i)} \cdot (\bar{1} - \mathbf{o})}{\|\mathbf{y}^{(\varepsilon i)}\|_\infty}\right)$$

$$\geq \left(1 - \|\mathbf{y}^{(\varepsilon i)}\|_\infty\right) \cdot F(\mathbf{o}) + m \cdot \|\mathbf{y}^{(\varepsilon t)}\|_\infty \cdot F(\mathbf{o})$$

$$= \left(1 - (1-m) \cdot \|\mathbf{y}^{(\varepsilon i)}\|_\infty\right) \cdot F(\mathbf{o}) \ ,$$

where the first inequality holds since the DR-submodularity of $F$ implies that $F$ is concave along positive directions (such as the direction $\mathbf{y}^{(\varepsilon i)} \cdot (\bar{1} - \mathbf{o})/\|\mathbf{y}^{(\varepsilon i)}\|_\infty$), and the second inequality holds since the monotonicity ratio of $F$ is at least $m$. Plugging Lemma J.2 into the previous inequality completes the proof of the lemma. $\square$

Using the previous lemma, we can now provide a lower bound on the increase in the value of $\mathbf{y}^{(t)}$ as a function of $t$.

**Lemma J.4.** *For every integer $0 \leq i < \varepsilon^{-1}$, $F(\mathbf{y}^{(\varepsilon(i+1))}) - F(\mathbf{y}^{(\varepsilon i)}) \geq \varepsilon \cdot [(m + (1-m) \cdot (1-\varepsilon)^i) \cdot F(\mathbf{o}) - F(\mathbf{y}^{(\varepsilon i)})] - \varepsilon^2 L D^2$.*

*Proof.* By the chain rule,

$$F(\mathbf{y}^{\varepsilon(i+1)}) - F(\mathbf{y}^{(\varepsilon i)}) = F(\mathbf{y}^{(\varepsilon i)} + \varepsilon \cdot \mathbf{s}^{(\varepsilon i)} \odot (\bar{1} - \mathbf{y}^{(\varepsilon i)})) - F(\mathbf{y}^{(\varepsilon i)})$$

$$= \int_0^\varepsilon \nabla F(\mathbf{y}^{(\varepsilon i)} + r \cdot \mathbf{s}^{(\varepsilon i)} \odot (\bar{1} - \mathbf{y}^{(\varepsilon i)})) \cdot (\mathbf{s}^{(\varepsilon i)} \odot (\bar{1} - \mathbf{y}^{(\varepsilon i)})) \, dr$$

$$\geq \int_0^\varepsilon \nabla F(\mathbf{y}^{(\varepsilon i)}) \cdot (\mathbf{s}^{(\varepsilon i)} \odot (\bar{1} - \mathbf{y}^{(\varepsilon i)})) \, dr - \varepsilon^2 L D^2$$

$$= \varepsilon \cdot \nabla F(\mathbf{y}^{(\varepsilon i)}) \cdot (\mathbf{s}^{(\varepsilon i)} \odot (1 - \mathbf{y}^{(\varepsilon i)})) - \varepsilon^2 L D^2 \ ,$$

where the first inequality holds by the $L$-smoothness of $F$. Furthermore,

$$\nabla F(\mathbf{y}^{(\varepsilon i)}) \cdot (\mathbf{s}^{(\varepsilon i)} \odot (\bar{1} - \mathbf{y}^{(\varepsilon i)})) = ((\bar{1} - \mathbf{y}^{(\varepsilon i)}) \odot \nabla F(\mathbf{y}^{(\varepsilon i)})) \cdot \mathbf{s}^{(\varepsilon i)}$$

$$\geq ((\bar{1} - \mathbf{y}^{(\varepsilon i)}) \odot \nabla F(\mathbf{y}^{(\varepsilon i)})) \cdot \mathbf{o}$$

$$= \nabla F(\mathbf{y}^{(\varepsilon i)})) \cdot ((\bar{1} - \mathbf{y}^{(\varepsilon i)}) \odot \mathbf{o})$$

$$\geq F(\mathbf{o} + \mathbf{y}^{(\varepsilon i)}(\bar{1} - \mathbf{o})) - F(\mathbf{y}^{(\varepsilon i)})$$

$$\geq \left[m + (1-m) \cdot (1-\varepsilon)^i\right] \cdot F(\mathbf{o}) - F(\mathbf{y}^{(\varepsilon i)}) \ ,$$

where the first inequality holds by the definition of $\mathbf{s}^{(\varepsilon i)}$ since $\mathbf{o}$ is a candidate to be this vector, the second inequality follows from the concavity of $F$ along positive directions, and the last inequality holds by Lemma J.3. The lemma now follows by combining the two above inequalities. $\square$

We are now ready to prove Theorem J.1.

*Proof of Theorem J.1.* Rearranging the guarantee of Lemma J.4, we get

$$F(\mathbf{y}^{\varepsilon(i+1)}) \geq (1 - \varepsilon) \cdot F(\mathbf{y}^{(\varepsilon i)}) + \varepsilon[m + (1 - m) \cdot (1 - \varepsilon)^i] \cdot F(\mathbf{o}) - \varepsilon^2 LD^2 \ .$$

Since this inequality applies for every integer $0 \leq i < \varepsilon^{-1}$, we can use it repeatedly to obtain

$$F(\mathbf{y}^{(1)}) \geq \varepsilon \cdot \sum_{i=1}^{1/\varepsilon} (1 - \varepsilon)^{1/\varepsilon - i} \cdot \left[(m + (1 - m) \cdot (1 - \varepsilon)^{i-1}) \cdot F(\mathbf{o}) - \varepsilon LD^2\right] + (1 - \varepsilon)^{1/\varepsilon} \cdot F(\bar{0})$$

$$\geq m\varepsilon \cdot \sum_{i=1}^{1/\varepsilon} (1 - \varepsilon)^{\frac{1}{\varepsilon} - i} \cdot F(\mathbf{o}) + \varepsilon(1 - m) \cdot \sum_{i=1}^{1/\varepsilon} [(1 - \varepsilon)^{1/\varepsilon - 1} \cdot F(\mathbf{o}) - \varepsilon LD^2]$$

$$= m\varepsilon \cdot \frac{1 - (1 - \varepsilon)^{1/\varepsilon}}{\varepsilon} \cdot F(\mathbf{o}) + \varepsilon(1 - m) \cdot \frac{(1 - \varepsilon)^{1/\varepsilon - 1} \cdot F(\mathbf{o}) - \varepsilon LD^2}{\varepsilon}$$

$$\geq \left[m(1 - e^{-1}) + (1 - m) \cdot e^{-1}\right] \cdot F(\mathbf{o}) - \varepsilon LD^2 \ ,$$

where the second inequality follows from the non-negativity of $F$, and the last inequality holds since $(1 - \varepsilon)^{1/\varepsilon} \leq e^{-1} \leq (1 - \varepsilon)^{1/\varepsilon - 1}$. $\square$