# OpenReview forum: "Using Partial Monotonicity in Submodular Maximization"
_NeurIPS.cc/2022/Conference — NeurIPS 2022 Accept_

### Official Review · Reviewer_8Qqa · 2022-06-23

**Rating:** 7
**Confidence:** 4
**Soundness:** 3 good
**Presentation:** 3 good
**Contribution:** 3 good

**Summary:**

In this paper, the authors consider the problem of maximizing submodular functions that are non-monotone. Non-monotone submodular optimization is typically more challenging than monotone submodular maximization. Although several algorithms are known for non-monotone submodular maximization, they are typically slow and they yield poor approximation guarantees. The author(s) address this problem, by proposing the definition of "m-monotonicity". This definition generalizes monotonicity, by quantifying the degree to which a function is far from being monotone.

The author(s) propose to incorporate the notion of m-monotonicity in the analysis of common submodular maximization algorithms, to improve state-of-the-art guarantees. The author(s) demonstrate the applicability of their framework to two relevant applications, namely Movie Recommendation and Quadratic Programming. On this instances, the approximation guarantee achieved in this paper is better than the state-of-the art.

**Questions:**

- Can you please describe technical challenges/contributions of this paper?
- Do you think that your upper-bounds can be further improved? Do these problems admit a lower-bound parameterized by m?

**Limitations:**

The authors adequately addressed limitations, given that the contribution of this paper is mostly theoretical.

**Strengths And Weaknesses:**

Maximizing non-monotone submodular functions is difficult in practice, particularly on large and heterogeneous datasets. Hence, this submission addresses a relevant research question. I think that the paper is overall well-written and easy to follow. The experiments show an improvement over the state-of-the art.

However, this submission also has weaknesses. (i) The authors only use their definition to study pre-existing algorithms. This leads me to believe that the technical contribution is limited, despite the many theoretical results provided in the appendix. It seems to me that m-monotonicity offers a simple way to generalize many known results, without making significant changes to the original proofs. (ii) It is unclear to me if their bound admits a matching lower bound.

I believe that this paper is overall a good submission. If the authors address my concerns, I am willing to raise my score.

-------------------- After the rebuttal:

I believe that the author(s) have addressed all my concerns and questions. It is clear that the contribution of this paper is novel and non-trivial. I believe that this submission is clearly written, and accessible to a broad audience. For these reasons, I am raising my score.

---

> ### Author Response · Authors · 2022-08-02
> **Response to Reviewer 8Qqa: Part 1. Opening statement, contribution of the m-monotonicity ratio.**
>
> **We would like to thank the reviewer for the important questions raised. Below we have answered all these questions, and we hope they address the reviewer's concerns. We certainly feel that these questions shed light on important motivational questions regarding the paper, and we will make sure to discuss them in more detail in the introduction of the paper itself for the benefit of future readers. We also appreciate the reviewer’s willingness to change the score if their concerns are sufficiently addressed. This is indeed motivating.**
>
> **Q1:** The authors only use their definition to study pre-existing algorithms. This leads me to believe that the technical contribution is limited, despite the many theoretical results provided in the appendix. It seems to me that m-monotonicity offers a simple way to generalize many known results, without making significant changes to the original proofs.
>
> **A1:** We agree that many existing proofs can be adapted to take advantage of the monotonicity ratio, although in some cases this requires non-trivial effort (the double greedy algorithm is the most significant example of that, and in fact, after the submission of the paper, we were able to show that a more sophisticated analysis of this algorithm shows that its approximation guarantee matches our inapproximability result). We view that as a positive thing, since it shows that our definition of the monotonicity ratio is useful in many cases rather than being tailored to the needs of a particular algorithm. We also would like to mention that the well-known submodularity ratio was originally analyzed only in the context of the standard greedy algorithm, and new algorithms designed to take advantage of it in a more sophisticated way appeared only in later works.

---

> ### Author Response · Authors · 2022-08-02
> **Response to Reviewer 8Qqa: Part 2. Matching lower bounds.**
>
> **Q2:** It is unclear to me if their bound admits a matching lower bound.
>
> **A2:** As mentioned above, for the unconstrained setting we now have tight bounds (which we will of course include in the updated version of the paper). For the constrained settings, our inapproximability bounds do not match the algorithmic bounds, which is similar to the state-of-the-art situation in general non-monotone submodular maximization. It is our hope that the fine-grained study allowed by the monotonicity ratio will shed light on why obtaining such bounds is so much more difficult for non-monotone submodular maximization as opposed to monotone submodular maximization.

---

> ### Author Response · Authors · 2022-08-02
> **Response to Reviewer 8Qqa: Part 3. Technical challenges and contributions of our work.**
>
> **Q3:** Can you please describe technical challenges/contributions of this paper?
>
> **A3:** As mentioned above, the analysis of the double greedy algorithm is quite technically involved, especially if one aims for a tight approximation guarantee. Another place in which our algorithmic results required significant technical work was in the random greedy for matroids algorithm. It is quite straightforward to design a version of this algorithm for the case in which the monotonicity ratio is known. However, handling the case in which it is unknown, without losing too much in the approximation guarantee, requires some non-trivial observations. In addition to the above, the main technical difficulty in the paper was probably extending the symmetry gap framework to the case of any fixed monotonicity ratio. Unfortunately, since this framework is very technical, the work that we had to put into this ended up in the appendix. However, this work was quite involved since the original proof of the framework is highly based on derivatives, and the monotonicity ratio cannot be easily restated in terms of derivatives. To see why that is the case, notice that a function can have a monotonicity ratio close to 1, even in the presence of very negative derivatives, as long as these derivatives do not occur over too long sections.

---

> ### Author Response · Authors · 2022-08-02
> **Response to Reviewer 8Qqa: Part 4. Improving our upper bounds.**
>
> **Q4:** Do you think that your upper-bounds can be further improved?
>
> **A4:** As mentioned above, we have an improvement in the approximation guarantee for the unconstrained setting, which will be included in the updated version of the paper. The approximation guarantees for the two other settings can probably be improved by incorporating the monotonicity ratio into the involved (but not very practical) algorithm of [1]. However, we chose to analyze more practical algorithms, even if they lead to slightly worse guarantees.
>
> [1] Niv Buchbinder and Moran Feldman. Constrained Submodular Maximization via a Non-symmetric Technique. Mathematics of Operations Research 44(3):988-1005, August 2019.

---

> ### Author Response · Authors · 2022-08-02
> **Response to Reviewer 8Qqa: Part 5. Lower bounds parametrized by m.**
>
> **Q5:** Do these problems admit a lower-bound parameterized by m?
>
> **A5:** We are not sure about the intention of this question. The paper includes such lower bounds for all three settings. For the unconstrained setting, this is proved as part of Theorem 3.1; for cardinality constraints, this is proved by Theorem D.4 (and depicted in Figure 1a in the main body of the paper); and for matroid constraints, this is proved by Theorem E.8 (and depicted in Figure 1b in the main body of the paper).

---

### Official Review · Reviewer_38EM · 2022-07-06

**Rating:** 7
**Confidence:** 4
**Soundness:** 4 excellent
**Presentation:** 3 good
**Contribution:** 4 excellent

**Summary:**

This paper studies a new, continuous notion of (non-)monotonicity for submodular functions. The notion in question is called monotonicity ratio and is defined as the minimum $m$ such that $m \cdot f(S) \le f(T)$ for all $S \subseteq T$. Clearly for $m=1$ we obtain the standard definition of monotonicity, while as m decreases, the more non-monotonicity the function exhibits.

The main contribution of the paper (apart from the definition in itself) is the analysis of well known algorithms for submodular maximization in terms of this new parameter. More precisely, the authors offer the following results:
- Double greedy for unconstrained submodular maximization is proved to have an approximation factor of $\max(m,(2+m)/4)$. The authors complement this result with a non tight lower bound that depends on m.
- For cardinality constraints, it is shown that the approximation of greedy is $m(1-1/e)$, while for random greedy is $m (1-1/e) + (1-m)/e$
- For matroid constraints, it is shown tha the approximation of greedy is m/2, for measured continuous greedy is approx $m (1-1/e) + (1-m)/e$. A result for random greedy is also provided.

Finally, the authors run experiments for personalized movie recommendation and quadratic programming.

**Questions:**

No questions

**Limitations:**

Other comment to the authors:
- line 10, continuous is misspelled


**Strengths And Weaknesses:**

This is a good paper that closes an important gap in the understanding of submodular functions. Studying the degree of submodularity or linearity of a function has already been proved to be a fruitful field of study; studying the degree of non-monotonicity will probably be as important.

Among the many positive results, I particularly like the fact that this paper offers the first (to the best of my knowledge) formal justification of the good empirical performances of greedy on non-monotone instances.

The paper is well written and offers a clear introduction to the problem for non-expert readers. I have only two suggestions for the authors. First, it would be interesting to give at least an idea of techniques used in the vast appendix; in particular it is a bit strange to give the definition of continuous relaxation in the intro and never use it in the body of the paper. Second, it is important to give some more arguments to the actual applicability of the monotonicity ratio. Even though some examples of upper bounds have been given in the experiments, the paper would benefit from other examples and/or general routines to compute (even a rough upper bound to ) the monotonicity ratio.

---

> ### Author Response · Authors · 2022-08-02
> **Response to Reviewer 38EM: Opening statement, addressing the reviewer's suggestions.**
>
> **We would like to thank the reviewer for the positive evaluation of our work and favorable expert review.**
>
> **Q1:** I have only two suggestions for the authors. First, it would be interesting to give at least an idea of techniques used in the vast appendix; in particular it is a bit strange to give the definition of continuous relaxation in the intro and never use it in the body of the paper. Second, it is important to give some more arguments to the actual applicability of the monotonicity ratio. Even though some examples of upper bounds have been given in the experiments, the paper would benefit from other examples and/or general routines to compute (even a rough upper bound to ) the monotonicity ratio.
>
> **A1:** We will add to the updated version of the paper some discussion of the technical ideas as suggested. There are of course additional examples that demonstrate the applicability of the monotonicity ratio, but given the space constraints, we have chosen to keep only 3 applications. We will be happy to add 2 additional applications to the updated version of the paper (distributed Ride-Share Optimization [1] when a diversity requirement is added to the problem, and Movie Summarization [2]). We also note that it is possible to obtain general bounds on the monotonicity ratio when additional function properties are assumed (such as second-order submodularity or supermodularity). Again, we will add some details about that to the updated version of the paper.
>
> [1] Kaushal et al. Demystifying multi-faceted video summarization: Tradeoff between diversity, representation, coverage and importance. In WACV, pages 452–461. IEEE, 2019.
>
> [2] Mitrovic et al. Data summarization at scale: A two-stage submodular approach. In ICML, pages 3593–3602. PMLR, 2018.

---

### Official Review · Reviewer_rgKu · 2022-07-06

**Rating:** 4
**Confidence:** 5
**Soundness:** 3 good
**Presentation:** 2 fair
**Contribution:** 2 fair

**Summary:**

This paper introduces the notion of monotonicity ratio $m\in [0,1]$ for submodular functions. Submodular maximization has been widely studied in the literature under the assumption that the function is either monotone or non-monotone. The authors study various standard submodular maximization algorithms for either of these two settings and show that new guarantees could be obtained that depend on the monotonicity ratio of the function. Finally, this paper considers a number of applications, including movie recommendations, and obtains bounds for the monotonicity ratio of the objective function.

**Questions:**

I've already mentioned a few questions earlier. Below are a few more questions and comments:
- In the personalized movie recommendation example, all the similarity scores need to be non-negative for the objective function to be submodular and non-negative, however, in the paper, it is mentioned that these scores are all non-positive.
- Have you thought about alternative notions of monotonicity ratio that are also easy to compute? In that case, one can design algorithms that assume access to this parameter and can guarantee better results.

**Limitations:**

The limitations are addressed. The paper is mostly theoretical and does not need a discussion about potential negative societal impact.

**Strengths And Weaknesses:**

Strengths:
- First work that tries to extend the binary property of monotonicity of submodular functions to continuous properties.
- The bounds for monotonicity ratio in the motivating applications (particularly movie recommendation and personalized image summarization) are interesting.
- The inapproximability result for unconstrained submodular maximization (Theorem 3.1) is novel and significant.

Weaknesses:
- The paper is extremely hard to read and follow even for someone who is familiar with the literature on submodular maximization. All the results in Sections 3-5 correspond to standard submodular maximization algorithms, however, none of these algorithms have been properly introduced in the paper. A brief discussion about each of these algorithms is missing.
- Unlike the curvature, the monotonicity ratio of a submodular function is hard to compute, and therefore, obtaining bounds in terms of such a parameter may not be very useful. While the monotonicity ratio bounds provided for the motivating applications are interesting, such bounds could not be derived in general.
- The inapproximability bound for matroid constraints for large values of $m$ (including $m=1$) seems to be larger than the known $1-\frac{1}{e}$ bound for the monotone setting and this result is really surprising. The authors need to explain this result in more detail.
- Besides the inapproximability result for unconstrained submodular maximization (Theorem 3.1), the other results of this paper seem to be a straightforward extension of prior works. The authors need to highlight the challenges faced when proving these more general results and specify how they managed to overcome the challenges.

---

> ### Author Response · Authors · 2022-08-02
> **Response to Reviewer rgKu: Part 1. Opening statement, discussing submodular maximization algorithms.**
>
> **We would like to thank the reviewer for the detailed review. We hope that our response below addresses all your concerns about the paper.**
>
> **Q1:** All the results in Sections 3-5 correspond to standard submodular maximization algorithms, however, none of these algorithms have been properly introduced in the paper. A brief discussion about each of these algorithms is missing.
>
> **A1:** All the algorithms are presented in detail in the supplementary material. We will be happy to move some of this discussion to the main body of the paper in the updated version.

---

> ### Author Response · Authors · 2022-08-02
> **Response to Reviewer rgKu: Part 2. Computing the monotonicity ratio.**
>
> **Q2:** Unlike the curvature, the monotonicity ratio of a submodular function is hard to compute, and therefore, obtaining bounds in terms of such a parameter may not be very useful. While the monotonicity ratio bounds provided for the motivating applications are interesting, such bounds could not be derived in general.
>
> **A2:** Indeed, the curvature can be calculated efficiently. However, this is a rare case, as most continuous properties studied are difficult to calculate, and probably the most significant example of that is the well-known submodularity ratio. Therefore, one can observe that such properties are influential even without a procedure for calculating them. As you have mentioned, our applications show that in specific cases it is possible to bound the monotonicity ratio. In fact, one can describe quite general conditions allowing for such bounds. For example, when the objective function is either second-order submodular or second-order supermodular (i.e., the third derivatives are either non-positive or non-negative), it is possible to optimize f(S | T) as a function of T, which yields a non-trivial bound also on the monotonicity ratio.

---

> ### Author Response · Authors · 2022-08-02
> **Response to Reviewer rgKu: Part 3. Inapproximability bound for matroid constraints.**
>
> **Q3:** The inapproximability bound for matroid constraints for large values of m (including m=1) seems to be larger than the known bound for the monotone setting and this result is really surprising. The authors need to explain this result in more detail.
>
> **A3:** Indeed, our inapproximability is weaker than the known bound for m = 1 (and values close to it), and this is explicitly discussed in the last paragraph of Section 5. From a technical point of view, it is important to mention that the known bound still applies to our setting. However, the fact that our proof was not able to recover it (or to improve over it for m values that are slightly smaller than 1) is an important observation. We hope that this hurdle that we have encountered will help future research by shedding light on the reasons why obtaining inapproximability results for non-monotone submodular maximization problems is so difficult (we are not aware of any known tight such bound in a constraint setting).

---

> ### Author Response · Authors · 2022-08-02
> **Response to Reviewer rgKu: Part 4. Challenges faced in our work.**
>
> **Q4:** Besides the inapproximability result for unconstrained submodular maximization (Theorem 3.1), the other results of this paper seem to be a straightforward extension of prior works. The authors need to highlight the challenges faced when proving these more general results and specify how they managed to overcome the challenges.
>
> **A4:** Indeed, some of the results of the paper are relatively straightforward extensions of existing work (but still obtain interesting new guarantees). However, other results required a novel technical contribution. The inapproximability for unconstrained submodular maximization, mentioned by the reviewer, is a prominent example of this. More generally, the extension of the symmetry gap technique underlying all our inapproximability results was non-trivial due to difficulty in stating the monotonicity ratio in terms of partial derivatives (as discussed in the second paragraph on Page 3). On the algorithmic side, adapting Random Greedy for Matroids (Algorithm 5) to work without knowledge of the monotonicity ratio required significant technical work. Furthermore, in the time since the submission of the paper, we have improved the analysis of the approximation ratio of double greedy, and our improved analysis is able to show that double greedy matches our inapproximability bound for the unconstrained case.

---

> ### Author Response · Authors · 2022-08-02
> **Response to Reviewer rgKu: Part 5. Similarity scores in the persionalized movie recommendation example.**
>
> **Q5:** In the personalized movie recommendation example, all the similarity scores need to be non-negative for the objective function to be submodular and non-negative, however, in the paper, it is mentioned that these scores are all non-positive.
>
> **A5:** This is a typo. Thank you for pointing it to us.

---

> ### Author Response · Authors · 2022-08-02
> **esponse to Reviewer rgKu: Part 6. Alternative notions of the monotonicity ratio.**
>
> **Q6:** Have you thought about alternative notions of monotonicity ratio that are also easy to compute? In that case, one can design algorithms that assume access to this parameter and can guarantee better results.
>
> **A6:** During our research, we have considered multiple properties for quantifying the monotonicity of set functions. One important example was based on a decomposition of the function into a monotone component and a non-monotone component. This definition was motivated by the well-known curvature, which can be presented as such a decomposition. However, we decided to drop the decomposition idea for two reasons. First, unlike in the case of the curvature, it did not seem to be efficiently calculable; and second, we were able to show that the monotonicity ratio as defined in the paper always leads to a better guarantee compared to every decomposition of the above-mentioned kind.

---

> ### Author Response · Authors · 2022-08-07
> **Please inform us of any concerns**
>
> Dear Reviewer rgKu,
>
> As the discussion period is coming to an end, we would like to reach out again and inquire about any potential lingering concerns you might have after reading our response. We are more than happy to continue addressing any issues you might have.

---

> > ### Comment · Reviewer_rgKu · 2022-08-08
> > **Re: Response**
> >
> > Thanks for your detailed answers to my questions. I agree that the inapproximability results in the paper are pretty significant and challenging. However, I'm still not sure about the current definition of monotonicity ratio (which is the reason for the "difficulty in stating the monotonicity ratio in terms of partial derivatives" challenge you mentioned). You said that the submodularity ratio is a prominent example of the fact that "such properties are influential even without a procedure for calculating them". However, for particular classes of non-submodular functions including BP functions, it is still possible (and strongly desirable) to obtain bounds that are in terms of calculable parameters. I believe that defining parameters that are hard to compute (including the submodularity ratio) and obtaining approximation bounds in terms of these parameters simply masks the hardness of the problem and is not necessarily influential.
> >
> > I slightly increased my score for now, but I need further discussion with other reviewers to finalize my review. I don't have any further questions for the author-reviewer discussion period.

---

### Official Review · Reviewer_ZNvk · 2022-07-11

**Rating:** 7
**Confidence:** 3
**Soundness:** 4 excellent
**Presentation:** 4 excellent
**Contribution:** 3 good

**Summary:**

This paper defines a new notion of the monotonicity ratio for submodular maximization. The authors analyze how the approximation ratios of standard submodular maximization algorithms (e.g., greedy) depend on this ratio under the unconstrained settings/cardinality constraints/matroid constraints, and then demonstrated that this leads to improved approximation guarantees for the applications of movie recommendation, image summarization, and quadratic programming. The authors also propose lower bounds depending on the monotonicity ratio.

**Questions:**

- This paper mainly considers greedy-based algorithms. Why not also considers local search-based algorithms?

**Limitations:**

Yes

**Strengths And Weaknesses:**

Strengths:
- The notion of the monotonicity ratio is novel and interesting for submodular maximization.
- The analysis of greedy-based algorithms is solid and convinces the reasonability of the monotonicity ratio.
- The lower bound analysis is interesting. Specifically, I like the conclusion that in the unconstrained submodular maximization case, the optimal algorithm has an approximation ratio whose dependence on the monotonicity ratio $m$ is non-linear.
- The writing is nice, including a detailed discussion of the novelties and applications.

Weaknesses:
- It will be better if this paper can propose an algorithm that beats greedy w.r.t. the monotonicity ratio.

---------------------------------------------------------------------------------------------------------------------------------------------------------------------------------------
Thanks for the response. Could you update the improved analysis of the double greedy algorithm for the unconstrained case in the revised pdf? This will make the paper stronger.

---

> ### Author Response · Authors · 2022-08-02
> **Response to Reviewer ZNvk: Part 1. Opening statement, beating greedy.**
>
> **We would like to thank the reviewer for the positive evaluation of our work and favorable expert review.**
>
> **Q1:** It will be better if this paper can propose an algorithm that beats greedy w.r.t. the monotonicity ratio.
>
> **A1:** Almost all the known tight approximation ratios in submodular maximization come from greedy-based algorithms, and therefore, this is likely to be the case also when the monotonicity ratio is taken into account. In fact, since the submission of the current paper, we have been able to improve the analysis of the double greedy algorithm for the unconstrained case, and show that the real approximation ratio of this algorithm matches our inapproximability result, and is thus, tight. Of course, we will include the improved analysis in the updated version of the paper.

---

> ### Author Response · Authors · 2022-08-02
> **Response to Reviewer ZNvk: Part 2. Considering local search-based algorithms.**
>
> **Q2:** This paper mainly considers greedy-based algorithms. Why not also considers local search-based algorithms?
>
> **A2:** Almost all the current state-of-the-art algorithms for submodular maximization are greedy-based, and therefore, we chose to study how to analyze such algorithms using the monotonicity ratio. However, in theory, it should be possible to obtain improved approximation guarantees also for local search-based algorithms using the monotonicity ratio. In particular, the state-of-the-art algorithm for maximizing a non-monotone submodular function subject to a matroid constraint combines a greedy component and a local search component, and therefore, obtaining an approximation ratio for this algorithm that depends on the monotonicity ratio will require an analysis of both these components in the context of the monotonicity ratio. However, this will be highly technical, and therefore, we feel that it is not a good fit for a paper whose main objective is to introduce the monotonicity ratio and demonstrate its value.

---

> ### Author Response · Authors · 2022-08-08
> **Improved analysis of the double greedy algorithm**
>
> Dear reviewer ZNvk,
>
> We are happy to update you that we have uploaded a revised version including the proof of the double greedy algorithm (as you have requested). For ease of viewing the new material, the proof was uploaded as a supplementary material, but of course, we will add it to the paper itself in a camera ready version. We are glad to see that you believe this improved analysis makes the paper stronger. Thank you for the suggestion.

---

> > ### Comment · Reviewer_ZNvk · 2022-08-08
> > **Response to the authors**
> >
> > Got it. Thanks for your update. By the way, in Line 2 of Algorithm 1, it should be $Y_0\leftarrow \mathcal{N}$.

---

### Meta-Review · Area_Chair_iMud · 2022-08-25

**Recommendation:** Accept
**Confidence:** Certain

**Metareview:**

This paper introduces a new notion of partial monotonicity in submodular maximization that is interesting and relevant. The authors derive approximation ratios for existing algorithms that are as a function of a monotonicity ratio. This new notion is likely to be studied and relevant in future work.

One concern raised by a reviewer is regarding the fact that the monotonicity ratio of a function is hard to compute. However, this is also the case for other properties of submodular functions and this concern is adequately discussed in the paper.

**Award:**

No

---

### Decision · Program_Chairs · 2022-09-14

Accept